# High-throughput profiling of sequence recognition by tyrosine kinases and SH2 domains using bacterial peptide display

**Allyson Li[1], Rashmi Voleti[1], Minhee Lee[1], Dejan Gagoski[1,2], Neel H Shah[1]\***

[1]Department of Chemistry, Columbia University, New York, United States;
[2]Department of Biological Sciences, Columbia University, New York, United States

**Abstract** Tyrosine kinases and SH2 (phosphotyrosine recognition) domains have binding specificities that depend on the amino acid sequence surrounding the target (phospho)tyrosine residue. Although the preferred recognition motifs of many kinases and SH2 domains are known, we lack a quantitative description of sequence specificity that could guide predictions about signaling pathways or be used to design sequences for biomedical applications. Here, we present a platform that combines genetically encoded peptide libraries and deep sequencing to profile sequence recognition by tyrosine kinases and SH2 domains. We screened several tyrosine kinases against a million-peptide random library and used the resulting profiles to design high-activity sequences. We also screened several kinases against a library containing thousands of human proteome-derived peptides and their naturally-occurring variants. These screens recapitulated independently measured phosphorylation rates and revealed hundreds of phosphosite-proximal mutations that impact phosphosite recognition by tyrosine kinases. We extended this platform to the analysis of SH2 domains and showed that screens could predict relative binding affinities. Finally, we expanded our method to assess the impact of non-canonical and post-translationally modified amino acids on sequence recognition. This specificity profiling platform will shed new light on phosphotyrosine signaling and could readily be adapted to other protein modification/recognition domains.

**\*For correspondence:**
neel.shah@columbia.edu

**Competing interest:** The authors declare that no competing interests exist.

## Editor's evaluation

This paper reports an improved bacterial surface peptide display technology and its use to survey the primary sequence specificities of a broad range of tyrosine kinases and to assess the effects of naturally-occurring positional variations around sites of tyrosine phosphorylation on the efficiency of phosphorylation. The versatility of this approach was demonstrated by using expanded genetic code technology to investigate the consequences of installing post-translationally modified amino acids, such as acetyl-lysine, at positions upstream and downstream of a target tyrosine on the efficiency of phosphorylation by different tyrosine kinases. In addition, pre-phosphorylated surface peptide display libraries were exploited to interrogate the primary sequence binding specificities of SH2 phosphotyrosine-binding domains.

## Introduction

Cells respond to external stimuli by activating a finely-tuned cascade of enzymatic reactions and protein-protein interactions. This signal transduction is governed, in large part, by post-translational modifications that alter protein activity, stability, and localization, as well as the formation of higher-order macromolecular complexes. Despite its low abundance relative to serine and threonine phosphorylation, tyrosine phosphorylation is an essential post-translational modification in metazoans (*Lim*

*and Pawson, 2010*). Tyrosine kinases, the enzymes that phosphorylate tyrosine residues on proteins, and Src homology 2 (SH2) domains, protein modules that bind tyrosine-phosphorylated sequences, must have the ability to discriminate among a myriad of potential phosphorylation sites (phospho-sites) in the proteome, in order to ensure proper signal transduction. The preferential engagement of specific phosphosites by tyrosine kinases and SH2 domains is dependent on the amino acid sequence surrounding the tyrosine or phosphotyrosine residue (*Songyang et al., 1995*; *Songyang et al., 1993*).

Isolated tyrosine kinase domains most efficiently engage phosphosites that conform to specific sequence motifs, which are defined by a small number of key residues that contribute significantly to recognition (*Songyang et al., 1995*). These motifs suggest a mechanism by which a specific set of phosphosites in a proteome is selectively engaged by an individual kinase, based on the presence of favorable sequence features around that site. Negative selection of specific sequence features can also play a role in kinase specificity (*Alexander et al., 2011*). For example, the T cell tyrosine kinase ZAP-70 cannot readily phosphorylate co-localized proteins that contain even a modest positive charge (*Shah et al., 2016*).

Phosphosite sequence recognition by kinase domains is just one mechanism of substrate selection for tyrosine kinases, and other interactions are necessary to achieve efficient substrate targeting *in vivo*. Binding domains, such as SH2 domains, can strongly influence specificity by localizing kinases to the vicinity of phosphorylation targets (*Pawson and Nash, 2000*). Secondary interactions between SH2 and kinase domains can also refine the substrate preferences of a tyrosine kinase by stabilizing its active state (*Filippakopoulos et al., 2008*). Thus, for signaling systems that involve a tyrosine kinase domain and a tethered SH2 domain, the sequence specificities of both domains contribute to the intricate control of phosphotyrosine signaling responses.

Many methods have been developed to characterize sequence recognition by tyrosine kinases and SH2 domains. The most prominent approach employs purified kinases/SH2 domains and oriented peptide libraries, which are synthetic, degenerate peptide libraries with a central tyrosine or phos-photyrosine residue (*Songyang et al., 1995*; *Songyang et al., 1993*). Several variations on this tech-nique have been reported to improve the throughput and quantification of sequence preferences (*Deng et al., 2014*; *Huang et al., 2008*; *Hutti et al., 2004*; *Mok et al., 2010*). Notably, this method is also applicable to serine/threonine kinases, and large swaths of the yeast and human kinomes have been characterized using oriented peptide libraries, providing significant insights into kinase-substrate recognition and phospho-signaling (*Deng et al., 2014*; *Johnson et al., 2023*; *Mok et al., 2010*; *Songyang et al., 1995*). Oriented peptide library screens have primarily been useful for deter-mining the preference for each amino acid at a given position, independent of sequence context, but evidence suggests that some amino acid preferences may depend on the surrounding sequence (*Cantor et al., 2018*).

Several groups have developed strategies to compare the phosphorylation of specific sequences, rather than obtain position-averaged amino acid preferences from pooled degenerate libraries. Strat-egies include 'one-bead-one-peptide' combinatorial libraries (*Imhof et al., 2006*; *Ren et al., 2011*; *Sweeney et al., 2005*; *Trinh et al., 2013*; *Wavreille et al., 2007*) and protein/peptide microarrays (*Amanchy et al., 2008*; *Jones et al., 2006*; *Koytiger et al., 2013*; *Mok et al., 2009*; *Schutkowski et al., 2004*; *Uttamchandani et al., 2003*). One-bead-one-peptide methods often require manual isolation and individual sequencing of positive (phosphorylated or SH2-bound) beads, making the method technically challenging. Microarrays offer the capacity to analyze thousands of discrete sequences and require small quantities of proteins, but their use can be limited by the high cost of reagents. As an alternative, several groups have conducted mass spectrometry proteomics on heter-ologously expressed purified peptide libraries, kinase-treated cell extracts, and cells over-expressing a kinase of interest (*Barber et al., 2018*; *Chou et al., 2012*; *Corwin et al., 2017*; *Douglass et al., 2012*; *Finneran et al., 2020*; *Imamura et al., 2014*; *Kettenbach et al., 2012*; *Lubner et al., 2018*; *Sugiyama et al., 2019*; *Xue et al., 2012*). This strategy has enabled the identification of potential substrates and can also be used to infer position-specific amino acid preferences. Studies using intact proteomes have the added benefit that the kinase of interest is operating on intact proteins, rather than isolated peptides, but interpretation of the results can be convoluted by the presence of endog-enous kinases.

Molecular display techniques, such as mRNA, phage, yeast, and bacterial display, have also been used for specificity profiling. Early investigations employed phage or mRNA display to profile

tyrosine kinase and SH2 specificity. These methods were relatively low-throughput, as they relied on Sanger sequencing of individual clones (*Cujec et al., 2002*; *Dente et al., 1997*). The advent of deep sequencing technologies has transformed this style of specificity profiling, by enabling rapid, quantitative analysis of library composition without requiring the sequencing of individual clones. This was demonstrated recently in a series of studies that employed bacterial/yeast peptide display, fluorescence-activated cell sorting (FACS), and deep sequencing to profile tyrosine kinase and SH2 domain specificity (*Cantor et al., 2018*; *Lo et al., 2019*; *Shah et al., 2018*; *Shah et al., 2016*; *Taft et al., 2019*). A key facet of these investigations was the facile generation of peptide libraries tailored to specific mechanistic questions: these included scanning mutagenesis libraries derived from individual substrates (*Shah et al., 2016*), as well as diverse peptide libraries encoding known phospho-sites in the human proteome (*Shah et al., 2018*).

In this report, we describe a high-throughput platform to profile the recognition of large peptide libraries by any tyrosine kinase or SH2 domain. Our approach uses biotinylated bait proteins (pan-phosphotyrosine antibodies or SH2 domains) and avidin-functionalized magnetic beads to isolate tyrosine kinase-phosphorylated bacterial cells, and is coupled to deep sequencing for a quantitative readout (*Figure 1A*). The use of magnetic bead-based separation, rather than FACS, permits simultaneous, benchtop processing of multiple samples and enables the analysis of larger libraries for less time and cost. Libraries can be custom-made for specific readouts: mutational scanning for structure-activity relationships, libraries derived from natural proteomes to answer specific signaling questions, or degenerate libraries for the generation of predictive models.

To demonstrate the versatility of our approach, we designed two new bacterial peptide display libraries that provide distinct insights into tyrosine kinase and SH2 sequence recognition. The first library contains $10^6$–107 random 11-residue sequences with a central tyrosine (referred to as the $X_5$-Y-$X_5$ library). Screens with the $X_5$-Y-$X_5$ library recapitulate previously reported specificity motifs and can be used to generate highly efficient peptide substrates. The second library contains defined sequences spanning 3000 human tyrosine phosphorylation sites, along with 5000 variant sequences bearing disease-associated mutations and natural polymorphisms (referred to as the pTyr-Var library). Kinase and SH2 screens with the pTyr-Var library reveal hundreds of phosphosite-proximal mutations that significantly impact phosphosite recognition by individual protein domains. These datasets will be a valuable resource in the growing efforts to understand the functional impact of protein variants across the human population that may contribute to disease (*Stein et al., 2019*). Finally, we show that our peptide display platform is compatible with Amber codon suppression, enabling analysis of how non-canonical or post-translationally modified amino acids impact sequence recognition. Overall, the method described in this report provides an accessible, high-throughput platform to study the specificity of phosphotyrosine signaling proteins.

## Results and discussion

### A bacterial display and deep sequencing platform to screen tyrosine kinases against large peptide libraries

We expanded upon a previously established screening platform that combines bacterial display of genetically encoded peptide libraries and deep sequencing to quantitatively compare phosphorylation efficiencies across a substrate library (*Shah et al., 2016*). In the published approach, peptides are displayed on the surface of *E. coli* cells as fusions to an engineered bacterial surface-display protein, eCPX (*Rice and Daugherty, 2008*), then phosphorylated by a purified kinase (*Henriques et al., 2013*). Following this, the cells are labeled with a pan-phosphotyrosine antibody, and cells with high phosphorylation levels are separated by FACS. The DNA encoding the peptides is then amplified and analyzed by Illumina deep sequencing to determine the frequency of each peptide in the library before and after selection (*Shah et al., 2018*; *Shah et al., 2016*). In order to determine the phosphorylation efficiency of each peptide by a particular kinase, an enrichment score is determined by calculating the frequency of that peptide in the kinase-selected sample normalized to the frequency in the input sample.

While peptide libraries of virtually any composition can theoretically be screened using this approach, previous implementations focused on libraries containing less than 5000 peptides, due to the low throughput of FACS (*Shah et al., 2018*). In those experiments, the objective was to

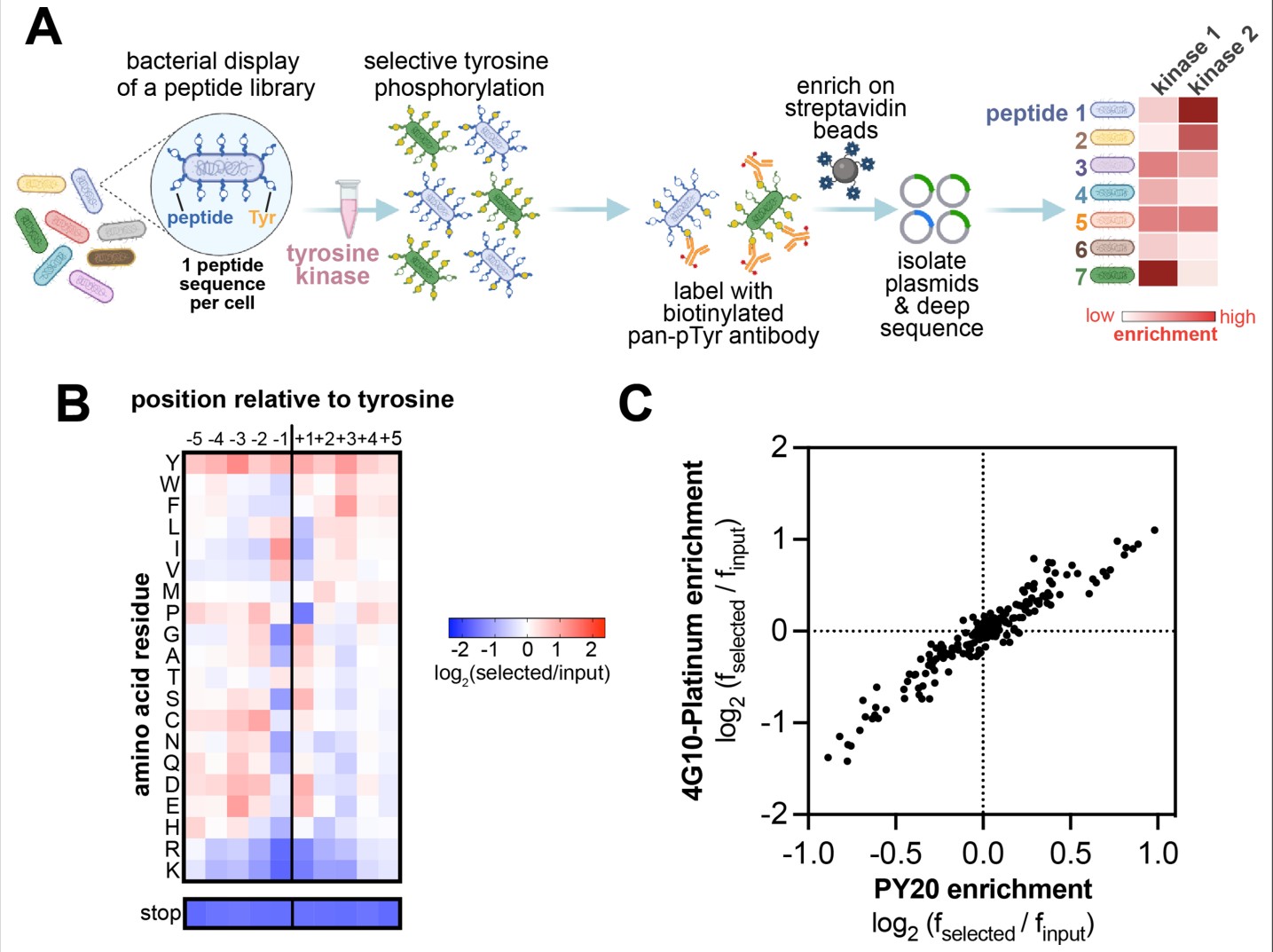

**Figure 1.** High-throughput profiling of tyrosine kinase substrate specificity using bacterial peptide display. (**A**) Schematic representation of the workflow for kinase specificity profiling. (**B**) Heatmap depicting the specificity of the c-Src kinase domain, measured using the $X_5$-Y-$X_5$ library. Enrichment scores were $log_2$-transformed and are displayed on a color scale from blue (disfavored sequence features, negative value), to white (neutral sequence features, near zero value), to red (favored sequence features, positive value). Values in the heatmap are the average of three replicates. (**C**) Correlation between position-specific amino acid enrichments from screens with the 4G10 Platinum and PY20 biotinylated pan-phosphotyrosine antibodies.

The online version of this article includes the following source data and figure supplement(s) for figure 1:

**Figure supplement 1.** Composition of the $X_5$-Y-$X_5$ library.

**Figure supplement 1—source data 1.** Counts table corresponding to one sequence run from an input $X_5$-Y-$X_5$ library.

**Figure supplement 2.** Phosphorylation of the $X_5$-Y-$X_5$ library by c-Src.

**Figure supplement 3.** Heatmap and logo depicting the specificity of the c-Src kinase domain, measured using the $X_5$-Y-$X_5$ library.

over-sample the library at the cell sorting step by a factor of 100–1000, to ensure that enrichment or depletion of every member of the library could be accurately quantified by deep sequencing. When multiple screens were conducted in parallel, the throughput of FACS limited experiments to small libraries (less than 5000 sequences). To improve the scalability and cost-effectiveness of this approach, we switched to a bead-based sorting method, using avidin-coated magnetic beads to enrich highly-phosphorylated cells, thus circumventing the need for FACS (**Figure 1A**). With this approach, the cells are instead labeled with biotinylated pan-phosphotyrosine antibodies and then sorted using magnetic beads. The use of magnetic beads permits simultaneous separation of multiple samples of virtually

any size, enabling larger library analysis for less time and cost. Notably, these screens can be carried out in any laboratory, without the need for a fluorescence-activated cell sorter.

To test our upgraded screening platform, we generated a random library of 11-residue sequences with a central tyrosine (the $X_5$-Y-$X_5$ library, where X is any of the 20 canonical amino acids). The library was generated using a degenerate synthetic oligonucleotide with five NNS codons (N=A,T,G,C and S=G,C) before and after the central codon that encodes for tyrosine (TAT). The NNS triplet has the benefit of encoding all 20 amino acids, but it can still contain an Amber stop codon (TAG) roughly 3% of the time. Therefore, up to 30% of the peptide-coding sequences in the library are expected to have an Amber stop codon – a feature that we take advantage of later in this study. The degenerate oligonucleotide mixture was cloned into a plasmid in between the DNA encoding a signal sequence and the eCPX surface-display scaffold. In a previously reported version of this platform, the eCPX scaffold contained a C-terminal strep-tag to detect surface-display level (**Shah et al., 2016**). Due to the potential background binding of the strep-tag with the avidin-coated magnetic beads during cell enrichment, we cloned both a strep-tagged and a myc-tagged version of the library. Deep sequencing of both versions of the $X_5$-Y-$X_5$ library confirmed that they have 1–10 million unique peptide sequences, 20% of which contain one or more stop codons. Furthermore, all 20 canonical amino acids were well-represented at each of the 10 variable positions surrounding the fixed tyrosine residue (**Figure 1—figure supplement 1**). Notably, our library includes peptides containing Cys residues and non-central Tyr residues, both of which are often excluded from tyrosine kinase specificity screens to avoid oxidation-related artifacts and challenges in interpreting signal from multi-Tyr sequences (**Deng et al., 2014**). These sequences can be filtered during data analysis, if needed, although they did not pose significant issues in our studies.

Using the myc-tagged $X_5$-Y-$X_5$ library, we determined the position-specific amino acid preferences of the kinase domain of c-Src. Cells displaying the library were phosphorylated by c-Src to achieve roughly 20–30% phosphorylation, as determined by flow cytometry (**Figure 1—figure supplement 2**). The phosphorylated cells were labeled with a biotinylated anti-phosphotyrosine antibody and enriched with magnetic beads, then peptide-coding DNA sequences were counted by deep sequencing. We visualized the sequence preferences of c-Src by generating a heatmap and sequence logo based on the position-specific enrichment scores of each amino acid residue surrounding the central tyrosine (**Figure 1B**, **Figure 1—figure supplement 3**). Sequences containing a stop codon were not considered in these calculations, but the depletion of stop codons at each position was separately confirmed and is reported below the heatmap on the same color scale. The preferences determined from this screen matched the sequence specificity of c-Src defined by prior reports using oriented peptide libraries (**Deng et al., 2014**; **Songyang et al., 1995**). We observed a strong preference for bulky aliphatic residues (Ile/Leu/Val) at the −1 position relative to the central tyrosine and a phenylalanine at the +3 position (**Figure 1B**, **Figure 1—figure supplement 3**). Our results showed modest differences from the specificity observed by oriented peptide libraries, including a strong preference for a+1 Asp/Glu/Ser in addition to the previously reported +1 Gly. To test whether these differences were due to biases introduced by the specific pan-phosphotyrosine antibody used, we obtained a different commercially available biotinylated pan-phosphotyrosine antibody and repeated the screen. The position-specific amino acid enrichments obtained using both antibodies were nearly identical (**Figure 1C**). This suggests that there is no significant bias in the enrichment of peptides introduced by the pan-phosphotyrosine antibody.

## Degenerate library screens capture specificity profiles for diverse tyrosine kinases

We next used the degenerate $X_5$-Y-$X_5$ library to characterize the sequence preferences of four additional tyrosine kinase domains, derived from the non-receptor tyrosine kinases c-Abl and Fer, and the receptor tyrosine kinases EPHB1 and EPHB2. The kinases were selected because they represent a few distinct branches of the tyrosine kinome and can be easily produced through bacterial expression (**Albanese et al., 2018**). The $X_5$-Y-$X_5$ library was screened against the kinases in triplicate, and the data from replicates were averaged to generate specificity profiles for each kinase (**Figure 2A** and **Figure 2—source data 1**). The amino acid preferences for c-Abl are well-characterized and were recapitulated in this screen (**Deng et al., 2014**; **Songyang et al., 1995**; **Till et al., 1999**; **Till et al., 1994**). Like c-Src, c-Abl preferred bulky aliphatic residues at the −1 position with respect to the central tyrosine.

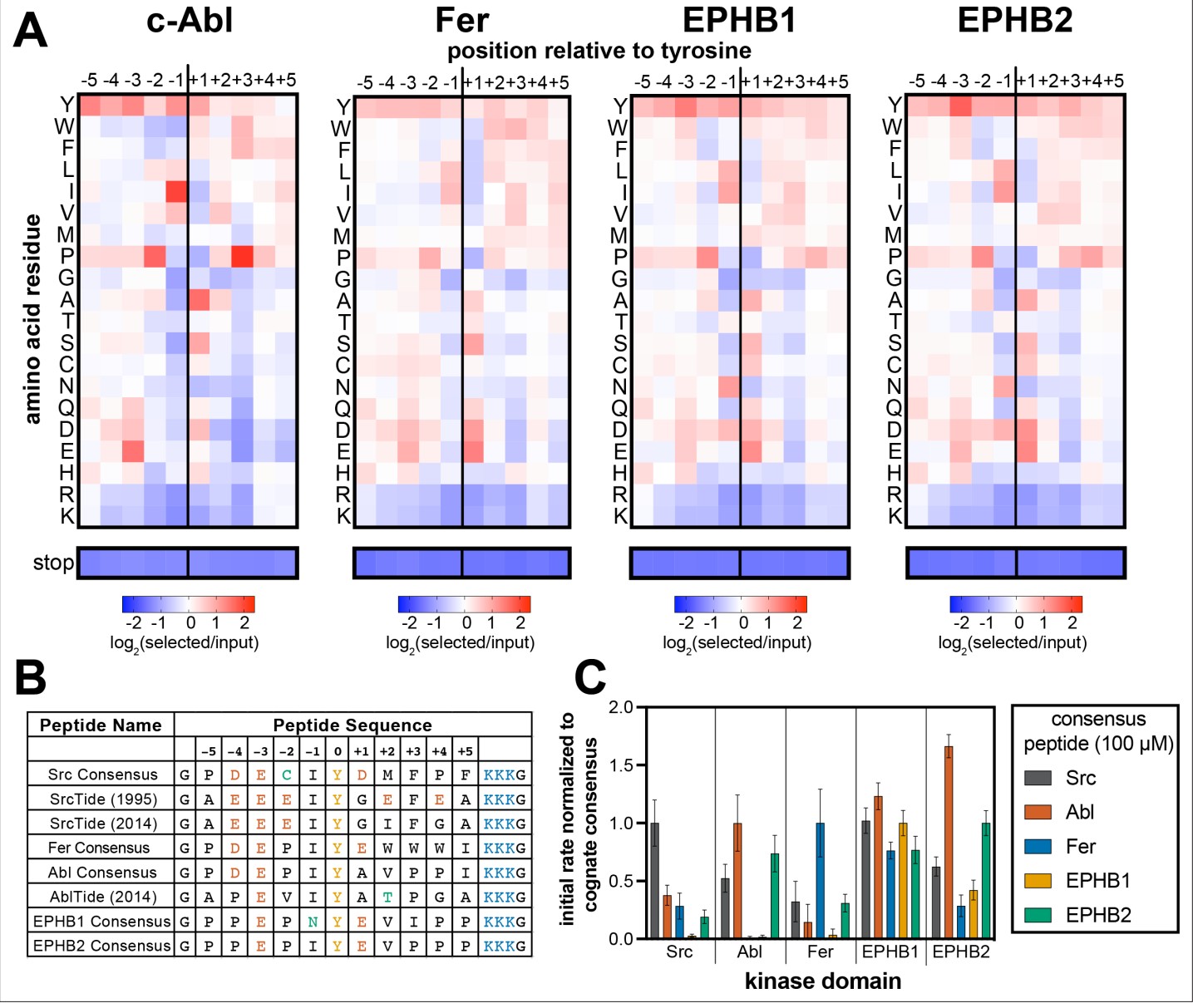

**Figure 2.** Specificity profiling of tyrosine kinases using the $X_5$-Y-$X_5$ library. (**A**) Heatmaps depicting the specificities of c-Abl, Fer, EPHB1, and EPHB2. Enrichment scores were $\log_2$-transformed and are displayed on a color scale from blue (disfavored sequence features, negative value), to white (neutral sequence features, near zero value), to red (favored sequence features, positive value). Values in the heatmaps are the average of three replicates. (**B**) Sequences of consensus peptides identified through $X_5$-Y-$X_5$ screens, compared with previously reported SrcTide and AblTide sequences. (**C**) Phosphorylation kinetics of five consensus peptides against five kinases. Initial rates were normalized to the rate of the cognate consensus peptide. All peptides were used at a concentration of 100 µM, and the kinases were used at a concentration of 10–50 nM. Error bars represent the standard deviation from at least three measurements.

The online version of this article includes the following source data and figure supplement(s) for figure 2:

**Source data 1.** Position-specific amino acid enrichment matrices from the tyrosine kinase $X_5$-Y-$X_5$ library screens.

**Figure supplement 1.** Heatmaps and logos depicting the specificities of c-Abl, Fer, EPHB1, and EPHB2.

**Figure supplement 2.** Phosphorylation kinetics of five consensus peptides against five kinases.

Unlike c-Src, c-Abl preferred an alanine at the +1 position and had a notably strong preference for proline at the +3 position (*Figures 1B and 2A*, *Figure 2—figure supplement 1*). Fer showed a specificity pattern distinct from both c-Src and c-Abl, which included a preference for tryptophan residues at the +2,+3, and +4 positions. As expected, the closely related EPHB1 and EPHB2 kinases had

**Table 1.** Michaelis-Menten parameters for consensus peptides against c-Src and c-Abl kinase domains.

All measurements were carried out using the ADP-Quest assay in three to five replicates. Errors represent the standard error in global fits of all replicates to the Michaelis-Menten equation.

| Entry | Kinase | Peptide name | Peptide sequence | $k_{cat}$ (s$^{-1}$) | $K_M$ (µM) |
|---|---|---|---|---|---|
| 1 | c-Src | Src Consensus | GPDECIYDMFPFKKKG | 4.9±0.4 | 196±38 |
| 2 | c-Src | Src Consensus (P-5C, D+1 G) | GCDECIYGMFPFKKKG | 4.4±0.2 | 97±10 |
| 3 | c-Src | SrcTide (1995) | GAEEEIYGEFEAKKKG | 3.1±0.2 | 64±10 |
| 4 | c-Src | SrcTide (2014) | GAEEEIYGIFGAKKKG | 1.8±0.1 | 7±3 |
| 5 | c-Src | Fer Consensus | GPDEPIYEWWWIKKKG | 0.4±0.1 | 8±4 |
| 6 | c-Src | Abl Consensus | GPDEPIYAVPPIKKKG | 2.0±0.2 | 159±31 |
| 7 | c-Abl | Abl Consensus | GPDEPIYAVPPIKKKG | 3.0±0.2 | 6±2 |
| 8 | c-Abl | AblTide (2014) | GAPEVIYATPGAKKKG | 2.5±0.2 | 35±8 |

similar specificities, which included a unique preference for Asn and Asp at the –1 residue that was not observed for the tested non-receptor tyrosine kinases (*Figure 2A*, *Figure 2—figure supplement 1*).

## Degenerate library screens can be used to design highly-efficient peptide substrates

Specificity profiling methods are often used to design consensus sequences that serve as optimal peptide substrates for biochemical assays and biosensor design (*Deng et al., 2014*; *Lin et al., 2019*; *Songyang et al., 1995*). We wanted to assess whether our method could also be used to generate high-efficiency substrates, and whether these would differ from sequences identified using oriented peptide libraries. To test this, we combined the most favorable amino acids in each position flanking the central tyrosine residue in our specificity profiles, excluding tyrosine, to generate unique consensus peptide substrates for c-Src, c-Abl, Fer, EPHB1, and EPHB2. Consensus sequences for c-Src and c-Abl have been identified previously using oriented peptide libraries (*Deng et al., 2014*; *Songyang et al., 1995*). These sequences, often referred to as SrcTide and AblTide, are different than our consensus sequences at a few residues surrounding the phospho-acceptor tyrosine (*Figure 2B*). The SrcTide and AblTide peptides are canonically embedded within a conserved peptide scaffold containing N-terminal Gly and C-terminal (Lys)$_3$-Gly flanks. For direct comparison, we embedded our consensus peptides in the same scaffold and conducted a series of kinetic studies.

First, we used an *in vitro* continuous fluorimetric assay to compare the steady-state kinetic parameters ($k_{cat}$ and $K_M$) for our c-Src and c-Abl consensus peptides with the SrcTide and AblTide peptides. The Michaelis-Menten parameters for our Src Consensus peptide were on par with one of the previously reported SrcTide substrates (SrcTide 1995, *Songyang et al., 1995*), but the $K_M$ value for a more recently reported SrcTide variant (SrcTide 2014, *Deng et al., 2014*) was substantially tighter (*Table 1*). Our Src Consensus peptide had a higher maximal catalytic rate ($k_{cat}$) but a lower apparent binding affinity ($K_M$) when compared to both SrcTides. We were surprised to see that our Src Consensus peptide had a+1 Asp residue, as opposed to the +1 Gly residue in both SrcTides. Substitution of the +1 Asp for a Gly in a related peptide marginally improved the $K_M$ value but reduced $k_{cat}$ (*Table 1*). These results indicate that our c-Src specificity screens may select for peptides with a high $k_{cat}$, and that there is a trade-off between $k_{cat}$ and $K_M$ for c-Src substrate recognition. For c-Abl, our consensus peptide had both a higher maximal rate ($k_{cat}$) and tighter apparent affinity ($K_M$) relative to the previously reported AblTide peptide (*Table 1*). Collectively, these experiments suggest that different methods may be biased toward slightly different realms of sequence space, and that there are multiple solutions to achieving high-efficiency phosphorylation.

Next, we assayed all of the consensus peptides generated using our approach against their cognate kinases, as well as the other kinases in our screens. For the non-receptor tyrosine kinases (c-Src, c-Abl, and Fer), the corresponding consensus peptides were the best substrates tested. At a higher substrate concentration (100 µM), c-Abl also efficiently phosphorylated the Src and EPHB2 consensus

peptides (*Figure 2C*), but selectivity for the Abl Consensus improved at a lower concentration (20 µM), consistent with selectivity being driven by $K_M$ for this set of peptides (*Figure 2—figure supplement 2* and *Table 1*). By contrast, c-Src was selective for the Src Consensus peptide at high concentrations (*Figure 2C*), but showed significant off-target activity toward the Fer Consensus at low concentrations (*Figure 2—figure supplement 2*). Michaelis-Menten analysis of the Fer Consensus with c-Src revealed that it has a remarkably tight $K_M$ for c-Src, with a low $k_{cat}$ as a trade-off (*Table 1*). Finally, we observed that the receptor tyrosine kinase EPHB1 showed very little selectivity across the consensus peptides and did not prefer its own cognate consensus sequence (*Figure 2C* and *Figure 2—figure supplement 2*). EPHB2, on the other hand, efficiently phosphorylated its own consensus peptide, as well as the Abl Consensus. Both of these sequences contain a −1 Ile and +3 Pro (*Figure 2B and C*). These experiments demonstrate the applicability of our bacterial peptide display method to the design of high-activity substrates. Our results also suggest that not all consensus peptides will be selective for their given kinase, as there can be overlap in substrate specificities.

## Data from $X_5$-Y-$X_5$ library screens can be used to predict the relative phosphorylation rates of peptides

Given that data from the $X_5$-Y-$X_5$ library screens could yield high-efficiency substrates, we investigated whether the same data could be used to quantitatively predict the relative phosphorylation rates of biologically interesting sequences. If so, this would be a potentially powerful tool for the identification of native substrates and the dissection of phosphotyrosine signaling pathways. Indeed, oriented peptide libraries have been applied extensively to predict the native substrates of protein kinases (*Johnson et al., 2023*; *Miller et al., 2008*; *Obenauer et al., 2003*). We are particularly interested in using high-throughput specificity screens to predict how mutations proximal to phosphorylation sites affect tyrosine kinase selectivity. The PhosphoSitePlus database documents thousands of missense mutations within five residues of tyrosine phosphorylation sites, many of which are associated with human diseases or are human polymorphisms, but the functional consequences of most of these mutations are unexplored (*Hornbeck et al., 2019*; *Hornbeck et al., 2015*; *Krassowski et al., 2018*; *Landrum et al., 2018*).

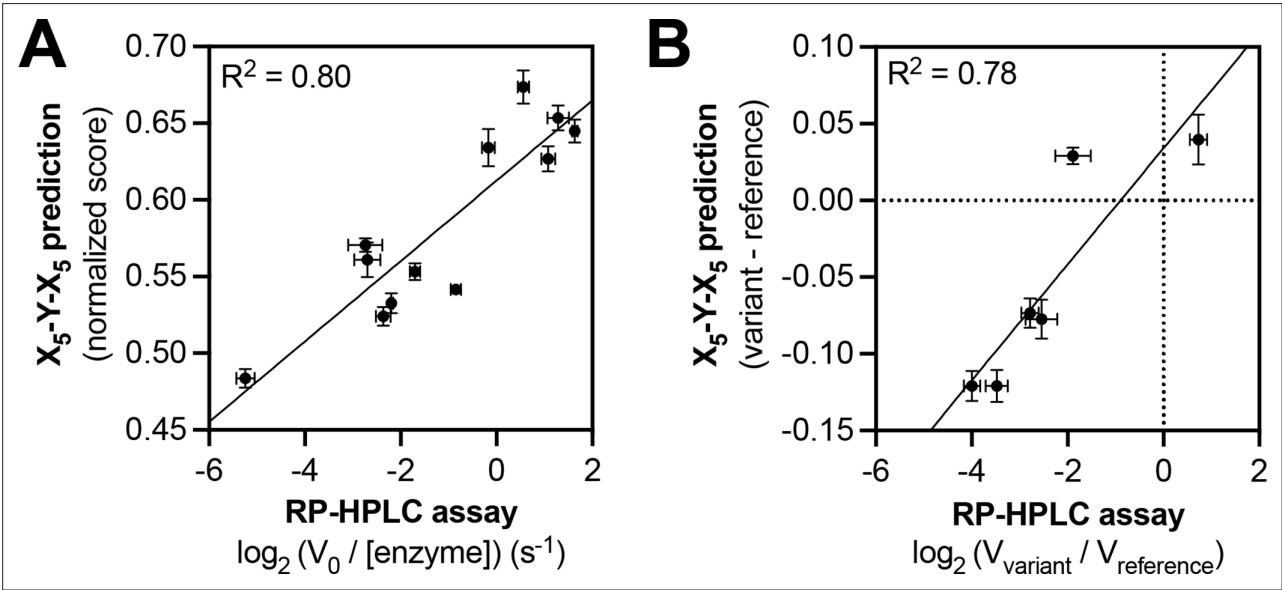

**Figure 3.** Predicting relative phosphorylation rates using data from $X_5$-Y-$X_5$ library screens. (**A**) Correlation between measured phosphorylation rates and $X_5$-Y-$X_5$ predictions for 12 peptides with c-Src. All peptides were used at a concentration of 100 µM, and c-Src was used at a concentration of 500 nM. Error bars represent the standard deviation from at least three rate measurements and three separate scores with individual replicates of the $X_5$-Y-$X_5$ screen. (**B**) Correlation between the magnitude of mutational effects for 6 peptide pairs with mutational effects predicted from $X_5$-Y-$X_5$ library screens. Error bars represent the standard deviation of at least three rate measurements and three separate scores with individual replicates of the $X_5$-Y-$X_5$ screen.

The online version of this article includes the following figure supplement(s) for figure 3:

**Figure supplement 1.** Assay to measure peptide phosphorylation rates using reverse-phase HPLC.

We used the c-Src $X_5$-Y-$X_5$ screening data to predict the relative phosphorylation rates of six peptide pairs, corresponding to reference and variant sequences derived from human phosphorylation sites. Each peptide sequence was scored using an approach that is similar to that used for oriented peptide libraries in the Scansite database (*Obenauer et al., 2003*; *Yaffe et al., 2001*). For each peptide sequence, we summed the $\log_2$-transformed enrichment values for the appropriate amino acid at each position in the peptide (the numerical values that make up the heatmaps in *Figures 1B and 2A*). This sum was divided by the number of variable positions (10 positions for all peptides in this study), then normalized to be on a scale from 0 (the worst possible sequence) to 1 (the best possible sequence). We compared the predicted scores to *in vitro* phosphorylation rates measured using a highly-sensitive assay based on reverse-phase high-performance liquid chromatography (RP-HPLC) (*Figure 3* and *Figure 3—figure supplement 1*). We found that our predictions, which were derived from log-transformed enrichment scores, correlated moderately well with the log-transformed rates of phosphorylation by c-Src (*Figure 3A*). The predictions could differentiate high, medium, and low activity substrates but could not accurately rank peptides within these clusters. Focusing specifically on the effects of the mutations in this set of peptides, we found that the $X_5$-Y-$X_5$ screening data could accurately predict the directionality of the effects of five out of six mutations (*Figure 3B*).

One drawback to the aforementioned scoring approach, like all models based on position-specific scoring matrices, is that it cannot capture context-dependent amino acid preferences. We recently explored a machine-learning approach, using screening data from a related degenerate library, to model c-Src kinase specificity (*Rube et al., 2022*). The model not only incorporated pairwise inter-residue dependencies, but also data from multiple time points. This approach could reasonably predict absolute rate constants, as well as the directionality and magnitude of several phosphosite-proximal mutational effects. As an alternative to building models based on random library screens, we reasoned that direct measurements of reference and variant peptides using our screening platform might also provide reliable assessment of mutational effects.

## A proteome-derived peptide library accurately measures sequence specificity and phosphorylation rates

To refine our assessment of phosphosite-proximal mutational effects, we designed a library, derived from the PhosphoSitePlus database, that is composed of 11-residue sequences spanning 3159 human phosphosites and 4760 disease-associated variants of these phosphosites bearing a single amino acid substitution (pTyr-Var library; *Figure 4—figure supplement 1*; *Hornbeck et al., 2019*). While the majority of sequences in this library contained a single tyrosine residue, some sequences contained multiple tyrosines, for which we included additional variants where the non-central tyrosine residues were mutated to phenylalanine. Including these tyrosine mutants and additional control sequences, such as previously designed consensus substrates, the library totaled ~10,000 unique sequences. As with the $X_5$-Y-$X_5$ library, we generated two versions of this library, bearing a C-terminal strep-tag or myc-tag. We conducted specificity screens with the myc-tagged pTyr-Var library against 7 non-receptor tyrosine kinases (c-Src, Fyn, Hck, c-Abl, Fer, Jak2, and AncSZ, an engineered homolog of Syk and ZAP-70 *Hobbs et al., 2022*) and 5 receptor tyrosine kinases (EPHB1, EPHB2, FGFR1, FGFR3, and MERTK). The majority of these kinases could be expressed in bacteria and purified in good yield (*Albanese et al., 2018*; *Hobbs et al., 2022*). One of these kinases (Jak2) was purchased from a commercial vendor.

Using the catalytically active tyrosine kinase constructs, we identified an optimal concentration (typically between 0.1–1.5 µM) to ensure 20–30% of maximal phosphorylation in three minutes. For some kinases (FGFR1, FGFR3, and MERTK), pre-incubation with ATP was required in order to activate the kinase by auto-phosphorylation (*Figure 4—figure supplement 2*). We conducted the screens analogously to those with the $X_5$-Y-$X_5$ library, but rather than calculate position-specific residue preferences from the deep sequencing data, we directly calculated enrichment scores for each peptide in the pTyr-Var library (*Figure 4A* and *Figure 4—source data 1*). Three to five replicates of the pTyr-Var screen were conducted with each kinase, and the results were reproducible across replicates (*Figure 4—figure supplement 3*). To validate our pTyr-Var screens, we examined enrichment scores from the c-Src experiments for the same six peptide pairs for which predictions using $X_5$-Y-$X_5$ screening data were only moderately accurate. We found a strong correlation between the pTyr-Var enrichment scores and phosphorylation rates, particularly for high-activity sequences (*Figure 4B*). Furthermore,

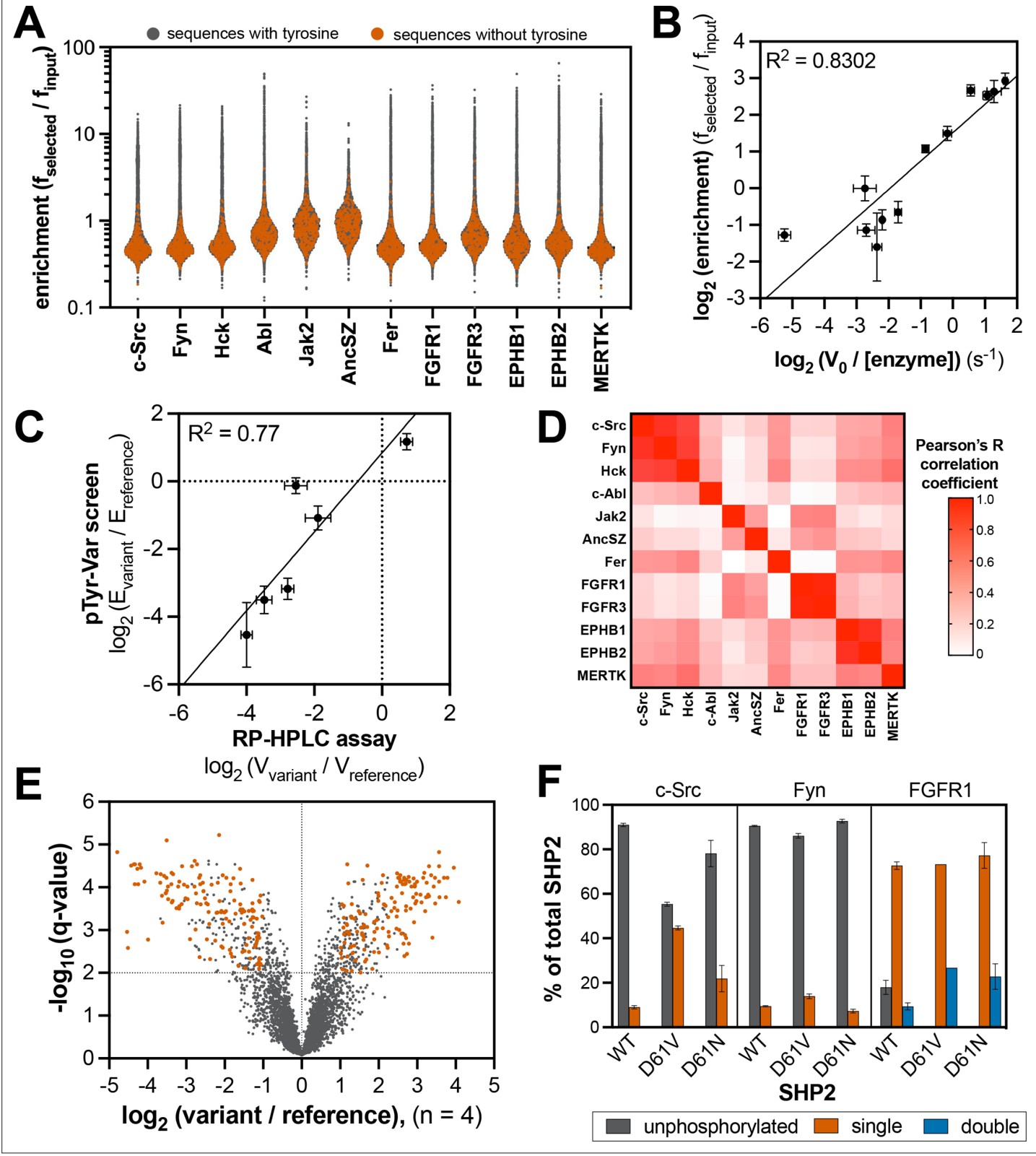

**Figure 4.** Specificity profiling of tyrosine kinases using the pTyr-Var library. (**A**) Distribution of enrichment scores from pTyr-Var screens with 13 tyrosine kinases. Each point represents a peptide sequence in the pTyr-Var library. Data points in orange-red represent sequences without a Tyr residue and data points in dark gray represent sequences with a Tyr residue. Each dataset represents the average of three to five replicates. (**B**) Correlation between enrichment scores and measured phosphorylation rates for 12 peptides (100 μM) with c-Src (500 nM). (**C**) Correlation between the magnitude of

*Figure 4 continued on next page*

*Figure 4 continued*

mutational effects for 6 peptide pairs in the pTyr-Var library with mutational effects measured using an *in vitro* kinetic assay. Error bars in panels **B** and **C** represent the standard deviation from 3 to 4 rate measurements and four pTyr-Var screens. (**D**) Matrix of Pearson's correlation coefficients for all pairwise comparisons between replicate-averaged pTyr-Var datasets for 13 kinases. (**E**) Volcano plot depicting mutational effects in the pTyr-Var screen with c-Src kinase domain. Data points represent the average of four replicates. Hits are colored orange-red. (**F**) Percent phosphorylation of SHP2 wild-type, D61V, and D61N (10 μM) after an hour incubation with c-Src, Fyn, and FGFR1 (1 μM). Error bars represent the standard deviation from 2 to 3 measurements.

The online version of this article includes the following source data and figure supplement(s) for figure 4:

**Source data 1.** Enrichment scores from tyrosine kinase pTyr-Var screens.

**Source data 2.** Position-specific amino acid enrichment matrices from the tyrosine kinase pTyr-Var library screens for sequences containing a single central tyrosine residue.

**Figure supplement 1.** Properties of the pTyr-Var library.

**Figure supplement 2.** Pre-activation of FGFR1, FGFR3, and MERTK by auto-phosphorylation.

**Figure supplement 3.** Matrix of Pearson's correlation coefficients for all replicates of pTyr-Var screens across all 12 kinases.

**Figure supplement 4.** Assessment of the extent of enrichment in pTyr-Var screens with 12 kinases.

**Figure supplement 5.** Heatmaps depicting the position-specific amino acid preferences for 12 tyrosine kinase domains, extracted from screens with the pTyr-Var library.

**Figure supplement 6.** Volcano plots depicting mutational effects in the pTyr-Var screen for 12 kinase domains.

**Figure supplement 7.** Number of significant mutations for each kinase at each position surrounding the central tyrosine residue.

**Figure supplement 8.** Enrichment scores from pTyr-Var screens for phosphorylation of the RET Tyr 981 reference and variant (R982C) peptides by 12 tyrosine kinases.

**Figure supplement 9.** Enrichment scores from pTyr-Var screens for phosphorylation of SHP2 Y62 reference and variant (D61N and D61V) peptides by c-Src, Fyn, and FGFR1.

the effects of mutations in the screens were consistent with those observed using the *in vitro* RP-HPLC assay with purified peptides (*Figure 4C*).

A total of 370 peptides in the pTyr-Var library contain no tyrosine residues and thus serve as controls to determine background noise in our screens. For every kinase tested, the tyrosine-free sequences showed distinctly low enrichment scores, consistent with signal in these screens being driven by tyrosine phosphorylation of the surface-displayed peptides (*Figure 4A*). For each kinase, a subset of the library (between 7% and 10%) showed enrichment scores above this background level (*Figure 4—figure supplement 4*). To confirm that the pTyr-Var screens were reporting on unique substrate specificities across these tyrosine kinases, we calculated Pearson's correlation coefficients for the average datasets of each kinase pair and visualized position-specific amino acid preferences as heatmaps (*Figure 4D*, *Figure 4—figure supplement 5*, and *Figure 4—source data 2*). We found strong correlation in specificity between kinases of the same family (the Src-family kinases c-Src/Fyn/Hck, and receptor pairs EPHB1/EPHB2 and FGFR1/FGFR3). We also observed that the specificity of Src-family kinases partly overlapped with the Ephrin receptors and MERTK. The specificity of AncSZ and Jak2 correlated with that of FGFRs.

Next, we compared the results of our pTyr-Var library screens with a curated list of kinase-substrate pairs found in the PhositePlus database (*Hornbeck et al., 2019*). For c-Src, Fyn, and c-Abl, out of the sequences that overlapped between our library and the curated list, 30–40% of the kinase-substrate pairs showed efficient phosphorylation in the peptide-display screen (*Figure 4—source data 1*). This is consistent with a previous study using bacterial display and a different proteome-derived peptide library (*Shah et al., 2018*). The modest overlap between peptide screens and literature-reported kinase-substrate pairs is not surprising, given that other mechanisms in kinase-substrate recognition, such as localization, may override kinase domain sequence preferences (*Miller and Turk, 2018*). Furthermore, the curated list of kinase-substrate pairs comes from both *in vitro* and *in vivo* studies and may not accurately represent *bona-fide* substrates for each kinase.

## Natural variants of tyrosine phosphorylation sites impact kinase recognition

For pairs of peptides in the pTyr-Var library that correspond to a disease-associated variant and a reference sequence, we calculated the $\log_2$-fold change in enrichment for the variant relative to the reference. The large number of replicates for each screen afforded a robust analysis of phosphosite-proximal mutational effects for each kinase. We filtered the results in five steps to identify significant mutations: (1) We omitted phosphosite pairs where there was no statistically significant difference in enrichment between the variant and reference (p-value cutoff of 0.05). (2) We then applied a second filtering step to remove phosphosite pairs where the fold-change in enrichment between the variant and reference sequence was less than two. (3) Next, we excluded pairs where both sequences were low-activity substrates (enrichment score less than 1.5). (4) We removed mutations that added or removed a tyrosine residue, as their interpretation is ambiguous in our assay. (5) Lastly, we excluded phosphosite pairs in which the average read count of either the variant or wild-type sequence was less than 50. This left us with unique set of 50–400 high-confidence candidates for each tyrosine kinase (*Figure 4E*, *Figure 4—figure supplement 6*, and *Figure 4—source data 1*). From this filtered list, we found that kinases showed distinct patterns of mutational sensitivity at each position around the central tyrosine, consistent with their distinct sequence preferences (*Figure 4—figure supplement 7*).

For c-Src, we identified 381 high-confidence mutations (*Figure 4E*). A number of these mutations were on proteins involved in neurotrophin-regulated signaling, cyclin-dependent serine/threonine kinase activity, and other receptor/non-receptor tyrosine kinase activity. We found notable mutational effects at a known target of c-Src, Tyr 149 of the tumor suppressor protein FHL1 (*Wang et al., 2018*), as well as on other proteins known to interact with c-Src, such as the lipid and protein phosphatase PTEN and the immune receptor LILRB4 (*Kang et al., 2016*; *Lu et al., 2003*). We were particularly interested in cases where a kinase not previously known to phosphorylate a specific phosphosite showed a dramatic gain-of-function upon phosphosite-proximal mutation. For example, we found that the R982C mutation, proximal to Tyr 981 on the receptor tyrosine kinase RET, significantly enhanced phosphorylation by c-Src (*Figure 4—figure supplement 8*). This phosphosite is a known to engage the SH2 domain of c-Src and facilitate c-Src activation upon recruitment to RET, but it is not considered a kinase substrate of c-Src (*Encinas et al., 2004*). This mutation could potentially rewire signaling by promoting phosphorylation of RET by c-Src, and in doing so, sustaining c-Src activation by its binding to phospho-RET. The RET R982C mutation also enhanced Tyr 981 phosphorylation by several other kinases, most notably Fer (*Figure 4—figure supplement 8*). These examples show how the pTyr-Var data could be used as a resource to guide mutation-focused signaling studies.

To further validate our approach, we examined the effects of phosphosite-proximal mutations on the phosphorylation of an intact protein, rather than a peptide. Tyr 62 in the tyrosine phosphatase SHP2 sits within a region of this protein that is frequently mutated in various human diseases (*Tartaglia et al., 2006*), and this residue is highly phosphorylated in receptor tyrosine kinase-driven cancers (*Gillette et al., 2020*; *Pfeiffer et al., 2022*). Several Tyr 62-proximal mutations are encoded in the pTyr-Var library. In our screens, the reference peptide for Tyr 62 was preferentially phosphorylated by receptor tyrosine kinases, such as FGFR1, over non-receptors such as c-Src and Fyn, and nearby mutations showed varied effects on Tyr 62 phosphorylation, depending on the kinase tested (*Figure 4—figure supplement 9*). For example, D61V enhanced and D61N attenuated phosphorylation by Src-family kinases, but these mutations had little impact on recognition by FGFR1. To assess whether the effects of D61 mutations in the screens were retained in the context of the intact protein, we monitored phosphorylation of wild-type, D61V, and D61N SHP2 by c-Src, Fyn, and FGFR1 using intact protein mass spectrometry. We made two modifications to SHP2 to facilitate measurements: (1) substitution of the catalytic residue (C459E) to prevent dephosphorylation by the SHP2 phosphatase domain and (2) deletion of the disordered C-terminal tail to avoid background phosphorylation of an accessible site. Our measurements recapitulated the relative phosphorylation efficiencies for the Tyr 62 reference peptides, with Fyn being the slowest, and FGFR1 being the fastest (*Figure 4F* and *Figure 4—figure supplement 9*). Both D61V and D61N dramatically enhanced phosphorylation by all three kinases, consistent with reports that mutations at this site dramatically alter SHP2 structure and probably also increase Tyr 62 accessibility (*Keilhack et al., 2005*). For c-Src and Fyn, but not FGFR1, D61V showed a stronger enhancement of phosphorylation than D61N, consistent with our peptide screens (*Figure 4F* and *Figure 4—figure*

*supplement 9*). The effects of these mutations in SHP2 on signal rewiring in cells warrants further investigation.

## Position-specific amino acid preferences for tyrosine kinases are context-dependent

As noted earlier, position-specific scoring matrices do not reflect context-dependent sequence preferences. To illustrate this further, we scored peptide sequences in the pTyr-Var library using the position-specific scoring matrices generated from the $X_5$-Y-$X_5$ library. For peptides that showed significant enrichment in the pTyr-Var screens (enrichment >1), there was a modest correlation with the scores predicted using the $X_5$-Y-$X_5$ library, with many outliers (*Figure 5A* and *Figure 5—figure supplement 1*). We selected peptides for c-Src and c-Abl that were high-activity sequences based on the pTyr-Var screens (enrichment >4) but deviated significantly from canonical recognition motifs, and therefore were low scoring (score <0.5). The peptides selected for c-Src had unfavorable residues downstream of the central tyrosine (+1 Arg and +3 Gly for MISP_Y95;+1 Asn,+2 Arg, and +3 Glu for HLA-DPB1_Y59_F64L_YF). For c-Abl, the peptides had an unfavorable –1 Glu and +2 Ser (SIRPA_Y496_P491L) or an unfavorable +2 Glu and +3 Gly (HGD_Y166_F169L). We measured phosphorylation rates for these peptides using our RP-HPLC assay. Phosphorylation rates for these peptides deviated from what would be expected based on a position-specific scoring matrix (*Figure 5B*, *Figure 5—figure supplement 1*, and *Figure 5—source data 1*). This suggests that the putatively unfavorable sequence features in these peptides were tolerated in their specific sequence contexts.

The observation that there are context-dependent sequence preferences for kinase-substrate interactions has important consequences for predicting the effects of phosphosite-proximal mutations. The same substitution could have different effects depending on the composition of the surrounding sequence. This phenomenon is uniquely visible in our screening approach, as we are measuring the phosphorylation of defined peptide sequences, and we are conducting screens with thousands of peptide pairs that vary by only a single amino acid substitution. To test our hypothesis, we assessed whether the directionality of mutational effects observed for specific peptides in the pTyr-Var screen could be predicted using the position-specific scoring matrix derived from the $X_5$-Y-$X_5$ screen (which would represent the effect of making a substitution averaged over all sequence contexts). While the directionality of the effect of most mutations could be predicted by the $X_5$-Y-$X_5$ screen, we observed many mutations that showed a significant effect where none was predicted, as well as mutations where the effect was the opposite of what was predicted (*Figure 5C*, *Figure 5—figure supplement 2*, and *Figure 5—source data 2*).

To validate this observation, we selected a peptide pair in the pTyr-Var library where a mutation (–2 Ser to Pro) had the opposite effect of that predicted by our $X_5$-Y-$X_5$ screen for c-Src (*Figure 5D*), as well as published results with oriented peptide libraries (*Begley et al., 2015*; *Obenauer et al., 2003*). Additionally, we made the same substitutions to the c-Src consensus peptide to determine whether the $X_5$-Y-$X_5$ predictions would hold true in that context. Measurements of these purified peptides by c-Src show that the same amino acid substitution had different impacts on c-Src recognition, depending on the sequence context (*Figure 5D*). A previous study that analyzed the specificity of the epidermal growth factor receptor (EGFR) kinase using bacterial peptide display showed that the effect of mutations at the –2 position was sometimes dependent on the identity of the –1 residue (*Cantor et al., 2018*). Molecular dynamics analyses in that report suggested that the amino acid identity at the –1 position determined how the side chain of the –2 residue was presented to the kinase, and vice versa, thereby dictating context-dependent preferences at both positions. Our pTyr-Var screens suggest that context dependent sequence preferences may be commonplace. Depending on the kinase, 5–15% of all significant mutations in the pTyr-Var screen had the opposite effect of that predicted using the $X_5$-Y-$X_5$ library data. Mapping these context-dependent effects comprehensively could have a significant impact on our ability to predict native substrates of kinases, and it will improve our understanding of the structural basis for substrate specificity.

## Phosphorylation of bacterial peptide display libraries enables profiling of SH2 domains

In previous implementations of our bacterial peptide display and deep sequencing approach, the specificities of phosphotyrosine recognition domains (e.g. SH2 domains and phosphotyrosine binding

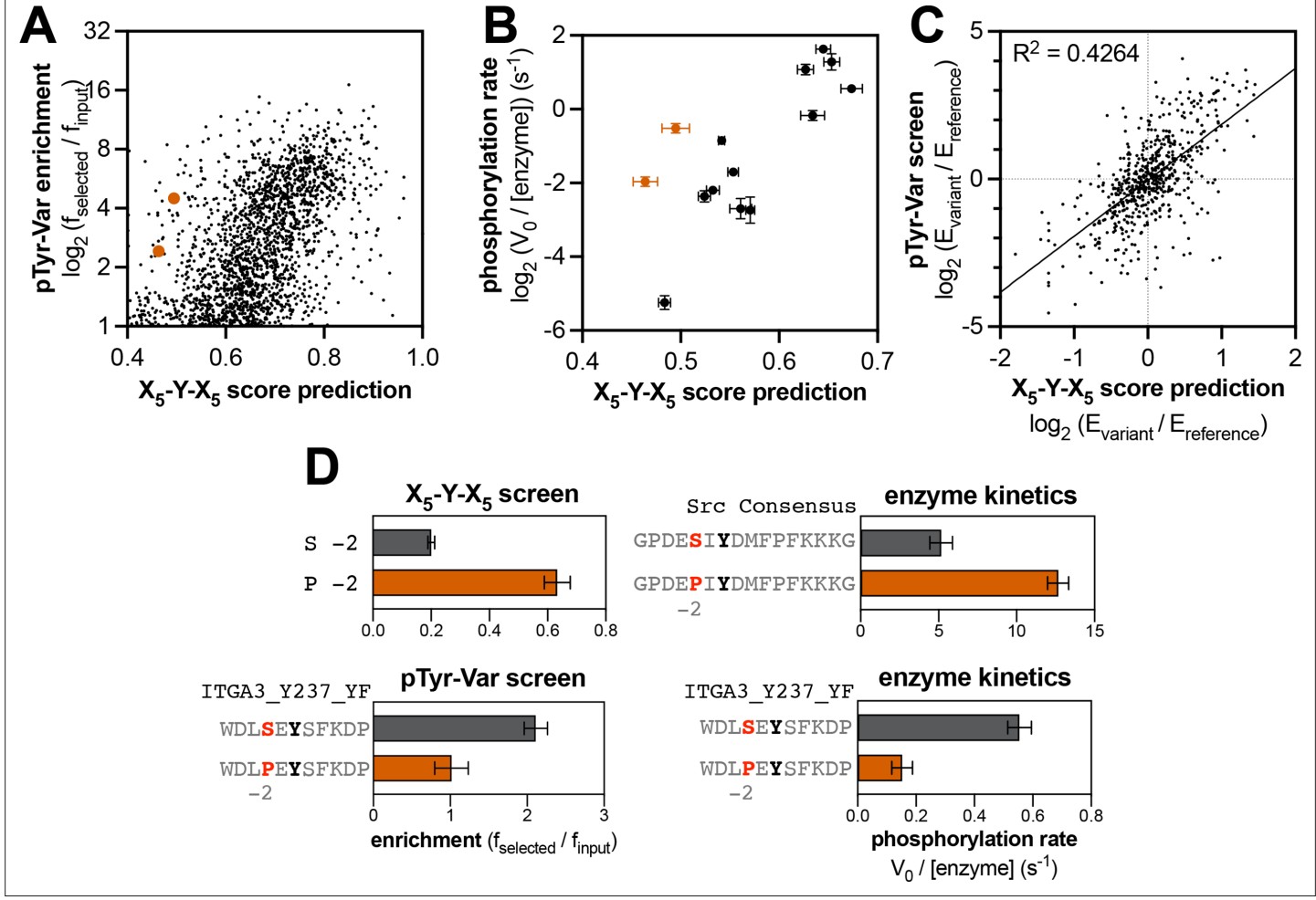

**Figure 5.** Context-dependent effects of tyrosine kinase recognition. (**A**) Correlation of enrichment scores measured for c-Src in the pTyr-Var library screen with scores predicted from the $X_5$-Y-$X_5$ library using a position-specific scoring matrix. (**B**) Correlation between predicted scores and measured phosphorylation rates for 14 peptides (100 μM) with c-Src (500 nM). Peptides that could not be accurately scored by the $X_5$-Y-$X_5$ data are highlighted in orange. (**C**) Correlation of variant effects measured in the pTyr-Var library screen with those predicted from the $X_5$-Y-$X_5$ library screen for c-Src. Several points lie in the top-left and bottom-right quadrants, indicating a discrepancy between the measured mutational effect in the pTyr-Var screen and the predicted mutational effect from the $X_5$-Y-$X_5$ screen. (**D**) Effects of serine-to-proline substitution at the −2 position in various assays with c-Src. The left panels show the enrichment levels of −2 serine and proline in the $X_5$-Y-$X_5$ screen (top), and the effect of a −2 serine to proline substitution in a specific peptide in the pTyr-Var screen, (bottom). The right panels show rate measurements using the RP-HPLC assay for the same substitution in the Src consensus peptide (top) and the peptide from the pTyr-Var screen (bottom).

The online version of this article includes the following source data and figure supplement(s) for figure 5:

**Source data 1.** Peptide sequences and their phosphorylation rates by c-Src or c-Abl, measured using the RP-HPLC kinetic assay.

**Source data 2.** Mutational effects measured from the pTyr-Var library screens and their corresponding predictions based on the $X_5$-Y-$X_5$ library screening data.

**Figure supplement 1.** Context-dependent effects of c-Abl substrate recognition.

**Figure supplement 2.** Correlation of variant effects measured in the pTyr-Var library screen with those predicted from the $X_5$-Y-$X_5$ library screen for c-Abl, Fer, EPHB1, and EPHB2.

(PTB) domains) were analyzed in addition to tyrosine kinase domains (***Cantor et al., 2018***; ***Lo et al., 2019***). This approach required two amendments to the kinase screening protocol. First, the surface-displayed libraries were phosphorylated to saturating levels using a cocktail of tyrosine kinases. Second, because phosphotyrosine recognition domains generally have fast dissociation rates from their ligands (***Morimatsu et al., 2007***; ***Oh et al., 2012***), making binding-based selection assays challenging, constructs were generated in which two identical copies of an SH2 domain were artificially

fused together. The tandem-SH2 constructs enhanced avidity for phosphopeptides displayed on the cell surface through multivalent effects, thereby enabling enrichment of cells via FACS (*Cantor et al., 2018*).

For this study, we reasoned that a multivalent SH2 construct could be mimicked by functionalizing avidin-coated magnetic beads with biotinylated SH2 domains. These SH2-coated beads could then be used to select *E. coli* cells displaying enzymatically phosphorylated peptide display libraries, followed by deep sequencing to determine SH2 sequence preferences (*Figure 6A*). Thus, we first established a protocol to produce site-specifically biotinylated SH2 domains in *E. coli*, by co-expressing an Avi-tagged SH2 construct with the biotin ligase BirA (*Gräslund et al., 2017*). This system yielded quantitatively biotinylated SH2 domains, as confirmed by mass spectrometry (*Figure 6—figure supplement 1*). Since the biotinylated SH2 domains could be produced in high yields through bacterial expression, the recognition domains were immobilized on the magnetic beads at saturating concentrations to ensure a uniform concentration across experiments. This also prevented background binding of strep-tagged libraries to the beads, making this method compatible with previously reported strep-tagged libraries (*Cantor et al., 2018*; *Shah et al., 2018*; *Shah et al., 2016*).

To implement SH2 specificity screens, the strep-tagged $X_5$-Y-$X_5$ library was phosphorylated to a high level using a mixture of c-Src, c-Abl, AncSZ, and EPHB1 (*Figure 6—figure supplement 2*). The phosphorylated library was screened against three SH2 domains that fall into distinct specificity classes and are derived from three different types of signaling proteins: the SH2 domain from the tyrosine kinase c-Src, the C-terminal SH2 (C-SH2) domain from the tyrosine phosphatase SHP2, and the SH2 domain from the non-catalytic adaptor protein Grb2 (*Figure 6B*, *Figure 6—figure supplement 3*, and *Figure 6—source data 1*). The $X_5$-Y-$X_5$ library screens recapitulated known sequences preferences for each SH2 domain. For c-Src, there was a distinctive preference for –2 His,+1 Asp/Glu, and +3 Ile, as previously reported from oriented peptide libraries (*Huang et al., 2008*). For Grb2, a characteristic +2 Asn preference dominated the specificity profile (*Gram et al., 1997*; *Huang et al., 2008*; *Kessels et al., 2002*; *Rahuel et al., 1996*; *Songyang et al., 1994*). Notably, our Grb2 screen also reveals subtle amino acid preferences at other positions, which could tune the affinity for +2 Asn-containing sequences. Several studies have measured the sequence specificity of the SHP2 C-SH2 domain using diverse methods, including peptide microarrays, oriented peptide libraries, and one-bead-on-peptide libraries (*Huang et al., 2008*; *Miller et al., 2008*; *Sweeney et al., 2005*; *Tinti et al., 2013*). The results of these reported screens are not concordant. Our method indicates a preference for β-branched amino acids (Thr/Val/Ile) at the –2 position, a small residue (Ala/Ser/Thr) at the +1 position, and strong preference for an aliphatic residue (Ile/Val/Leu) at the +3 position. Our results are most in-line with the one-bead-one-peptide screens (*Sweeney et al., 2005*).

We next phosphorylated and screened the pTyr-Var library against the same three SH2 domains in triplicate (*Figure 6C*, *Figure 6—source data 2*, and *Figure 6—source data 3*). The replicates for each SH2 domain were highly correlated, but datasets between SH2 domains had poor correlation, suggesting distinct ligand specificities (*Figure 6—figure supplement 4*). As observed for kinases, we saw negligible enrichment of peptides lacking a tyrosine residue, but each SH2 domain showed strong enrichment of a few hundred peptides containing one or more tyrosines (*Figure 6C*). With the phosphorylated pTyr-Var library, we also carried out selection with a biotinylated pan-phosphotyrosine antibody to assess the level of bias in phosphorylation across the library. Compared to selection with SH2 domains, selection with the antibody yielded a narrower distribution of enrichment scores, with very few highly enriched sequences, suggesting relatively uniform phosphorylation (*Figure 6C*). We further validated the SH2 screening method by measuring the binding affinities of 9 peptides from the pTyr-Var library with the c-Src SH2 domain, using a fluorescence polarization binding assay. Enrichment scores from the pTyr-Var screen showed a good linear correlation with measured $K_d$ values over two orders of magnitude (*Figure 6D*).

The pTyr-Var library screens with the SH2 domains were analyzed and filtered similarly to those with kinase domains. For each SH2 domain, we identified 50–300 phosphosite-proximal mutations that significantly and reproducibly enhanced or attenuated binding (*Figure 6—figure supplement 5* and *Figure 6—source data 2*). As expected, given their distinct specificities, the c-Src, SHP2-C, and Grb2 SH2 domains showed unique sensitivities to mutations (*Figure 6—figure supplement 6*). We identified several phosphosite-proximal mutations that were selectively gain-of-function for one or two SH2 domains (*Figure 6E* and *Figure 6—source data 2*). These mutations could drive the rewiring

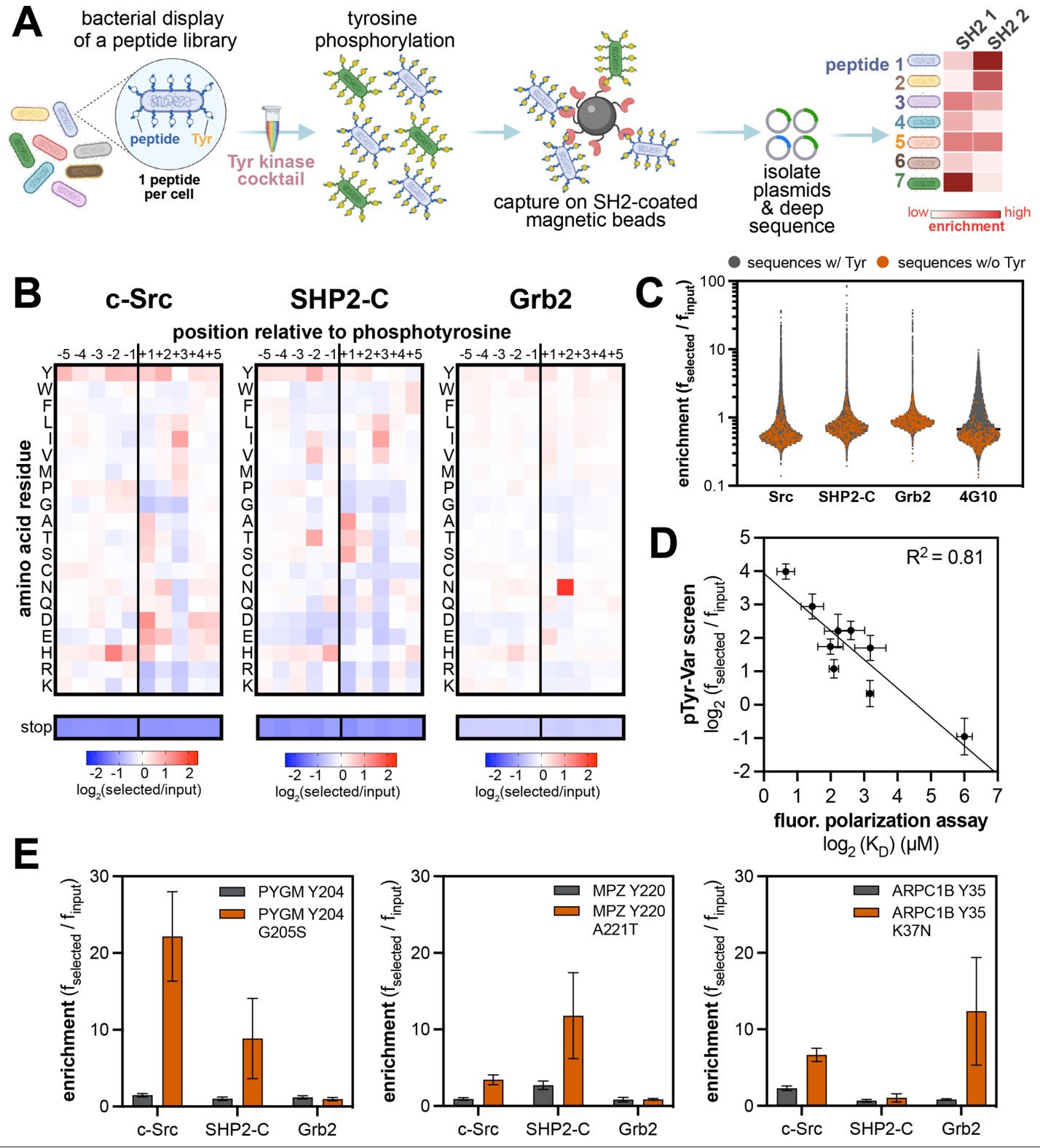

**Figure 6.** High-throughput profiling of SH2 domain ligand specificity using bacterial peptide display. (**A**) Schematic representation of the workflow for SH2 domain specificity profiling. (**B**) Heatmaps depicting the specificities of the c-Src, SHP2-C, and Grb2 SH2 domains, measured using the $X_5$-Y-$X_5$ library. Enrichment scores were $\log_2$-transformed and are displayed on a color scale from blue (disfavored), to white (neutral), to red (favored). Values in the heatmaps are the average of three replicates. (**C**) Distribution of enrichment scores from pTyr-Var screens with three SH2 domains and the pan-phosphotyrosine antibody 4G10 Platinum. Each point represents a peptide sequence in the library. The antibody selection was done similar to the

*Figure 6 continued on next page*

*Figure 6 continued*

kinase screens, with antibody labeling of cells, followed by bead-based enrichment, as opposed to cell enrichment with antibody-saturated beads. Each dataset represents the average of three replicates. (**D**) Correlation between enrichment scores for 9 peptides from the pTyr-Var screen and binding affinities measured using a fluorescence polarization assay. Error bars represent the standard deviations from three screens or binding measurements. (**E**) Examples of phosphosite-proximal mutations that selectively enhance binding to specific SH2 domains. Error bars represent the standard deviations from three screens.

The online version of this article includes the following source data and figure supplement(s) for figure 6:

**Source data 1.** Position-specific amino acid enrichment matrices from the SH2 domain $X_5$-Y-$X_5$ library screens.

**Source data 2.** Enrichment scores from SH2 domain pTyr-Var screens.

**Source data 3.** Position-specific amino acid enrichment matrices from the SH2 domain pTyr-Var library screens for sequences containing a single central tyrosine residue.

**Figure supplement 1.** Mass spectrometry analysis of biotinylated SH2 domains.

**Figure supplement 2.** Flow cytometry analysis of library phosphorylation by a cocktail of tyrosine kinases.

**Figure supplement 3.** Heatmaps and logos depicting the specificities of the c-Src, SHP2-C, and Grb2 SH2 domains, measured using the $X_5$-Y-$X_5$ library.

**Figure supplement 4.** Matrix of Pearson's correlation coefficients for all replicates of pTyr-Var screens across all 3 SH2 domains and 4G10 platinum.

**Figure supplement 5.** Volcano plots depicting mutational effects in the pTyr-Var screen for 3 SH2 domains.

**Figure supplement 6.** Number of significant mutations for each SH2 domain at each position surrounding the central phosphotyrosine residue.

**Figure supplement 7.** Comparison of the pTyr-Var screens for the c-Src kinase and SH2 domains.

**Figure supplement 8.** Divergent effects of phosphosite-proximal mutations on c-Src kinase and SH2 domain recognition.

of signaling pathways by changing which downstream effector engages a phosphosite. This phenomenon was recently reported for lung-cancer associated mutations near phosphorylation sites in EGFR, which impacted the recruitment of Grb2 and SHP2 to the receptor and altered downstream signaling (*Lundby et al., 2019*).

Finally, we note that our pTyr-Var datasets included screens with both the kinase and SH2 domains of c-Src. When the SH2 domain of c-Src interacts with phosphoproteins, it both localizes the kinase domain in proximity to its substrates and activates the enzyme (*Liu et al., 1993*). Our screens revealed that the phosphorylation profiles of c-Src kinase and SH2 domains against the pTyr-Var were completely orthogonal (*Figure 6—figure supplement 7*). Their starkly different activities toward the pTyr-Var library can largely be attributed to kinase domain preference for a+3 Phe and SH2 domain preference for a+3 Ile/Val/Leu/Met. This is in contrast to previous observations for c-Abl, which has kinase and SH2 domains with largely overlapping sequence specificities, dominated by a+3 Pro preference (*Songyang et al., 1995*). For c-Src, phosphosite mutations that impacted recognition by one domain generally had no effect on the other, because preferred sequence features for one domain were typically tolerated (neutral) for the other (*Figure 6—figure supplement 8*). A consequence of this is that phosphosite-proximal mutations may alter c-Src function in two mechanistically distinct ways: (1) mutations that enhance SH2 binding can alter the localization and local activation of c-Src or (2) mutations that enhance kinase recognition will directly increase phosphorylation rates by c-Src. These insights highlight value in profiling multiple domains of the same signaling protein against the same peptide library.

## Amber codon suppression yields an expanded repertoire of peptides for specificity profiling

The specificity profiling screens described thus far were constrained to sequences that contain the canonical twenty amino acids. Several studies have suggested that non-canonical amino acids and post-translationally modified amino acids can also impact sequence recognition by kinases and SH2 domains (*Alfaro-Lopez et al., 1998*; *Begley et al., 2015*; *Chapelat et al., 2012*; *Johnson et al., 2023*; *Yeh et al., 2001*). The most notable example of this is phospho-priming, whereby phosphorylation of one residue on a protein enhances the ability of a kinase to recognize and phosphorylate a proximal residue. This phenomenon was recently described for EGFR, which preferentially phosphorylates sequences containing a tyrosine followed by a+1 phosphotyrosine (*Begley et al., 2015*). Other prevalent post-translational modifications, such as lysine acetylation, may also impact the ability

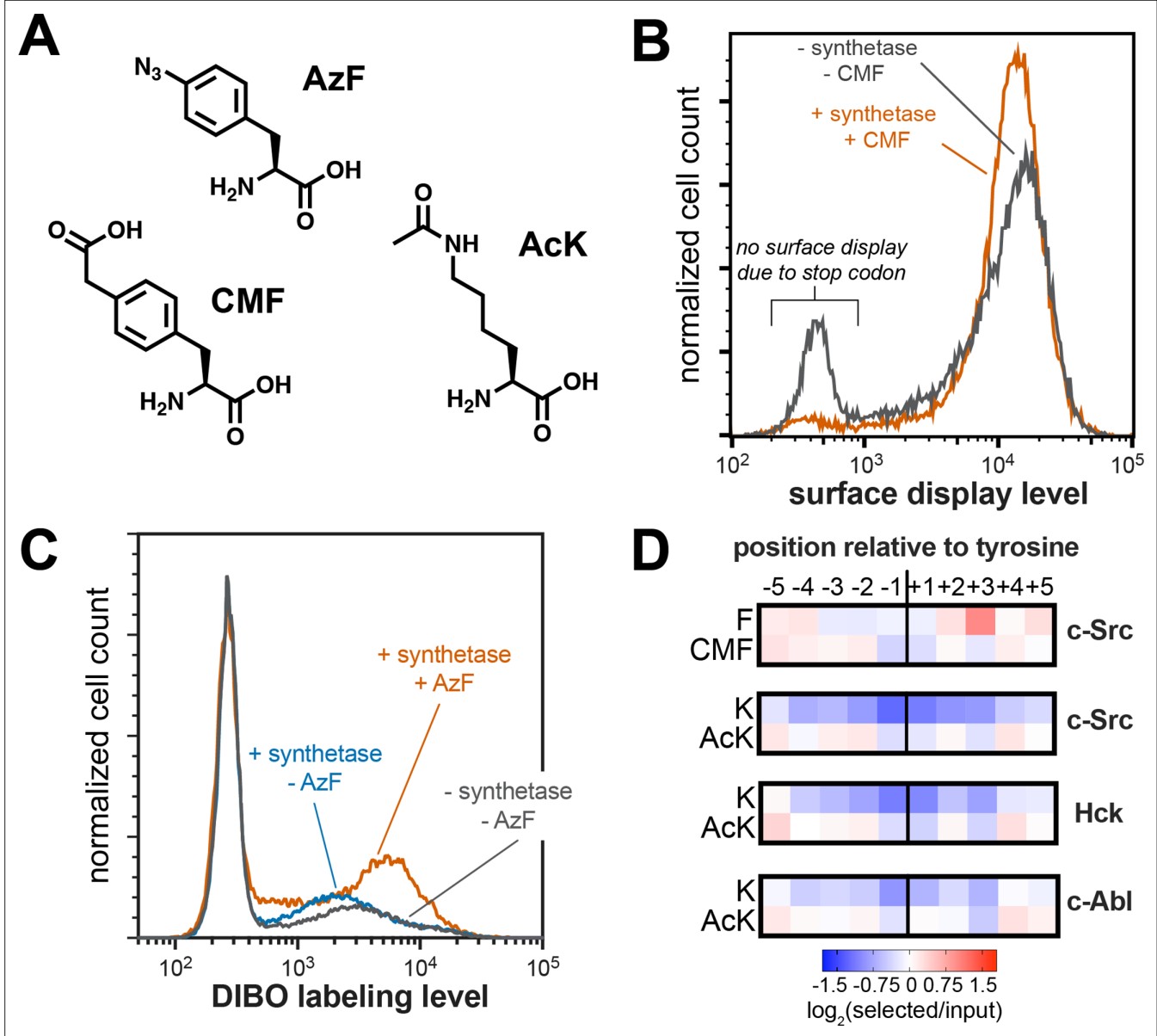

**Figure 7.** Expansion of peptide display libraries using Amber suppression. (**A**) Non-canonical amino acids used in this study. CMF = 4-carboxymethyl phenylalanine, AzF = 4-azido phenylalanine, and AcK = N-ε-acetyl-L-lysine. (**B**) Amber suppression in the strep-tagged $X_5$-Y-$X_5$ library using CMF. Library surface-display level was monitored by flow cytometry using a fluorophore-labeled StrepMAB antibody for samples with or without Amber suppression components. (**C**) AzF labeling on bacterial cells using a DIBO-conjugated fluorophore. Cells expressing the $X_5$-Y-$X_5$ library, with and without various Amber suppression components, were treated with DIBO-conjugated Alexa Fluor 555 then analyzed by flow cytometry. (**D**) Heatmaps depicting the specificities of c-Src, Hck, and c-Abl after CMF or acetyl lysine incorporation. Only sequences with one stop codon were used in this analysis. Enrichment scores were $\log_2$-transformed and are displayed on a color scale from blue (disfavored), to white (neutral), to red (favored). Values in heatmaps are the average of three replicates.

The online version of this article includes the following figure supplement(s) for figure 7:

**Figure supplement 1.** Stop codon enrichment levels in c-Src $X_5$-Y-$X_5$ screens using different analysis methods.

**Figure supplement 2.** Comparison of position-specific enrichments in screens with Amber suppression analyzed in two different ways.

**Figure supplement 3.** Phosphorylation kinetics of Lys- and AcK-containing consensus peptides against c-Src and c-Abl.

of kinases or SH2 domains to recognize a particular phosphosite (*Parker et al., 2014*; *Rust and Thompson, 2011*).

We sought to expand our specificity profiling method to incorporate non-canonical and post-translationally modified amino acids (*Figure 7A*). Since our libraries are genetically encoded, we employed Amber codon suppression and repurposing, using engineered tRNA molecules and amino-acyl tRNA synthetases (*Amiram et al., 2015*; *Xie et al., 2007*; *Zheng et al., 2018*). The degenerate (X) positions in our $X_5$-Y-$X_5$ library are encoded using an NNS codon, which means that an Amber codon (TAG) is sampled at each position 3% of the time. Thus, this library theoretically contains a sufficiently large number of diverse sequences to profile specificity with a 21 amino acid alphabet. For Amber suppression in *E. coli*, tRNA/synthetase pairs are commonly expressed from pEVOL or pULTRA plasmids (chloramphenicol and streptomycin resistant, respectively) (*Chatterjee et al., 2013*). Both of these systems are incompatible with our surface-display platform, which uses MC1061 cells (strep-tomycin resistance encoded in the genome) and libraries in a pBAD33 vector (chloramphenicol resis-tant). Thus, we designed a variant of the pULTRA plasmid in which we swapped the streptomycin resistance gene for an ampicillin resistance gene from a common pET vector for protein expression (pULTRA-Amp).

To confirm that non-canonical amino acids could be incorporated into the $X_5$-Y-$X_5$ library, we co-transformed *E. coli* with the library and a pULTRA-Amp plasmid encoding a tRNA/synthetase pair that can incorporate 4-carboxymethyl phenylalanine (CMF) via Amber suppression (*Figure 7A*; *Xie et al., 2007*). We measured peptide display levels by flow cytometry for cultures that were grown with or without CMF in the media. For the cultures grown without CMF, roughly 20% of the cells had no surface-displayed peptides, consistent with termination of translation at Amber codons within the peptide-coding region (*Figure 7B*). In the presence of CMF, this premature termination was signifi-cantly suppressed, and a larger fraction of the cells displayed peptides. As an additional test, we incorporated 4-azido phenylalanine (AzF) into the $X_5$-Y-$X_5$ library (*Figure 7A*; *Amiram et al., 2015*). Cells expressing this expanded library were treated with a dibenzocyclooctyne (DIBO)-functionalized fluorophore, which should selectively react with the azide on AzF via strain-promoted azide-alkyne cycloaddition (*Ning et al., 2008*). Only cells expressing the synthetase and grown in the presence of AzF showed significant DIBO labeling, confirming Amber suppression and non-canonical amino acid incorporation into our library (*Figure 7C*).

Using this library expansion strategy, we assessed how substrate recognition by c-Src is impacted by neighboring CMF or acetyl-lysine residues. We subjected CMF- or AcK-containing $X_5$-Y-$X_5$ libraries to c-Src phosphorylation, selection, and sequencing, using the same methods described above. When analyzing $X_5$-Y-$X_5$ libraries in standard kinase and SH2 screens, we typically omit all Amber-containing sequences from our calculations, as they do not encode expressed peptides (*Figure 1B* and *Figure 2A*). For these experiments, we included Amber-containing sequences in our analysis. Using this strategy, we found that the Amber codon was less depleted at each position surrounding the central tyrosine than we observed for libraries without Amber suppression, but the log-transformed enrichment scores for Amber codons at all positions surrounding the tyrosine residue were still negative (*Figure 7—figure supplement 1*). We reasoned that, if Amber suppression efficiency was not 100%, any Amber-containing sequence would still be depleted relative to a sequencing lacking a stop codon, due to some premature termination. Thus, we re-analyzed the data by exclusively counting sequences that contained one Amber codon, under the assumption that every sequence would have approximately the same amount of premature termination. This revealed positive enrichment for CMF and AcK at select positions (*Figure 7D* and *Figure 7—figure supplement 1*). Although we only included a fraction of the total library in our new analysis, the overall specificity profile was almost identical to that observed when including the whole library, indicating that this sub-sampling approach was valid (*Figure 7—figure supplement 2*).

Next, we compared the preferences for CMF and AcK at each position to their closest canonical amino acids, phenylalanine (Phe) and lysine (Lys). CMF was enriched at the −3 and −2 positions, where Phe is not tolerated by c-Src (*Figure 7D*). Negatively-charged amino acids (Asp and Glu) are also preferred at these positions, and the negative charge on the carboxymethyl group of CMF at neutral pH may be able to mimic this recognition. c-Src has a strong selective preference for Phe at the +3 position, which it engages via a well-formed hydrophobic pocket near the active site (*Bose et al., 2006*; *Shah et al., 2018*). The charged carboxymethyl group on CMF is likely to be incompatible

with this mode of binding, consistent with depletion of CMF at this site (*Figure 7D*). The difference between Lys and AcK was even more striking. Lys is unfavorable for c-Src at every position around the phospho-acceptor tyrosine. By contrast, AcK was not only tolerated, but even favorable at a few positions (*Figure 7D*).

To determine whether the position-specific responsiveness to lysine acetylation was kinase-dependent, we also performed additional screens of the AcK-containing $X_5$-Y-$X_5$ library with Hck and c-Abl. These screens showed that all three kinases had very similar position-dependent tolerance for AcK over Lys, with the closely-related c-Src and Hck being more similar to one another than their distant relative c-Abl (*Figure 7D*). Finally, we assessed how the effect of lysine acetylation translated to actual changes in phosphorylation rates. We produced variants of the c-Src and c-Abl consensus peptides with Lys or AcK at various positions and measured their rates of phosphorylation by their respective cognate kinases (*Figure 7—figure supplement 3*). Of the positions tested (−2,+1, and +5 relative to the tyrosine), we saw the largest effect at the +1 position, consistent with the screens. At the +1 position, where Lys is not tolerated, acetylation enhanced activity as much as five-to-ten-fold, depending on the peptide concentration. In the long-term, we envision using this approach to predict sites in the proteome where lysine acetylation creates new, high-activity substrates for tyrosine kinases. Furthermore, the same analysis could be applied to other tyrosine kinases and to SH2 domains, and our strategy could be readily expanded to other post-translational modifications that can be encoded using Amber suppression.

## Concluding remarks

In this report, we describe a significant expansion to a previously developed method for profiling the sequence specificities of tyrosine kinases and SH2 (phosphotyrosine recognition) domains (*Cantor et al., 2018*; *Shah et al., 2018*; *Shah et al., 2016*). Our method relies on bacterial display of DNA-encoded peptide libraries and deep sequencing, and it enables the simultaneous analysis of multiple phosphotyrosine signaling proteins against thousands-to-millions of peptides or phosphopeptides. The resulting data can be used to design high-activity consensus sequences, predict the activities of uncharacterized sequences, and accurately measure the effects of amino acid substitutions on sequence recognition. A notable feature of our platform is that it relies on deep sequencing as a readout, yielding quantitative results. Furthermore, the data generated from our screens show a strong correlation with phosphorylation rates and binding affinities measured using orthogonal biochemical assays.

We envision a number of exciting applications of this expanded specificity profiling platform. Several recent reports have aimed to explain the molecular basis for tyrosine kinase and SH2 sequence specificity and affinity, by combining protein sequence and structure analysis with specificity profiling data (*Bradley et al., 2021*; *Creixell et al., 2015a*; *Kaneko et al., 2010*; *Liu et al., 2019*). The rich datasets generated using our platform will augment these approaches, particularly when coupled with screening data for additional proteins. A long-term goal of these efforts will undoubtedly be to accurately predict the sequence specificity and signaling properties of any uncharacterized phosphotyrosine signaling protein, such as a disease-associated kinase variant (*Creixell et al., 2015b*). Given the nature of the data generated by our platform, we expect that it will also aid the development and implementation of machine learning models for sequence specificity and design (*Creixell et al., 2015a*; *Cunningham et al., 2020*; *Kundu et al., 2013*). Indeed, our initial efforts in this realm suggest that specificity profiling data using the $X_5$-Y-$X_5$ library, without any protein structural information, may be sufficient to build models of sequence specificity that can accurately predict phosphorylation rates (*Rube et al., 2022*).

The pTyr-Var Library described in this report provides a unique opportunity to investigate variant effects across the human proteome. The vast majority of mutations near tyrosine phosphorylation sites are functionally uncharacterized (*Hornbeck et al., 2019*; *Krassowski et al., 2018*). Our screens are yielding some of the first mechanistic biochemical hypotheses about how many of these mutations could impact cell signaling. For example, these datasets will allow us to identify mutations that tune signaling pathways by altering the phosphorylation efficiency of specific phosphosites or the binding of SH2-containing effector proteins to those sites. Alternatively, these screens may help identify instances of network rewiring, in which a phosphosite-proximal mutation alters the canonical topology of a pathway by changing which kinases phosphorylate a phosphosite or which SH2-containing

proteins get recruited to that site. The biological effects of signal tuning and rewiring caused by phosphosite-proximal mutations remain largely unexplored.

Our high-throughput platform to profile tyrosine kinase and SH2 sequence recognition is accessible and easy to use in labs that are equipped to culture *E. coli* and execute common molecular biology and biochemistry techniques. Screens can be conducted on the benchtop with proteins produced in-house or obtained from commercial vendors. Peptide libraries of virtually any composition, tailored to address specific biochemical questions, can be produced using commercially available oligonucleotides and standard molecular cloning techniques. Furthermore, facile chemical changes to the library (e.g. enzymatic phosphorylation or the introduction of non-canonical amino acids via Amber suppression) afford access to new biochemical questions. For example, the tyrosine-phosphorylated libraries described here will also be useful for the characterization of tyrosine phosphatase specificity, and acetyl-lysine-containing libraries could be used to profile lysine deacetylases and bromodomains. Additional amendments to this platform will enable the analysis of serine/threonine kinases and other protein modification or recognition domains, adding to the growing arsenal of robust methods for the high-throughput biochemical characterization of cell signaling proteins.

## Materials and methods

### Expression and purification of tyrosine kinase domains

Constructs for the kinase domains of c-Src, c-Abl, Fyn, Hck, AncSZ, Fer, FGFR1, FGFR3, EPHB1, EPHB2, and MERTK all contained an N-terminal His$_6$-tag followed by a TEV protease cleavage site. These proteins were co-expressed in *E. coli* BL21(DE3) cells with the YopH tyrosine phosphatase. Cells transformed with YopH and the tyrosine kinase domains were grown in LB supplemented with 100 μg/mL ampicillin and 100 μg/mL streptomycin at 37 °C. Once cells reached an optical density of 0.5 at 600 nm, 500 uM of Isopropyl-β-D-1-thiogalactopyranoside (IPTG) was added to induce the expression of proteins and the cultures were incubated at 18 °C for 14–16 hours. Cells were harvested by centrifugation (4000 rpm at 4 °C for 30 min), resuspended in a lysis buffer containing 50 mM Tris, pH 7.5, 300 mM NaCl, 20 mM imidazole, 2 mM β-mercaptoethanol (BME), 10% glycerol, plus protease inhibitor cocktail, and lysed using sonication (Fisherbrand Sonic Dismembrator). After separation of insoluble material by centrifugation (33,000 g at 4 °C for 45 min), the supernatant was applied to a 5 mL HisTrap Ni-NTA column (Cytiva). The resin was washed with 10 column volumes of lysis buffer and wash buffer containing 50 mM Tris, pH 8.5, 50 mM NaCl, 20 mM imidazole, 2 mM BME, 10% glycerol. The protein was eluted with 50 mM Tris, pH 8.5, 300 mM NaCl, 500 mM imidazole, 2 mM BME, and 10% glycerol.

The eluted protein was further purified by anion exchange on a 5 mL HiTrap Q column (Cytiva) and eluted with a gradient of 50 mM to 1 M NaCl in 50 mM Tris, pH 8.5, 1 mM TCEP-HCl and 10% glycerol. The His$_6$-TEV tag of the collected fractions were cleaved by the addition of 0.10 mg/mL TEV protease overnight. The reaction mixture was subsequently flowed through 2 mL of Ni-NTA resin (ThermoFisher). The cleaved protein was collected in the flow-through and washes, then concentrated by centrifugation in an Amicon Ultra-15 30 kDa MWCO spin filter (Millipore). The concentrate was separated on a Superdex 75 16/600 gel filtration column (Cytiva), equilibrated with 10 mM HEPES, pH 7.5, 100 mM NaCl, 1 mM TCEP, 5 mM MgCl$_2$, 10% glycerol. Pure fractions were pooled, aliquoted, and flash frozen in liquid N$_2$ for long-term storage at –80 °C.

### Expression and purification of biotinylated SH2 domains

Grb2 SH2 (56-152), c-Src SH2 (143-250), and SHP2 CSH2 (105-220) domains were cloned into a His$_6$-SUMO-SH2-Avi construct and were co-expressed with biotin ligase BirA in *E. coli* C43(DE3) cells. Specifically, cells transformed with both BirA and SH2 domains were grown in LB supplemented with 100 μg/mL kanamycin and 100 μg/mL streptomycin at 37 °C until cells reached an optical density of 0.5 at 600 nm. The temperature was brought down to 18 °C, protein expression was induced with 1 mM IPTG, and the media was also supplemented with 250 μM biotin to facilitate biotinylation of the Avi-tagged SH2 domains *in vivo*. Proteins expression was carried out at 18 °C for 14–16 hours. After removal of media by centrifugation, the cells were resuspended in a lysis buffer containing 50 mM Tris, pH 7.5, 300 mM NaCl, 20 mM imidazole, 10% glycerol, and 2 mM BME, supplemented with protease inhibitor cocktail. The cells were lysed using sonication

(Fisherbrand Sonic Dismembrator), and the lysate was clarified by ultracentrifugation. The supernatant was applied to a 5 mL Ni-NTA column (Cytiva). The resin was washed with 10 column volumes each of buffers containing 50 mM Tris, pH 7.5, 300 mM NaCl, 20 mM imidazole, 10% glycerol, and 2 mM BME and 50 mM Tris, pH 7.5, 50 mM NaCl, 20 mM imidazole, 10% glycerol, and 2 mM BME. The protein was eluted in a buffer containing 50 mM Tris pH 7.5, 300 mM NaCl, 500 mM imidazole, 10% Glycerol.

The eluted protein was further purified by ion exchange on a 5 mL HiTrap Q anion exchange column (Cytiva). The following buffer was used: 50 mM Tris, pH 7.5, 50 mM NaCl, 1 mM TCEP and 50 mM Tris, pH 7.5, 50 mM NaCl, 1 mM TCEP. The protein was eluted off the column over a salt gradient from 50 mM to 1 M NaCl. The $His_6$-SUMO tag was cleaved by addition of 0.05 mg/mL Ulp1 protease. The reaction mixture was flowed over 2 mL Ni-NTA column (ThermoFisher) to remove the Ulp1, the uncleaved protein, and $His_6$-SUMO fragments. The cleaved protein was further purified by size-exclusion chromatography on a Superdex 75 16/60 gel filtration column (Cytiva) equilibrated with buffer containing 20 mM HEPES, pH 7.4, 150 mM NaCl, and 10% glycerol. Pure fractions were pooled, aliquoted, and flash frozen in liquid $N_2$ for long-term storage at –80 °C.

## Synthesis and purification of peptides for *in vitro* validation measurements

All the peptides used for *in vitro* kinetic assays were synthesized using 9-fluorenylmethoxycarbonyl (Fmoc) solid-phase peptide chemistry. All syntheses were carried out using the Liberty Blue automated microwave-assisted peptide synthesizer from CEM under nitrogen atmosphere, with standard manufacturer-recommended protocols. Peptides were synthesized on MBHA Rink amide resin solid support (0.1 mmol scale). Each Nα-Fmoc amino acid (6 eq, 0.2 M) was activated with diisopropylcarbodiimide (DIC, 1.0 M) and ethyl cyano(hydroxyamino)acetate (Oxyma Pure, 1.0 M) in dimethylformamide (DMF) prior to coupling. Each coupling cycle was done at 75 °C for 15 s then 90 °C for 110 s. Deprotection of the Fmoc group was performed in 20% (v/v) piperidine in DMF (75 °C for 15 s then 90 °C for 50 s). The resin was washed (4 x) with DMF following Fmoc deprotection and after Nα-Fmoc amino acid coupling. All peptides were acetylated at their N-terminus with 10% (v/v) acetic anhydride in DMF and washed (4 x) with DMF.

After peptide synthesis was completed, including N-terminal acetylation, the resin was washed (3 x each) with dichloromethane (DCM) and methanol (MeOH) and dried under reduced pressure overnight. The peptides were cleaved and the side chain protecting groups were simultaneously deprotected in 95% (v/v) trifluoroacetic acid (TFA), 2.5% (v/v) triisopropylsilane (TIPS), and 2.5% water ($H_2O$), in a ratio of 10 μL cleavage cocktail per mg of resin. The cleavage-resin mixture was incubated at room temperature for 90 min, with agitation. The cleaved peptides were precipitated in cold diethyl ether, washed in ether, pelleted, and dried under air. The peptides were redissolved in 50% (v/v) water/acetonitrile solution and filtered from the resin.

The crude peptide mixture was purified using reverse-phase high-performance liquid chromatography (RP-HPLC) on a semi-preparatory C18 column (Agilent, ZORBAX 300 SB-C18, 9.4x250 mm, 5 μm) with an Agilent HPLC system (1260 Infinity II). Flow rate was kept at 4 mL/min with solvents A ($H_2O$, 0.1% (v/v) TFA) and B (acetonitrile, 0.1% (v/v) TFA). Peptides were generally purified over a 40-min linear gradient from solvent A to solvent B, with the specific gradient depending on the peptide sample. Peptide purity was assessed with an analytical column (Agilent, ZORBAX 300 SB-C18, 4.6x150 mm, 5 μm) at a flow rate of 1 mL/min over a 0–90% B gradient in 30 minutes. All peptides were determined to be ≥95% pure by peak integration. The identities of the peptides were confirmed by mass spectroscopy (Waters Xevo G2-XS QTOF). Pure peptides were lyophilized and redissolved in 100 mM Tris, pH 8.0, as needed for experiments.

## Preparation of the $X_5$-Y-$X_5$ and pTyr-Var libraries for specificity profiling

All bacterial display libraries used in this study are embedded within the pBAD33 plasmid (chloramphenicol resistant), with the surface-display construct inducible by L-(+)-arabinose (*Rice and Daugherty, 2008*). All libraries have the same general structure:

[signal sequence: MKKIACLSALAAVLAFTAGTSVA]-[GQSGQ]-[peptide-coding sequence]-[GGQSGQ]-[eCPX scaffold]-[GGQSGQ]-[strep-tag: WSHPQFEK or myc-tag: EQKLISEEDL]

The X$_5$-Y-X$_5$ library contains 11-residue peptide sequences with five randomized amino acids flanking both sides of a fixed central tyrosine residue. The library was produced using the X$_5$-Y-X$_5$ library oligo, with each X encoded by an NNS codon, and Y encoded by a TAT codon (see key resources table for all primer sequences). This oligo included a 5' SfiI restriction site and DNA sequences encoding the flanking linkers that connect library peptide sequences to the 5' signal sequence and 3' eCPX scaffold.

The sequences in the pTyr-Var library were derived from the PhosphoSitePlus database and include 3159 human tyrosine phosphorylation sites and 4,760 variants of these phosphosites bearing a single amino acid mutation (*Hornbeck et al., 2019*). The sequences in this library are named as 'GeneName_pTyr-position' and 'GeneName_pTyr-position' (e.g. 'SRC_Y530' and 'SRC_Y530_527 K'). In this initial list, about 2,133 sequences had more than one tyrosine residue, and so a second version of those sequences were included in which the tyrosines except the central tyrosine were substituted with phenylalanine (denoted with a '_YF' suffix). In addition, 24 previously reported consensus substrate sequences were included (*Begley et al., 2015*; *Deng et al., 2014*; *Marholz et al., 2018*; *Rube et al., 2022*; *Songyang et al., 1995*). In total, our designed pTyr-Var library contained 9,898 unique 11-residue peptide sequences, which were then converted into DNA sequences using the most frequently used codon in *E. coli*. The DNA sequences were further optimized, swapping synonymous codons to achieve a GC content of all sequence between 30% and 70%. Sequences were also inspected and altered to remove any internal SfiI recognition sites. The 33-base peptide-coding sequences were flanked by 5'-GCTGGCCAGTCTGGCCAG-3' on the 5' side and 5'- GGAGGGCA GTCTGGGCAGTCTG-3' on the 3' side, the same flanks used for the X$_5$-Y-X$_5$ library oligo. An oligonucleotide pool based on all 9,898 sequences was generated by on-chip massively parallel synthesis (Twist Bioscience). This oligo-pool was amplified by PCR in ten cycles with the Oligopool-fwd-primer and Oligopool-rev-primer, using the NEB Q5 polymerase with a slow ramping speed (2 °C/s) and long denaturation times.

Next, we integrated the oligonucleotide sequences encoding the X$_5$-Y-X$_5$ and pTyr-Var library into a pBAD33 vector as a fusion to the eCPX bacterial display scaffold, in a series of steps. The eCPX gene was previously fused to a sequence encoding a 3' strep-tag (pBAD33-eCPX-cStrep) (*Shah et al., 2018*), and we produced a myc-tagged eCPX construct analogously, using standard molecular cloning techniques (pBAD33-eCPX-cMyc). The coding sequences for the eCPX-strep and eCPX-myc constructs were amplified from these plasmids by PCR using the link-eCPX-fwd primer and the link-eCPX-rev primer. These PCR products contained a 3' SfiI restriction site. The peptide-coding sequences were then fused to the eCPX scaffold at the 5' end of the scaffold in another PCR step to generate the library-scaffold inserts. For the X$_5$-Y-X$_5$ Library, this step used the X$_5$-Y-X$_5$ library oligo and the link-eCPX-rev primer, along with the amplified eCPX gene. For the pTyr-Var library, this step used the amplified oligo-pool, the amplified eCPX gene, and the Oligopool-fwd-primer and link-eCPX-rev primer. The resulting PCR products contained the peptide-scaffold fusion constructs flanked by two unique SfiI sites.

In parallel, the pBAD33-eCPX backbone was amplified by PCR from the pBAD33-eCPX plasmid using the BB-fwd-primer and BB-rev primer. Both the amplified insert and backbone were purified over spin columns and then digested with the SfiI restriction endonuclease overnight at 50 °C. After digestion, the backbone was treated with Quick CIP (NEB) to prevent self-ligation from occurring. Both the digested insert and backbone were gel purified. The purified library insert was ligated into the digested pBAD33-eCPX backbone using T4 DNA ligase (NEB) overnight at 16 °C. Typically, this reaction was done with a total of approximately 1.5 µg of DNA, with a 1:5 molar ratio of backbone:insert. The ligation reaction was concentrated and desalted over a spin column and then used to transform commercial DH5a cells by electroporation. The transformed DH5a cells were grown in liquid culture overnight, and the plasmid DNA was isolated and purified using a commercial midiprep kit (Zymo).

## Experimental procedure for high-throughput specificity screening of tyrosine kinases

### Preparation of cells displaying peptide libraries

The high-throughput specificity screens for tyrosine kinases using the X$_5$-Y-X$_5$ and the pTyr-Var peptide library were carried out as described previously (*Shah et al., 2018*), with the main difference being the use of magnetic beads to isolate phosphorylated cells, rather than fluorescence-activated cell sorting. 25 µL of electrocompetent *E. coli* MC1061 F- cells were transformed with 200 ng of library

DNA. Following electroporation, the cells were resuspended in 1 mL of LB and allowed to recover at 37 °C for 1 hr with shaking. These cells were resuspended in 250 mL of LB with 25 µg/ml chloramphenicol and incubated overnight at 37 °C. Of the overnight culture, 150 µL was used to inoculate 5.5 mL of LB containing 25 µg/mL chloramphenicol. This culture was grown at 37 °C for 1–2 hr until the cells reached an optical density of 0.5 at 600 nm. Expression of the library was induced by adding arabinose to a final concentration of 0.4% (w/v). The cells were incubated at 25 °C with shaking at 220 rpm for 4 hr. Small aliquots of the cells (75–150 µL) were transferred to microcentrifuge tubes and centrifuged at 1000 g at 4 °C for 10–15 min. The media was removed and the cells were resuspended in PBS and centrifuged again. The PBS was removed and the cells were stored at 4 °C. Experiments were performed with cells stored at 4 °C between 1–4 days. Typical screens were carried out on a 50 µL to 100 µL scale, with cells that were 50% more concentrated than in culture (OD$_{600}$ value around 1.5). Thus, for a 100 µL reaction, typically 150 µL of cell culture was pelleted and washed.

## Phosphorylation of peptides displayed on cells

Phosphorylation reactions of the library were conducted with the purified kinase domain and 1 mM ATP in a buffer containing 50 mM Tris, pH 7.5, 150 mM NaCl, 5 mM MgCl$_2$, 1 mM TCEP, and 2 mM sodium orthovanadate. To achieve similar library phosphorylation levels across the kinases, an optimal concentration of kinase was determined to achieve 20–30% phosphorylation of the library after 3 minutes of incubation at 37 °C. This was assessed by flow cytometry based on anti-phosphotyrosine antibody labeling (Attune NxT, Invitrogen). To label the phosphorylated cells, 50 µL pellets were resuspended with a 1:25 dilution of the PY20-PerCP-eFluor 710 conjugate (eBioscience) in PBS containing 0.2% bovine serum albumin (BSA). The cells were incubated with the antibody for 1 hr on ice in the dark, then centrifuged, washed once with PBS with 0.2% BSA, and finally resuspended in 100 µL of PBS with 0.2% BSA. For flow cytometry analysis, 20 µL of cells were diluted in 130 µL of PBS with 0.2% BSA.

The following concentrations were used: 0.5 µM for Src, 1.5 µM for Abl, 0.4 µM for Fer, 1.5 µM for EPHB1, 1.25 µM for EPHB2, 0.1 µM for JAK2, 0.5 µM for AncSZ, 0.45 µM for FGFR1, 0.5 µM for FGFR3, and 0.7 µM for MERTK. For some tyrosine kinases, such as FGFR1, FGFR3, and MERTK, preactivation with ATP was required to enhance its kinetic activity. To accomplish this, autophosphorylation reactions were performed with 25 µM kinase and 5 mM ATP for 0.5–2 hours at 25 °C. The preactivated kinase mixture was then desalted and concentrated using an Amicon Ultra-15 30 kDa MWCO spin filter (Millipore) to remove the residual ATP.

After the desired time of library phosphorylation, kinase activity was quenched with 25 mM EDTA and the cells were washed with PBS containing 0.2% BSA. Kinase-treated cells were then labeled with a 1:1000 dilution of biotinylated 4G10 Platinum anti-phosphotyrosine antibody (Millipore) for an hour on ice and washed with PBS containing 0.1% BSA and 2 mM EDTA (isolation buffer). The cells were finally resuspended in PBS containing 0.1% BSA. The phosphorylated, antibody-labeled cells were then mixed with magnetic beads from Dynabeads FlowComp Flexi kit (Invitrogen), at a ratio of 37.5 µL of washed beads per 50 µL of cell suspension, diluted into 450 µL of isolation buffer. The suspension was rotated at 4 °C for 30 minutes, then 375 µL of isolation buffer was added and the beads were separated from the bulk solution on a magnetic rack. The beads were washed once with 1 mL of isolation buffer, and then the supertantant were removed by aspiration. The beads were resuspended in 50 µL of fresh water, vortexed, and boiled at 100 °C for 10 minutes to extract DNA from cells bound to Dynabeads. The bead/lysate mixture was centrifuged to pellet the beads and the mixture was stored at –20 °C.

## DNA sample preparation and deep sequencing

To amplify the peptide-coding DNA sequence for deep sequencing, the supernatant from this lysate was used as a template in a 50 µL, 15-cycle PCR reaction using the TruSeq-eCPX-Fwd and TruSeq-eCPX-Rev primers and Q5 polymerase. The resulting mixture from this PCR reaction was used without purification as a template for a second, 20 cycle PCR reaction to append a unique pair of Illumina sequencing adapters and 5′ and 3′ indices for each sample (D700 and D500 series primers). The resulting PCR products were purified by gel extraction, and the concentration of each sample was determined using QuantiFluor dsDNA System (Promega). Each sample was pooled to equal molarity and sequenced by paired-end Illumina sequencing on a MiSeq or NextSeq instrument using a 150

cycle kit. The number of samples multiplexed in one run, and the loading density on the sequencing chip, were adjusted to obtain at least 1–2 million reads for each index/sample.

## Experimental procedure for high-throughput specificity screening of SH2 domains

### Preparation of cells displaying peptide libraries

Bacteria displaying peptide libraries for SH2 screens were prepared similarly to the bacteria for the kinase screens, with some small modifications. Specifically, after transformation with the library DNA and outgrowth of an overnight culture, 1.8 mL of the overnight culture was added to a 100 mL of LB containing 25 µg/mL of chloramphenicol. This culture was grown at 37 °C until the cells reached an optical density of 0.5 at 600 nm. Then, 20 mL of this culture was transferred to a 50 mL flask, and expression was induced by addition of arabinose to a final concentration of 0.4% (w/v). Expression was carried out at 25 °C for 4 hr, then cells were aliquoted, pelleted, and washed as described for kinase screens.

### Phosphorylation of peptides displayed on cells

Phosphorylation of cells was performed in a buffer containing 50 mM Tris, pH 7.5, 150 mM NaCl, 5 mM $MgCl_2$, 1 mM TCEP. A mixture of 2.5 µM of c-Abl kinase domain, 2.5 µM c-Src kinase domain, 2.5 µM of EPHB1 kinase domain, 2.5 µM of AncSZ, 50 µg/mL rabbit muscle creatine phosphokinase, and 5 mM creatine phosphate was prepared in this buffer. Cells were resuspended in this solution such that a pellet derived from 50 µL of cell culture was resuspended in 50 µL of solution. To initiate the phosphorylation reaction, ATP was added from a concentrated stock to a final concentration of 5 mM, and the mixture was incubated at 37 °C for 3 hr. Following this, the kinase activity was quenched by addition of 25 mM EDTA. Library phosphorylation was assessed by flow cytometry based on anti-phosphotyrosine antibody labeling, as described above for the kinase screens (Attune NxT, Invitrogen).

### Preparation of magnetic beads functionalized with SH2 domains (SH2-dynabeads)

First, 37.5 µL of magnetic beads from the Dynabeads FlowComp Flexi kit (Invitrogen) were washed with 1 mL of SH2 screen buffer containing 50 mM HEPES, pH 7.5, 150 mM NaCl. After washing, the beads were resuspended in 75 µL of 20 µM biotinylated SH2 domain and incubated at 4 °C for 2.5–3 hr. Unbound SH2 domain protein was removed by washing twice with 1 mL of SH2 screen buffer twice. The beads were finally resuspended in 37.5 µL of SH2 screen buffer.

### Selection with SH2-dynabeads

Fifty µL of the phosphorylated cells were centrifuged at 4000 $g$ at 4 °C for 15 min. After the supernatant was discarded, the cells were resuspended in SH2 screen buffer with 0.1% BSA, mixed with 37.5 µL of SH2-dynabeads, and rotated for 1 hr at 4 °C. Then, the magnetic beads were separated from the bulk solution using a magnetic rack, and the supernatant was removed by aspiration. After the supernatant was discarded, the SH2-beads were washed by incubating them with 1 mL of SH2 screen buffer for 30 min at 4 °C. After discarding the wash solution, the beads were resuspended in 50 µL of fresh water, vortexed, and boiled at 100 °C for 10 min to extract DNA from cells bound to SH2-dynabeads. DNA samples were prepped and sequenced identically as done for the kinase screens.

## Procedure for incorporating non-canonical amino acids in the high-throughput specificity screen

### General protocol

*E. coli* MC1061 electrocompetent bacteria were transformed with genetically-encoded peptide libraries and grown in liquid LB media as described in the regular screens, but with an additional plasmid encoding the corresponding non-canonical amino acid aminoacyl synthetase and tRNA pair and the addition of 100 µg/mL ampicillin to the growth medium. The cells were grown to an optical density of 0.5 at 600 nm. Peptide expression was induced with 0.4% (w/v) arabinose, 1 mM

isopropylβ-D-1-thiogalactopyranoside (IPTG), and 5 mM CMF, 5 mM AzF, or 10 mM AcK, and incubated at 25 °C for 4 h. Cell pellets were collected and washed in PBS as described in the regular screens. Bacteria bearing surface-displayed peptides containing the non-canonical amino acid of interest were phosphorylated with 0.5 µM Src kinase for 3 min using the same buffer conditions as in the regular kinase screens. The reaction was carried out in buffer containing 50 mM Tris, 150 mM NaCl, 5 mM MgCl$_2$, pH 7.5, 1 mM TCEP, and 2 mM activated sodium orthovanadate for 3 min. The reactions were initiated with 1 mM ATP and quenched with 25 mM EDTA, then washed with PBS containing 0.2% BSA, as described for the regular screens. Downstream processing of the samples, including phospho-tyrosine labeling, separation using magnetic beads, and deep sequencing were done exactly as in the regular kinase screens.

## Fluorophore labeling of surface-displayed AzF using click chemistry

The DIBO labeling solution was prepared by dissolving 0.5 mg of DIBO-alkyne Alexa Fluor 555 dye (ThermoFisher) in dimethyl sulfoxide (DMSO) to a concentration of 1 mM, and the solution was kept protected from light. The c-Myc tag labeling solution was prepared by a 1:100 dilution of c-Myc Alexa Fluor 488 conjugate (ThermoFisher) in PBS containing 0.2% BSA. The cell pellets treated with AzF were resuspended in 50 µM of the DIBO labeling solution and incubated overnight at RT with gentle nutation, protected from light (*Tian et al., 2014*). The cell suspension was pelleted and washed 4 x in PBS containing 0.2% BSA to ensure all excess DIBO dye was removed. The cell pellets were then resuspended in the c-Myc antibody solution and incubated on ice for 1 hr, protected from light. The cell suspension was pelleted and washed using PBS with 0.2% BSA. The pellets were resuspended in PBS with 0.2% BSA and analyzed by flow cytometry (Attune NxT, Invitrogen).

## A note about replicates for the bacterial peptide display screens

We define technical replicates as sets of screens conducted with library-expressing cells that are all derived from the same library transformation reaction. Biological replicates are screens done using different transformations with the library DNA, often on different days. The replicates in this study are generally all biological replicates or two biological replicate sets of two to three technical replicates.

## Processing and analysis of deep sequencing data from high-throughput specificity screens

The raw paired-end reads for each index pair from an Illumina sequencing run were merged using the FLASH (*Magoč and Salzberg, 2011*). The resulting merged sequences were then searched for the following 5' and 3' flanking sequences surrounding the peptide-coding region of the libraries: 5' flanking sequence = 5'-NNNNNNACCGCAGGTACTTCCGTAGCTGGCCAGTCTGGCCAG-3', and 3' flanking sequence = 5'-GGAGGGCAGTCTGGGCAGTCTGGTGACTACAACAAAANNNNNN-3'. These flanks were removed using the software Cutadapt to yield a filed named 'SampleName.trimmed.fastq' (*Martin, 2011*). Sequences that did not contain both flanking regions were discarded at this stage (typically less than 5%). From this point onward, all analysis was carried out using Python scripts generated in-house, which can be found in a GitHub repository https://github.com/nshahlab/2022_Li-et-al_peptide-display (copy archived at *Li et al., 2023*). Trimmed and translated FastQ and FastA files for all data used in this paper can be found in a Dryad repository (https://doi.org/10.5061/dryad.0zpc86727).

## Analysis of data from screens with thepTyr-Var Library

For the samples screened with the pTyr-Var Library, we ran scripts that identify every 33 base trimmed DNA sequence, translate those DNA sequences into amino acid sequences, count the abundance of each translated sequence that matches a peptide in the pTyr-Var library. In one format of this analysis, we used the countPeptides.py script on a trimmed input file, or batch-countPeptides.py script for multiple input files, to generate a list of every unique peptide and its corresponding counts. In a second format of this analysis, we used the countPeptides-var-ref.py (or batch-countPeptides-var-ref.py), along with paired text files listing each variant (pTyr-Var_variant.txt) and their corresponding reference sequence (pTyr-Var_reference.txt), line-by-line, to yield side-by-side counts for each variant-reference pair. These processing steps were conducted for both selected samples (after kinase phosphorylation or SH2 binding), as well as unselected input samples. Next, the number of reads for every

sequence ($n_{peptide}$) was normalized to the total number of peptide-coding reads in that sample ($n_{total}$), to yield a frequency ($f_{peptide}$, **equation 1**). Then, the frequency of each peptide in a selected sample ($f_{peptide,selected}$) was further normalized to the frequency of that same peptide in the unselected input sample ($f_{peptide,input}$) to yield an enrichment score ($E_{peptide}$, **equation 2**).

$$f_{peptide} = \frac{n_{peptide}}{n_{total}} \tag{1}$$

$$E_{peptide} = \frac{f_{peptide,selected}}{f_{peptide,input}} \tag{2}$$

## Analysis of data from screens with the $X_5$-Y-$X_5$ Library

For data from the $X_5$-Y-$X_5$ library, we did not calculate enrichments for individual sequences, as the sequencing depth per sample was generally on-par with the library size was ($10^6$–107 sequences). Instead, we computed the counts for each amino acid (or a stop codon) at every position along peptides of the expected length (11 amino acid residues). To accomplish this, we first translated all of the DNA sequences in the trimmed sequencing files using the translateUnique.py (or batch-translateUnique.py) script. When stop codons were encountered, they were translated as an asterisk symbol. In addition to producing a file of translated reads named 'SampleName.translate.fasta', this script also produced lists of every unique translated 11-residue peptide and the corresponding counts for that peptide. These files allowed us to assess whether any individual sequence was disproportionately enriched (not expected for a single round of selection with a library of this size), how many unique sequences were in each sample, and what fraction of the unique sequences contained a stop codon.

Using the translated read files, we then calculated the position-specific amino acid counts in three formats. In the simplest format, we exclusively counted 11-residue sequences that contained a central tyrosine and no stop codons (AA-count-nostop.py and batch-AA-count-nostop.py). In order to calculate stop codon depletion, we run a version of the script that counted amino acid and stop codon composition across all 11-residue sequences (AA-count-full.py and batch-AA-count-full.py). Finally, for Amber suppression datasets, we exclusively counted sequences containing one stop and a central tyrosine residue (AA-count-1stop.py and batch-AA-count-1stop,py). Each of these scripts generated an 11x21 counts matrix with each position in the peptide represented by a column (from –5 to +5), and each row represented by an amino acid (in alphabetical order, with the stop codon in the 21st row). Frequencies of each amino acid at each position were determined by taking the position-specific count for each amino acid and dividing that by the column total. Frequencies in a matrix from a selected sample were further normalized against frequencies from an input sample, and the resulting enrichment values were $\log_2$-transformed to yield the data represented in the heatmaps in **Figures 1, 2, 6 and 7**.

## Scoring sequences using data from the $X_5$-Y-$X_5$ Library

In order to score peptides using position-weighted counts matrixes from the $X_5$-Y-$X_5$ Library, we wrote a Python script called score_peptide_nostop.py. This script requires the selected and input counts matrices for a kinase or SH2 domain, produced by the AA-count-nostop.py script, along with a list of peptides as a text file, with one peptide per line. The script first calculates the normalized enrichments for each amino acid at each position across the matrices. Then, it reads each target sequence, sums up the $\log_2$-normalized enrichments for each residue according to the enrichment matrix, ignoring the central tyrosine, and divides the sum by the number of scored residues (10 for the $X_5$-Y-$X_5$ Library). The script also calculates the score for the best and worst sequence, according to the enrichment matrix. Both unnormalized and normalized scores for the whole peptide list are outputted as text files.

## *In vitro* measurements of phosphorylation rates with purified kinases and peptides

### RP-HPLC assay to measure peptide phosphorylation kinetics

To validate the enrichment scores observed in the c-Src screening data, the phosphorylation rates were measured *in vitro* with the purified catalytic domain of c-Src and synthetic 11-residue peptides derived from sequences in the pTyr-Var library. Kinetic measurements were carried out at 37 °C by mixing 500 nM c-Src and 100 µM peptide in a buffer containing 50 mM Tris, pH 7.5, 150 mM NaCl, 5 mM $MgCl_2$, 1 mM TCEP, and 2 mM activated sodium orthovanadate. Reactions were initiated by

adding 1 mM ATP. At various time points, 100 μL aliquots were removed and quenched by the addition of EDTA to a final concentration of 25 mM. Each time point sample was analyzed by analytical RP-HPLC, monitoring absorbance at 214 nm. Forty μL of each time point was injected onto a C18 column (ZORBAX 300 SB-C18, 5 μm, 4.6x150 mm). The solvent system used was water with 0.1% trifluoroacetic acid (solvent A) and acetonitrile with 0.1% trifluoroacetic acid (solvent B). Peptides were eluted at a flow rate of 1 mL/min, using the following set of linear gradients: 0–2 min: 5% B, 2–12 min: 5–95% B, 12–13 min: 95% B, 13–14 min: 95–5% B, and 14–17 min: 5% B. The areas under the peaks corresponding to the unphosphorylated and phosphorylated peptides were calculated using the Agilent OpenLAB ChemStation software. The fractional product peak area was plotted as a function of reaction time, and the initial linear regime of this plot was fitted to a straight line to determine a reaction rate. Rates were corrected for substrate and enzyme concentration. Reactions were done in triplicate or quadruplicate.

## Michaelis-Menten analysis using the ADP-Quest assay

A fluorescence-based assay from Eurofins (ADP Quest) was used to measure the Michaelis-Menten kinetic parameters for phosphorylation of the consensus peptides by purified tyrosine kinase domains. In this assay, ADP production as a result of kinase activity is coupled to the production of resorufin, a fluorophore that emits signal at 590 nm. For all experiments, the assay reactions were set up as described in the provided assay kit protocol, in a 384 well plate format. The peptide solutions were serially diluted in 100 mM Tris, pH 8.0, and the kinases were diluted to 50 nM in buffer (10 mM HEPES, 100 mM NaCl, 1 mM TCEP, 5 mM MgCl$_2$, 10% (v/v) glycerol). The final reaction mixtures contained 10 nM of kinase with 100 μM of ATP. Reactions were initiated with the addition of 1 mM ATP into a 50 μL reaction mixture for a final concentration of 100 μM ATP. Phosphorylation reaction progress was monitored by measuring fluorescence at excitation 530 nm and emission 590 nm every 2 min at 37 °C on a plate reader (BioTek Synergy Neo 2). The fluorescence units (RFU) were converted to μM ADP by comparison to a standard curve, and the initial rates were extracted from the linear regime of the reaction progress curves. Initial rates were also measured for samples containing each kinase but lacking a peptide substrate, to account for background ATP hydrolysis. This background rate was subtracted from the rates measured in the presence of peptide. The subtracted rates were plotted as a function of substrate concentration and fit to the Michaelis-Menten equation to extract $k_{cat}$ and $K_M$ values.

## *In vitro* measurements of binding affinities with purified SH2 and phospho-peptides

Binding affinities of SH2 domains and phospho-peptides were measured using fluorescence polarization-based competition binding assay, following previously reported methods (*Cushing et al., 2008*). The fluorescent peptide (FITC-Acp-GDG(pY)EEISPLLL) used for $K_D$ measurements was a gift from the Amacher lab. A buffer containing 60 mM HEPES, pH 7.2, 75 mM KCl, 75 mM NaCl, 1 mM EDTA, and 0.05% Tween 20 was used for the experiments. For $K_D$ measurement, varying concentrations of the c-Src SH2 protein were incubated with 30 nM fluorescent peptide for 15 min in a black, half-area, 96-well plate. The plate was centrifuged for 5 min at 1000 *g* to remove air bubbles. Following this, fluorescence polarization data was collected on a plate reader at 25 °C (BioTek Synergy Neo 2). The samples were excited at a wavelength of 485 nm and emission data was collected at 525 nm. Data was analyzed and fitted to a quadratic binding equation to determine the $K_D$ for the fluorescent peptide with c-Src. A $K_D$ of 160 nM was obtained for the fluorescent peptide with the c-Src SH2 domain, and this value was used in subsequent calculations for the competition binding experiments.

Competition binding experiments were performed similarly. A stock solution was prepared by mixing 60 nM fluorescent peptide with SH2 domain at a concentration of 480 nM (3 x $K_D$) and incubated at room temp for 15 min. Unlabeled competitor peptide was serially diluted in buffer. Each serial dilution was mixed with fluorescent peptide-SH2 stock solution at a 1:1 ratio in a black, half-area, 96 well plate. After mixing the samples by pipetting, the plate was centrifuged at 1000 *g* for 5 min to remove air bubbles. The final fluorescent peptide concentration was 30 nM and the final SH2 concentration was 1.5 x $K_D$ (240 nM). Fluoresce polarization was measured as previously described for initial $K_D$ measurements. Competition binding data were fit to a cubic binding equation as described previously (*Cushing et al., 2008*).

### *In vitro* measurements of phosphorylation rates with purified kinases and SHP2 substrate

### Expression and Purification of SHP2 WT, D61V, and D61N

All SHP2 variants contained a catalytic cysteine mutation (C459E), C-terminal tail (526-593) deletion, and N-terminal His$_6$-tag followed by a TEV protease cleavage site. The same protocol used to express and purify SH2 domains, excluding co-expression of BirA and addition of biotin, was applied to the expression and purification of the SHP2 variants.

### LC-MS assay to measure protein phosphorylation kinetics

To pre-activate the kinases, 1 µM of each purified kinase domain was preincubated at 37 °C for 30 min in the same buffer conditions used in the kinase domain peptide display screen, with 1 mM ATP. The reaction of the kinase with SHP2 was initiated with the addition of 10 µM SHP2, and the mixture was incubated in 37 °C for 1 hr. To terminate the reaction, the mixture was quenched with 200 mM EDTA. The reaction mixture was diluted 3:2 in water and injected onto a BEH C8 column (Waters) on a UPLC-MS system (Xevo QToF, Waters). Reverse-phase liquid chromatography was carried out at 0.3 mL/min with solvents A (H$_2$O, 0.1% (v/v) formic acid) and B (acetonitrile, 0.1% (v/v) formic acid). Proteins were eluted over a gradient of 5–95% B for 8.5 min. The protein peak on the chromatogram was deconvoluted using the MaxEnt1 algorithm from 32,000–65,000 Da with a resolution of 1 Da/channel over 30 iterations. Peaks were chosen according to the theoretical MW of the protein within a range of 5 Da, and integrated for the signal intensity.

### Materials and data availability

The key reagents produced in this study (the X$_5$-Y-X$_5$ Library, the pTyr-Var Library, and protein expression plasmids) will be made freely available to any researcher interested in using our specificity profiling platform. Data from the specificity screens in this study, in the form of enrichment scores, are available alongside this publication as source data files. Trimmed and translated deep sequencing data (.fastq and.fasta files) are available via Dryad: https://doi.org/10.5061/dryad.0zpc86727. Code used in this study to process and analyze the data can be found in this GitHub repository: https://github.com/nshahlab/2022_Li-et-al_peptide-display (copy archived at *Li et al., 2023*). The plasmid libraries and unprocessed data can be requested by directly contacting the corresponding author.

## Acknowledgements

We thank Fereshteh Zandkarimi and Brandon Fowler from the Columbia Chemistry mass spectrometry facility for their assistance with mass spectrometry; Jia Ma from the Columbia Precision Biomolecular Characterization Facility for his guidance with biophysical measurements; and the Columbia Genome Center for their support with deep sequencing. We thank Neil Vasan for his guidance with SHP2 phosphorylation assays. We thank Harmen Bussemaker, Tomas Rube, and Chaitanya Rastogi for their insightful discussions, and members of the Shah lab for their technical and conceptual guidance throughout this project. The fluorescently-labeled c-Src SH2 ligand was a gift from the Jeanine Amacher. The pULTRA chAcKRS3 plasmid was a gift from Abhishek Chatterjee. Bacterial expression vectors for Fer, FGFR1, FGFR3, EPHB1, EPHB2, and MERTK were gifts from John Chodera, Nicholas Levinson, and Markus Seeliger (Addgene plasmid #s 79686, 79719, 79731, 79694, 79697, and 79705). The pEVOL pAzFRS.2.t1 plasmid was a gift from Farren Isaacs (Addgene plasmid #73546). This work was supported by NIH grant R35 GM138014 and a Damon Runyon-Dale F Frey Award for Breakthrough Scientists (DFS 31–18), awarded to NHS.

## Additional information

### Funding

| Funder | Grant reference number | Author |
|---|---|---|
| National Institute of General Medical Sciences | R35GM138014 | Neel H Shah |
| Damon Runyon Cancer Research Foundation | DFS 31-18 | Neel H Shah |

The funders had no role in study design, data collection and interpretation, or the decision to submit the work for publication.

### Author contributions

Allyson Li, Conceptualization, Formal analysis, Validation, Investigation, Visualization, Methodology, Writing - original draft, Writing – review and editing; Rashmi Voleti, Conceptualization, Formal analysis, Validation, Investigation, Visualization, Methodology, Writing – review and editing; Minhee Lee, Conceptualization, Validation, Investigation, Methodology, Writing – review and editing; Dejan Gagoski, Validation, Investigation, Methodology, Writing – review and editing; Neel H Shah, Conceptualization, Resources, Formal analysis, Supervision, Funding acquisition, Visualization, Methodology, Writing - original draft, Project administration, Writing – review and editing

### Author ORCIDs

Allyson Li ⓘ http://orcid.org/0000-0003-2359-7703
Rashmi Voleti ⓘ http://orcid.org/0000-0002-3705-7460
Minhee Lee ⓘ http://orcid.org/0000-0002-7141-7351
Dejan Gagoski ⓘ http://orcid.org/0000-0001-6194-8514
Neel H Shah ⓘ http://orcid.org/0000-0002-1186-0626

### Decision letter and Author response

Decision letter https://doi.org/10.7554/eLife.82345.sa1
Author response https://doi.org/10.7554/eLife.82345.sa2

## Additional files

### Supplementary files
• MDAR checklist

### Data availability

All of the processed data from the high-throughput specificity screens are provided as source data files. The raw fastq and fasta sequencing files are available as a Dryad repository (DOI:https://doi.org/10.5061/dryad.0zpc86727). Custom code used to process/analyze screening data can be found in a GitHub repository (copy archived at *Li et al., 2023*) as specified in the manuscript.

The following dataset was generated:

| Author(s) | Year | Dataset title | Dataset URL | Database and Identifier |
|---|---|---|---|---|
| Li A, Voleti R, Lee M, Gagoski D, Shah NH | 2023 | Data from: High-throughput profiling of sequence recognition by tyrosine kinases and SH2 domains using bacterial peptide display | http://dx.doi.org/10.5061/dryad.0zpc86727 | Dryad Digital Repository, 10.5061/dryad.0zpc86727 |

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

# Appendix 1

## Appendix 1—key resources table

| Reagent type (species) or resource | Designation | Source or reference | Identifiers | Additional information |
|---|---|---|---|---|
| Strain, strain background (*E. coli*) | MC1061 | Lucigen | Lucigen: 10361012 | bacterial cells used for surface-display screens |
| Strain, strain background (*E. coli*) | DH5α | Invitrogen | Invitrogen: 18265017 | bacterial cells used for general cloning and library cloning |
| Strain, strain background (*E. coli*) | BL21(DE3) | ThermoFisher Scientific | Thermo: C600003 | bacterial cells for general protein-expression; pre-transformed with pCDF-YopH for tyrosine kinase overexpression |
| Strain, strain background (*E. coli*) | C43(DE3) | Lucigen | Lucigen: NC9581214 | bacterial cells used for SH2 domain over-expression; pre-transformed with pCDFDuet-BirA-WT for biotinylation |
| Antibody | 4 G10 Platinum, Biotin (mouse monoclonal) | Millipore Sigma | Millipore Sigma: 16–452-MI | biotin conjugated mouse monoclonal pan-phosphotyrosine antibody dilution: (1:1000) |
| Antibody | PY20-PerCP-eFluor 710 (mouse monoclonal) | eBioscience | eBioscience: 46-5001-42 | PerCP-eFluor 710-conjugated mouse monoclonal pan-phosphotyrosine antibody, clone PY20 dilution: (1:25) |
| Antibody | PY20-biotin (mouse monoclonal) | Exalpha | Exalpha: 50-210-1865 | biotin conjugated mouse monoclonal pan-phosphotyrosine antibody dilution (1:500) |
| Antibody | StrepMAB Chromeo 488 (mouse monoclonal) | IBA LifeSciences | IBA: 2-1546-050 | Chromeo 488-conjugated antibody that recognizes the strep-tag dilution: (1:50–100). Discontinued, but can be replaced with IBA LifeSciences StrepMAB-Classic conjugate DY-488 (IBA: 2-1563-050) |
| Recombinant DNA reagent | pBAD33-eCPX | PMID:18480093 | Addgene: 23336 | pBAD33 plasmid encoding the eCPX bacterial display gene with flanking 5' and 3' SfiI restriction sites |
| Recombinant DNA reagent | pBAD33-eCPX-cStrep | PMID:29547119 | | pBAD33 plasmid encoding the eCPX bacterial display gene with a 3' sequence encoding a strep-tag and flanking 5' and 3' SfiI restriction sites |
| Recombinant DNA reagent | pBAD33-eCPX-cMyc | this paper | | pBAD33 plasmid encoding the eCPX bacterial display gene with a 3' sequence encoding a myc-tag and flanking 5' and 3' SfiI restriction sites |
| Recombinant DNA reagent | X5-Y-X5 Library (myc-tagged) | this paper | | peptide display library in the pBAD33 vector, fused to the eCPX scaffold, containing 1–10 million unique sequences with the structure X5-Y-X5, where X is encoded by an NNS codon. The scaffold protein is encoded to have a C-terminal myc-tag: EQKLISEEDL. |
| Recombinant DNA reagent | X5-Y-X5 Library (strep-tagged) | this paper | | peptide display library in the pBAD33 vector, fused to the eCPX scaffold, containing 1–10 million unique sequences with the structure X5-Y-X5, where X is encoded by an NNS codon. The scaffold protein is encoded to have a C-terminal strep-tag: WSHPQFEK. |
| Recombinant DNA reagent | pTyr-Var Library (myc-tagged) | this paper | | peptide display library in the pBAD33 vector, fused to the eCPX scaffold, containing ~10,000 unique sequences encoding reference and variant phosphosite pairs deried from the PhosphoSitePlus database. The scaffold protein is encoded to have a C-terminal myc-tag: EQKLISEEDL. |
| Recombinant DNA reagent | pTyr-Var Library (strep-tagged) | this paper | | peptide display library in the pBAD33 vector, fused to the eCPX scaffold, containing ~10,000 unique sequences encoding reference and variant phosphosite pairs deried from the PhosphoSitePlus database. The scaffold protein is encoded to have a C-terminal strep-tag: WSHPQFEK. |
| Recombinant DNA reagent | pET-23a-His6-TEV-Src(KD) | PMID:29547119 | | bacterial expression vector encoding the human c-Src kinase domain (residues 260–528), with an N-terminal His6-tag and TEV protease recognition sequence |

*Appendix 1 Continued on next page*

*Appendix 1 Continued*

| Reagent type (species) or resource | Designation | Source or reference | Identifiers | Additional information |
|---|---|---|---|---|
| Recombinant DNA reagent | pET-23a-His6-TEV-Fyn(KD) | PMID:29547119 | | bacterial expression vector encoding the human Fyn kinase domain (residues 261–529) with an N-terminal His6-tag and TEV protease recognition sequence |
| Recombinant DNA reagent | pET-23a-His6-TEV-Hck(KD) | PMID:29547119 | | bacterial expression vector encoding the human Hck kinase domain (residues 252–520) with an N-terminal His6-tag and TEV protease recognition sequence |
| Recombinant DNA reagent | pET-23a-His6-TEV-Abl(KD) | PMID:29547119 | | bacterial expression vector encoding the mouse c-Abl kinase domain (residues 232–502) with an N-terminal His6-tag and TEV protease recognition sequence |
| Recombinant DNA reagent | pET-23a-His6-TEV-AncSZ(KD) | DOI: 10.1101/2022.04.24.489292 | | bacterial expression vector encoding the AncSZ kinase domain (residues 352–627) with an N-terminal His6-tag and TEV protease recognition sequence |
| Recombinant DNA reagent | pET23a-His6-TEV-Fer(KD) | this paper | | bacterial expression vector encoding the mouse Fer kinase domain (residues 553–823) with an N-terminal His6-tag and TEV protease recognition sequence |
| Recombinant DNA reagent | pET-His6-TEV-FGFR1(KD) | PMID:30004690 | Addgene: 79719 | bacterial expression vector encoding the human FGFR1 kinase domain (residues 456–763) with an N-terminal His6-tag and TEV protease recognition sequence |
| Recombinant DNA reagent | pET-His6-TEV-FGFR3(KD) | PMID:30004690 | Addgene: 79731 | bacterial expression vector encoding the human FGFR3 kinase domain (residues 449–759) with an N-terminal His6-tag and TEV protease recognition sequence |
| Recombinant DNA reagent | pET-His6-TEV-EPHB1(KD) | PMID:30004690 | Addgene: 79694 | bacterial expression vector encoding the human EPHB1 kinase domain (residues 602–896) with an N-terminal His6-tag and TEV protease recognition sequence |
| Recombinant DNA reagent | pET-His6-TEV-EPHB2(KD) | PMID:30004690 | Addgene: 79697 | bacterial expression vector encoding the human EPHB2 kinase domain (residues 604–898) with an N-terminal His6-tag and TEV protease recognition sequence |
| Recombinant DNA reagent | pET-His6-TEV-MERTK(KD) | PMID:30004690 | Addgene: 79705 | bacterial expression vector encoding the human MERTK kinase domain (residues 570–864) with an N-terminal His6-tag and TEV protease recognition sequence |
| Recombinant DNA reagent | pCDF-YopH | PMID:16260764 | | bacterial expression vector for co-expression of untagged YopH phosphatase with tyrosine kinases |
| Recombinant DNA reagent | pET28-His6-TEV-SHP2-C459E-no tail | this paper | | bacterial expression vector encoding the human SHP2 (residues 1–526) with the C459E mutation, an N-terminal His6-tag, and TEV protease recognition sequence |
| Recombinant DNA reagent | pET28-His6-TEV-SHP2-C459E-no tail-D61V | this paper | | bacterial expression vector encoding the human SHP2 (residues 1–526) with C459E and D61V mutations, an N-terminal His6-tag, and TEV protease recognition sequence |
| Recombinant DNA reagent | pET28-His6-TEV-SHP2-C459E-no tail-D61N | this paper | | bacterial expression vector encoding the human SHP2 (residues 1–526) with C459E and D61N mutations, an N-terminal His6-tag, and TEV protease recognition sequence |
| Recombinant DNA reagent | pCDFDuet-BirA-WT | this paper | | bacterial expression vector encoding BirA biotin ligase, used to coexpress with SH2 domain expression vector for biotinylation of SH2 domain |

*Appendix 1 Continued on next page*

*Appendix 1 Continued*

| Reagent type (species) or resource | Designation | Source or reference | Identifiers | Additional information |
|---|---|---|---|---|
| Recombinant DNA reagent | pET-His6-SUMO-Src(SH2) | this paper | | bacterial expression vector encoding the human cSrc SH2 domain (residues 143–250) with an N-terminal His6-SUMO tag |
| Recombinant DNA reagent | pET-His6-SUMO-SHP2(CSH2) | this paper | | bacterial expression vector encoding the human SHP2 CSH2 domain (residues 105–220) with an N-terminal His6-SUMO tag |
| Recombinant DNA reagent | pET-His6-SUMO-Grb2(SH2) | this paper | | bacterial expression vector encoding the human Grb2 SH2 domain (residues 56–152) with an N-terminal His6-SUMO tag |
| Recombinant DNA reagent | pULTRA CMF | PMID:28604693 | | bacterial expression vector encoding the tRNA/syntetase pair for incorporation of 4-carboxymethyl phenylalanine via Amber suppression |
| Recombinant DNA reagent | pEVOL pAzFRS.2.t1 | PMID:26571098 | Addgene: 73546 | bacterial expression vector encoding the tRNA/syntetase pair for incorporation of 4-azido phenylalanine and other Phe derivatives via Amber suppression |
| Recombinant DNA reagent | pULTRA chAcKRS3 | PMID:29544052 | | bacterial expression vector encoding the tRNA/syntetase pair for incorporation of acetyl-lysine via Amber suppression; gift from Abhishek Chatterjee at Boston College |
| Recombinant DNA reagent | pULTRA-Amp CMF | this paper | | bacterial expression vector encoding the tRNA/syntetase pair for incorporation of 4-carboxymethyl phenylalanine via Amber suppression, altered to have an ampicillin resistance marker |
| Recombinant DNA reagent | pULTRA-Amp pAzFRS.2.t1 | this paper | | bacterial expression vector encoding the tRNA/syntetase pair for incorporation of 4-azido phenylalanine and other Phe derivatives via Amber suppression, altered to have an ampicillin resistance marker |
| Recombinant DNA reagent | pULTRA-Amp chAcKRS3 | this paper | | bacterial expression vector encoding the tRNA/syntetase pair for incorporation of acetyl-lysine via Amber suppression, altered to have an ampicillin resistance marker |
| Sequence-based reagent | X5-Y-X5 library oligo; eCPX-rand-lib | this paper, purchased from Millipore Sigma | | primer sequence: 5'-GCTGGCCAGTCTGGCCAGNNS NNSNNSNNSNNStatNNSNNSNNSNNSNNSGGAGG GCAGTCTGGGCAGTCTG 3' |
| Sequence-based reagent | Oligopool-fwd-primer | this paper, purchased from Millipore Sigma | | primer sequence: 5'-GCTGGCCAGTCTG-3' |
| Sequence-based reagent | Oligopool-rev-primer | this paper, purchased from Millipore Sigma | | primer sequence: 5'-CAGACTGCCCAGACT-3' |
| Sequence-based reagent | link-eCPX-fwd | this paper, purchased from Millipore Sigma | | 5'-GGAGGGCAGTCTGGGCAGTCTG-3' |
| Sequence-based reagent | link-eCPX-rev | this paper, purchased from Millipore Sigma | | 5'-GCTTGGCCACCTTGGCCTTATTA-3' |

*Appendix 1 Continued on next page*

*Appendix 1 Continued*

| Reagent type (species) or resource | Designation | Source or reference | Identifiers | Additional information |
|---|---|---|---|---|
| Sequence-based reagent | BB-fwd-primer | this paper, purchased from Millipore Sigma | | 5'-TAATAAGGCCAAGGTGGCCAAGC-3' |
| Sequence-based reagent | BB-rev primer | this paper, purchased from Millipore Sigma | | 5'-CTGGCCAGACTGGCCAGCTACG-3' |
| Sequence-based reagent | TruSeq-eCPX-Fwd | sequence from PMID:29547119, purchased from Millipore Sigma | round one amplicon PCR primer | primer sequence: 5'-TGACTGGAGTTCAGACGTG TGCTCTTCCGATCTNNNNNNACCGCA GGTACTTCCGTAGCT-3' |
| Sequence-based reagent | TruSeq-eCPX-Rev | sequence from PMID:29547119, purchased from Millipore Sigma | round one amplicon PCR primer | primer sequence: 5'-CACTCTTTCCCTACACGACG CTCTTCCGATCTNNNNNN TTTTGTTGTAGTCACCAGACTG-3' |
| Sequence-based reagent | D701 | sequence from Illumina, purchased from Millipore Sigma | round two amplicon/ indexing PCR primer | primer sequence: 5'-CAAGCAGAAGACGG CATACGAGATcgagtaatGTG ACTGGAGTTCAGACGTG-3' |
| Sequence-based reagent | D702 | sequence from Illumina, purchased from Millipore Sigma | round two amplicon/ indexing PCR primer | primer sequence: 5'-CAAGCAGAAGA CGGCATACGAGATtctccgga GTGACTGGAGTTCAGACGTG-3' |
| Sequence-based reagent | D703 | sequence from Illumina, purchased from Millipore Sigma | round two amplicon/ indexing PCR primer | primer sequence: 5'-CAAGCAGAAGA CGGCATACGAGATaatgagcg GTGACTGGAGTTCAGACGTG-3' |
| Sequence-based reagent | D704 | sequence from Illumina, purchased from Millipore Sigma | round two amplicon/ indexing PCR primer | primer sequence: 5'-CAAGCAGAAGAC GGCATACGAGATggaatctcG TGACTGGAGTTCAGACGTG-3' |
| Sequence-based reagent | D705 | sequence from Illumina, purchased from Millipore Sigma | round two amplicon/ indexing PCR primer | primer sequence: 5'-CAAGCAGA AGACGGCATACGAGA TttctgaatGTGACTGGAGT TCAGACGTG-3' |
| Sequence-based reagent | D706 | sequence from Illumina, purchased from Millipore Sigma | round two amplicon/ indexing PCR primer | primer sequence: 5'-CAAGCAGAAGA CGGCATACGAGATacgaattc GTGACTGGAGTTCAGACGTG-3' |
| Sequence-based reagent | D707 | sequence from Illumina, purchased from Millipore Sigma | round two amplicon/ indexing PCR primer | primer sequence: 5'-CAAGCAGAAG ACGGCATACGAGATagcttcag GTGACTGGAGTTCAGACGTG-3' |
| Sequence-based reagent | D708 | sequence from Illumina, purchased from Millipore Sigma | round two amplicon/ indexing PCR primer | primer sequence: 5'-CAAGCAGAAGACG GCATACGAGATgcgcattaGT GACTGGAGTTCAGACGTG-3' |
| Sequence-based reagent | D709 | sequence from Illumina, purchased from Millipore Sigma | round two amplicon/ indexing PCR primer | primer sequence: 5'-CAAGCAGAAG ACGGCATACGAGATcatagccg GTGACTGGAGTTCAGACGTG-3' |
| Sequence-based reagent | D710 | sequence from Illumina, purchased from Millipore Sigma | round two amplicon/ indexing PCR primer | primer sequence: 5'-CAAGCAGA AGACGGCATACGAGATttcgcgga GTGACTGGAGTTCAGACGTG-3' |
| Sequence-based reagent | D711 | sequence from Illumina, purchased from Millipore Sigma | round two amplicon/ indexing PCR primer | primer sequence: 5'-CAAGCAGAAGACG GCATACGAGATgcgcgaga GTGACTGGAGTTCAGACGTG-3' |
| Sequence-based reagent | D712 | sequence from Illumina, purchased from Millipore Sigma | round two amplicon/ indexing PCR primer | primer sequence: 5'-CAAGCAGAA GACGGCATACGAGATctatcgctGT GACTGGAGTTCAGACGTG-3' |
| Sequence-based reagent | D501 | sequence from Illumina, purchased from Millipore Sigma | round two amplicon/ indexing PCR primer | primer sequence: 5'-AATGATACGGCGA CCACCGAGATCTACACtatagcct ACACTCTTTCCCTACACGAC-3' |
| Sequence-based reagent | D502 | sequence from Illumina, purchased from Millipore Sigma | round two amplicon/ indexing PCR primer | primer sequence: 5'-AATGATACGGCG ACCACCGAGATCTACACatagaggc ACACTCTTTCCCTACACGAC-3' |
| Sequence-based reagent | D503 | sequence from Illumina, purchased from Millipore Sigma | round two amplicon/ indexing PCR primer | primer sequence: 5'-AATGATACGGCGA CCACCGAGATCTACACcctatcct ACACTCTTTCCCTACACGAC-3' |
| Sequence-based reagent | D504 | sequence from Illumina, purchased from Millipore Sigma | round two amplicon/ indexing PCR primer | primer sequence: 5'-AATGATACGGCGA CCACCGAGATCTACACggctctga ACACTCTTTCCCTACACGAC-3' |

*Appendix 1 Continued on next page*

*Appendix 1 Continued*

| Reagent type (species) or resource | Designation | Source or reference | Identifiers | Additional information |
|---|---|---|---|---|
| Sequence-based reagent | D505 | sequence from Illumina, purchased from Millipore Sigma | round two amplicon/ indexing PCR primer | primer sequence: 5'-AATGATACGGC GACCACCGAGATCTACACaggcgaag ACACTCTTTCCCTACACGAC-3' |
| Sequence-based reagent | D506 | sequence from Illumina, purchased from Millipore Sigma | round two amplicon/ indexing PCR primer | primer sequence: 5'-AATGATACGG CGACCACCGAGATCTACACtaatctta ACACTCTTTCCCTACACGAC-3' |
| Sequence-based reagent | D507 | sequence from Illumina, purchased from Millipore Sigma | round two amplicon/ indexing PCR primer | primer sequence: 5'-AATGATACGGC GACCACCGAGATCTACACcaggacgt ACACTCTTTCCCTACACGAC-3' |
| Sequence-based reagent | D508 | sequence from Illumina, purchased from Millipore Sigma | round two amplicon/ indexing PCR primer | primer sequence: 5'-AATGATACG GCGACCACCGAGATCTACAC gtactgacACACTCTTTCCCTACACGAC-3' |
| Peptide, recombinant protein | Src(KD) | this paper, expressed/purified in-house | | human c-Src kinase domain (residues 260–528) |
| Peptide, recombinant protein | Fyn(KD) | this paper, expressed/purified in-house | | human Fyn kinase domain (residues 261–529) |
| Peptide, recombinant protein | Hck(KD) | this paper, expressed/purified in-house | | human Hck kinase domain (residues 252–520) |
| Peptide, recombinant protein | Abl(KD) | this paper, expressed/purified in-house | | mouse c-Abl kinase domain (residues 232–502) |
| Peptide, recombinant protein | JAK2 Protein, active | Millipore Sigma | Millipore Sigma: 14–640 M | Active, C-terminal His6-tagged, recombinant, human JAK2, amino acids 808-end, expressed by baculo virus in Sf21 cells, for use in Enzyme Assays. |
| Peptide, recombinant protein | AncSZ(KD) | this paper, expressed/purified in-house | | AncSZ kinase domain (residues 352–627) designed by ancestral sequence reconstruction |
| Peptide, recombinant protein | Fer(KD) | this paper, expressed/purified in-house | | mouse Fer kinase domain (residues 553–823) |
| Peptide, recombinant protein | FGFR1(KD) | this paper, expressed/purified in-house | | human FGFR1 kinase domain (residues 456–763) |
| Peptide, recombinant protein | FGFR3(KD) | this paper, expressed/purified in-house | | human FGFR3 kinase domain (residues 449–759) |
| Peptide, recombinant protein | EPHB1(KD) | this paper, expressed/purified in-house | | human EPHB1 kinase domain (residues 602–896) |
| Peptide, recombinant protein | EPHB2(KD) | this paper, expressed/purified in-house | | human EPHB2 kinase domain (residues 604–898) |

*Appendix 1 Continued on next page*

*Appendix 1 Continued*

| Reagent type (species) or resource | Designation | Source or reference | Identifiers | Additional information |
|---|---|---|---|---|
| Peptide, recombinant protein | MERTK(KD) | this paper, expressed/purified in-house | | human MERTK kinase domain (residues 570–864) |
| Peptide, recombinant protein | Src(SH2) | this paper, expressed/purified in-house | | human c-Src SH2 domain (residues 143–250) |
| Peptide, recombinant protein | SHP2(C-SH2) | this paper, expressed/purified in-house | | human SHP2 C-SH2 domain (residues 105–220) |
| Peptide, recombinant protein | Grb2(SH2) | this paper, expressed/purified in-house | | human Grb2 SH2 domain (residues 56–152) |
| Peptide, recombinant protein | SHP2(PTP; C459E) | this paper, expressed/purified in-house | | human full-length SHP2 (residues 1–526; C459E) |
| Peptide, recombinant protein | SHP2(PTP; C459E, D61V) | this paper, expressed/purified in-house | | human full-length SHP2 (residues 1–526; C459E, D61V) |
| Peptide, recombinant protein | SHP2(PTP; C459E, D61N) | this paper, expressed/purified in-house | | human full-length SHP2 (residues 1–526; C459E, D61N) |
| Peptide, recombinant protein | SHP2(PTP; C459E, G60V) | this paper, expressed/purified in-house | | human full-length SHP2 (residues 1–526; C459E, G60V) |
| Peptide, recombinant protein | Src Consensus | this paper, synthesized in-house | | peptide sequence: Ac-GPDECIYDMFPFKKKG-NH2 |
| Peptide, recombinant protein | Src Consensus (P-5C, D+1 G) | this paper, synthesized in-house | | peptide sequence: Ac-GCDECIYGMFPFRRRG-NH2 |
| Peptide, recombinant protein | Abl Consensus | this paper, synthesized in-house | | peptide sequence: Ac-GPDEPIYAVPPIKKKG-NH2 |
| Peptide, recombinant protein | Fer Consensus | this paper, synthesized in-house | | peptide sequence: Ac-GPDEPIYEWWWIKKKG-NH2 |
| Peptide, recombinant protein | EPHB1 Consensus | this paper, synthesized in-house | | peptide sequence: Ac-GPPEPNYEVIPPKKKG-NH2 |
| Peptide, recombinant protein | EPHB2 Consensus | this paper, synthesized in-house | | peptide sequence: Ac-GPPEPIYEVPPPKKKG-NH2 |

*Appendix 1 Continued on next page*

*Appendix 1 Continued*

| Reagent type (species) or resource | Designation | Source or reference | Identifiers | Additional information |
|---|---|---|---|---|
| Peptide, recombinant protein | SrcTide (1995) | sequence from PMID:7845468, synthesized in-house | | peptide sequence: Ac-GAEEEIYGEFEAKKKG-NH2 |
| Peptide, recombinant protein | SrcTide (2014) | sequence from PMID:25164267, purchased from Synpeptide | | peptide sequence: Ac-GAEEEIYGIFGAKKKG-NH2 |
| Peptide, recombinant protein | AblTide (2014) | sequence from PMID:7845468, synthesized in-house | | peptide sequence: Ac-GAPEVIYATPGAKKKG-NH2 |
| Peptide, recombinant protein | HRAS_Y64 | sequence from PMID:35606422, purchased from Synpeptide | | peptide sequence: Ac-AGQEEYSAMRD-NH2 |
| Peptide, recombinant protein | HRAS_Y64_E63K | sequence from PMID:35606422, purchased from Synpeptide | | peptide sequence: Ac-AGQEKYSAMRD-NH2 |
| Peptide, recombinant protein | CDK13_Y716_YF | this paper, synthesized in-house | | peptide sequence: Ac-IGEGTYGQVFK-NH2 |
| Peptide, recombinant protein | CDK13_Y716_G717R_YF | this paper, synthesized in-house | | peptide sequence: Ac-IGEGTYRQVFK-NH2 |
| Peptide, recombinant protein | CDK5_Y15 | sequence from PMID:35606422, purchased from Synpeptide | | peptide sequence: Ac-IGEGTYGTVFK-NH2 |
| Peptide, recombinant protein | CDK5_Y15_G16R | sequence from PMID:35606422, purchased from Synpeptide | | peptide sequence: Ac-IGEGTYRTVFK-NH2 |
| Peptide, recombinant protein | PLCG1_Y210 | this paper, synthesized in-house | | peptide sequence: Ac-SGDITYGQFAQ-NH2 |
| Peptide, recombinant protein | PLCG1_Y210_T209N | this paper, synthesized in-house | | peptide sequence: Ac-SGDINYGQFAQ-NH2 |
| Peptide, recombinant protein | GLB1_Y294 | this paper, synthesized in-house | | peptide sequence: Ac-VASSLYDILAR-NH2 |
| Peptide, recombinant protein | GLB1_Y294_L297F | this paper, synthesized in-house | | peptide sequence: Ac-VASSLYDIFAR-NH2 |
| Peptide, recombinant protein | MISP_Y95 | this paper, synthesized in-house | | peptide sequence: Ac-EGWQVYRLGAR-NH2 |

*Appendix 1 Continued on next page*

*Appendix 1 Continued*

| Reagent type (species) or resource | Designation | Source or reference | Identifiers | Additional information |
|---|---|---|---|---|
| Peptide, recombinant protein | HLA-DPB1_Y59_F64L_YF | this paper, synthesized in-house | | peptide sequence: Ac-LERFIYNREEL-NH2 |
| Peptide, recombinant protein | PEAK1_Y797 | this paper, synthesized in-house | | peptide sequence: Ac-SVEELYAIPPD-NH2 |
| Peptide, recombinant protein | SIRPA_Y496_P491L | this paper, synthesized in-house | | peptide sequence: Ac-LFSEYASVQV-NH2 |
| Peptide, recombinant protein | HGD_Y166_F169L | this paper, synthesized in-house | | peptide sequence: Ac-GNLLIYTELGK-NH2 |
| Peptide, recombinant protein | ITGA3_Y237_YF | this paper, synthesized in-house | | peptide sequence: Ac-WDLSEYSFKDP-NH2 |
| Peptide, recombinant protein | ITGA3_Y237_S235P_YF | this paper, synthesized in-house | | peptide sequence: Ac-WDLPEYSFKDP-NH2 |
| Peptide, recombinant protein | Src Consensus (C-2S) | this paper, synthesized in-house | | peptide sequence: Ac-GPDESIYDMFPFKKKG-NH2 |
| Peptide, recombinant protein | Src Consensus (C-2P) | this paper, synthesized in-house | | peptide sequence: Ac-GPDEPIYDMFPFKKKG-NH2 |
| Peptide, recombinant protein | ACTA1_Y171_YF | this paper, synthesized in-house | | peptide sequence: Ac-QPIFEG(pY)ALPHAG-NH2 |
| Peptide, recombinant protein | ACTA1_Y171_A172G_YF | this paper, synthesized in-house | | peptide sequence: Ac-QPIFEG(pY)GLPHAG-NH2 |
| Peptide, recombinant protein | ACTB_Y240 | this paper, synthesized in-house | | peptide sequence: Ac-QSLEKS(pY)ELPDGG-NH2 |
| Peptide, recombinant protein | ACTB_Y240_P243L | this paper, synthesized in-house | | peptide sequence: Ac-QSLEKS(pY)ELLDGG-NH2 |
| Peptide, recombinant protein | CCDC39_Y593 | this paper, synthesized in-house | | peptide sequence: Ac-QRKQQL(pY)TAMEEG-NH2 |
| Peptide, recombinant protein | CLIP2_Y972 | this paper, synthesized in-house | | peptide sequence: Ac-QSDQRR(pY)SLIDRG-NH2 |

*Appendix 1 Continued on next page*

*Appendix 1 Continued*

| Reagent type (species) or resource | Designation | Source or reference | Identifiers | Additional information |
|---|---|---|---|---|
| Peptide, recombinant protein | CLIP2_Y972_R977P | this paper, synthesized in-house | | peptide sequence: Ac-QSDQRR(pY)SLIDPG-NH2 |
| Peptide, recombinant protein | CBS_Y308 | this paper, synthesized in-house | | peptide sequence: Ac-QVEGIG(pY)DFIPTG-NH2 |
| Peptide, recombinant protein | CBS_Y308_G307S | this paper, synthesized in-house | | peptide sequence: Ac-QVEGIS(pY)DFIPTG-NH2 |
| Peptide, recombinant protein | fluorescently-labeled c-Src-SH2 consensus peptide | sequence from PMID:7680959 | | peptide sequence: FITC-Ahx-GDG(pY)EEISPLLL-NH2; gift from Jeanine Amacher at Western Washignton University |
| Peptide, recombinant protein | Src Consensus (D+1 K) | this paper, synthesized in-house | | peptide sequence: Ac-GPDECIYKMFPFKKKG-NH2 |
| Peptide, recombinant protein | Src Consensus (D1AcK) | this paper, synthesized in-house | | peptide sequence: Ac-GPDECIY(AcK)MFPFKKKG-NH2 |
| Peptide, recombinant protein | Src Consensus (C-2K) | this paper, synthesized in-house | | peptide sequence: Ac-GPDEKIYDMFPFKKKG-NH2 |
| Peptide, recombinant protein | Src Consensus (C-2AcK) | this paper, synthesized in-house | | peptide sequence: Ac-GPDE(AcK)IYDMFPFKKKG-NH2 |
| Peptide, recombinant protein | Abl Consensus (A+1 K) | this paper, synthesized in-house | | peptide sequence: Ac-GPDEPIYKVPPIKKKG-NH2 |
| Peptide, recombinant protein | Abl Consensus (A+1 AcK) | this paper, synthesized in-house | | peptide sequence: Ac-GPDEPIY(AcK)VPPIKKKG-NH2 |
| Peptide, recombinant protein | Abl Consensus (I+5 K) | this paper, synthesized in-house | | peptide sequence: Ac-GPDEPIYAVPPKKKKG-NH2 |
| Peptide, recombinant protein | Abl Consensus (I+5 AcK) | this paper, synthesized in-house | | peptide sequence: Ac-GPDEPIYAVPP(AcK)KKKG-NH2 |
| Commercial assay or kit | MiSeq Reagent Kit v3 (150 cycles) | Illumina | Illumina: MS-102–3001 | |
| Commercial assay or kit | NextSeq 500 Mid-Output v2 Kit (150 cycles) | Illumina | Illumina: FC-404–2001 | |

*Appendix 1 Continued on next page*

*Appendix 1 Continued*

| Reagent type (species) or resource | Designation | Source or reference | Identifiers | Additional information |
|---|---|---|---|---|
| Commercial assay or kit | Promega QuantiFluor dsDNA Sample Kit | Promega | Promega: E2671 | |
| Commercial assay or kit | ADP Quest Assay Kit | Eurofins Discoverx | Eurofins Discoverx: 90–0071 | |
| Commercial assay or kit | Dynabeads FlowComp Flexi Kit | ThermoFisher Scientific | ThermoFisher Scientific: 11061D | |
| Chemical compound, drug | 4-carboxymethyl phenylalanine (CMF) | Millipore Sigma | Millipore Sigma: ENA423210770 | |
| Chemical compound, drug | 4-azido-L-phenylalanine (AzF) | Chem-Impex International | Chem-Impex: 06162 | |
| Chemical compound, drug | N-ε-Acetyl-L-Lysine (AcK) | MP Biomedicals | MP Biomedicals: 02150235.2 | |
| Chemical compound, drug | Click-iT sDIBO -Alexa fluor 555 | ThermoFisher | Thermo: C20021 | |
| Other | Creatine Phosphokinase from rabbit muscle | Millipore Sigma | Millipore Sigma: C3755-500UN | purified enzyme extracted from rabbit muscle |
| Software, algorithm | FLASH (version FLASH2-2.2.00) | PMID:21903629 | | https://ccb.jhu.edu/software/FLASH/ |
| Software, algorithm | Cutadapt (version 3.5) | DOI:10.14806/ej.17.1.200 | | https://cutadapt.readthedocs.io/en/stable/ |
| Software, algorithm | Python scripts for processing and analysis of adeep sequencing data | this paper (*Li et al., 2023*) | | https://github.com/nshahlab/2022_Li-et-al_peptide-display |
| Software, algorithm | Logomaker | PMID:31821414 | | https://logomaker.readthedocs.io/en/latest/index.html |

