## [Editor Report]

This paper reports an improved bacterial surface peptide display technology and its use to survey the primary sequence specificities of a broad range of tyrosine kinases and to assess the effects of naturally-occurring positional variations around sites of tyrosine phosphorylation on the efficiency of phosphorylation. The versatility of this approach was demonstrated by using expanded genetic code technology to investigate the consequences of installing post-translationally modified amino acids, such as acetyl-lysine, at positions upstream and downstream of a target tyrosine on the efficiency of phosphorylation by different tyrosine kinases. In addition, pre-phosphorylated surface peptide display libraries were exploited to interrogate the primary sequence binding specificities of SH2 phosphotyrosine-binding domains.

---

## [Decision Letter]

**Decision letter after peer review:**

Thank you for sending your article entitled "High-throughput profiling of sequence recognition by tyrosine kinases and SH2 domains using bacterial peptide display" for peer review at *eLife*. Your article is being evaluated by 4 peer reviewers, including Tony Hunter as the Reviewing Editor and Reviewer #1, and the evaluation is being overseen by Jonathan Cooper as the Senior Editor.

The reviewers were impressed by the new sequence specificities you have obtained for additional tyrosine kinases and SH2 domains by using your improved bacterial peptide surface display technology. However, in their opinion, the technical improvements you describe are not a significant enough advance to warrant publication of the paper as it stands. The reviewers indicate that validation of the biological significance of at least one of your novel findings is required to establish that the specificities obtained by your approach will be useful to the community. For this reason, prior to requesting you to submit a revised version, we ask you to submit a written plan outlining additional experiments that you could do within a reasonable time frame to validate one of your new findings. Each of the reviewers has suggestions for the sort of experiments you might be able to do.

*Reviewer #1 (Recommendations for the authors):*

Here, the authors report an improved version of the X5-Tyr-X5 peptide bacterial surface display technology, which they had developed previously to determine the primary sequence specificities of the LCK and ZAP70 tyrosine kinases (TKs) and the EGF receptor. Taking advantage of the new protocol they defined the primary sequence specificities of five additional TKs: the c-Src, c-Abl, and Fer nonreceptor TKs, and the EPHB1 and EPHB2 RTKs. Consensus peptide substrates were synthesized for the five TKs using the most favorable amino acid at every position, and their kinetic properties and TK selectivity were compared with those of the SrcTide and AblTide substrate peptides. The consensus peptides had reasonable (10-200 μM Km) affinities and were relatively selective substrates for the c-Src, cAbl, and Fer TKs, respectively. However, although consensus sequences were deduced for EPHB1 and EPHB2, these kinases showed relatively little sequence selectivity. Next, they used their specificity data to predict the consequences of known naturally occurring single amino acid sequence variations – either disease-associated mutations or polymorphisms – in the five residues on either side of c-Src phosphosites on phosphorylation, and compared these predictions with actual experimental data, showing that they could predict relative rates of phosphorylation of the variant peptides with reasonable accuracy. Based on this success, they built a 10,000 member MYC-tagged variant expression library (pTyr-Var), which excluded Tyr in the X5 flanking residues, and tested the ability of the c-Src, Fyn, and Hck SFKs, c-Abl, Fer, JAK2, and AncSZ, an engineered homologue of the SYK/ZAP-70 family members, and five RTKs – EPHB1, EPHB2, *FGFR1*, FGFR3, and MERTK. They found that the different TKs exhibited distinct patterns of sensitivity to sequence variation at each position around the Tyr, consistent with individual sequence preferences. As an example, the variant R982C RET peptide was strongly preferred by several TKs compared to the Tyr981 reference peptide. They also showed that there were sequence context-dependent effects of mutations proximal to a phosphosite, which were not predicted based on the X5-Y-X5 library results. In addition, they exploited expanded genetic code technology to incorporate CMF, a pTyr analogue, or acetyl-lysine (AcK) residues randomly in the X5-Y-X5 library. They found that unlike Lys, AcK was not only tolerated but preferred in some positions for c-Src phosphorylation, whereas CMF could not replace Phe at preferred positions in c-Src peptide substrates. The authors had previously used pre-phosphorylated peptide surface displays to survey the binding specificity of the GRB2 SH2 domain, and here they extended this to two additional SH2 domains, SHP2-C and c-Src. For this purpose, they pre-phosphorylated the X5-Y-X5 library with a mix of c-Src, c-Abl, AncSZ, and EPHB1 TKs, and then used biotinylated SH2 constructs, tandemized to increase binding avidity, to screen the c-Src, SHP2-C and GRB2 SH2 domains. With the exception of SHP2-C, they found sequence preferences largely concordant with those reported using other approaches. By screening the pTyr-Var library for SH2 binding they also found natural variant phosphosite sequences that exhibited gain of function for SH2 binding that could be of functional significance.

This extension of the authors' previous studies with bacterial surface peptide display technology has provided some additional insights into the primary sequence specificities of the large tyrosine kinase family by examining a broader range of TKs, and by checking the consequences of natural positional variations around sites of Tyr phosphorylation on the efficiency of phosphorylation. They demonstrated the versatility of this approach by using expanded genetic code technology to investigate the consequences of installing post-translationally modified amino acids at specific positions upstream and downstream of the target Tyr on the efficiency of phosphorylation by different TKs. They also exploited pre-phosphorylated peptide display libraries to interrogate the primary sequence specificities of two additional SH2 domains.

One advantage of the surface display method is that in principle it provides individual sequences that are preferred substrates for a TK, but in practice this information is not used, and preferred residues at each position are obtained. The new TK specificities they have determined reveal some interesting motifs, and the use of this method to define the effects of sequence variants in the vicinity of target phosphosites on their phosphorylation or SH2/PTB domain binding will be valuable in predicting possible functional consequences of such variants, for instance in disease. The use of the display method to detect the consequences of posttranslational modifications of amino acids in the vicinity of a phosphosite is also an advance, but it is limited by the availability of cell permeant, stable unnatural amino acid analogues and cognate evolved tRNA charging enzymes

Overall, given their prior publications using the original version of this technology, this paper is a relatively modest technical advance, but the significant amount of new TK and SH2 specificity information they have obtained will no doubt be useful to aficionados of tyrosine phosphorylation signaling systems. Even though this paper was submitted for the Tools and Resources category, it would be strengthened by inclusion of a follow-up experimental analysis of at least one or two instances where a novel specificity was observed to demonstrate its biological relevance.

Points:

1. The potential issues with including Cys in the peptide libraries were not discussed. For instance, a Cys residue in a peptide may be partially oxidized on the surface because of its exposure to oxygen, or, because the Cys is unpaired, disulfide bonds may form between adjacent peptide molecules on the surface. In addition, having multiple Tyr in addition to the central Tyr and using high phosphorylation stoichiometries means that the extra tyrosine in the peptide may be the preferred target.

2. As the authors showed, the method will be useful for studying the influence of neighboring modified amino acids on TK phosphosite selectivity. Indeed, it is already known that pTyr can serve as a positive determinant for TK phosphorylation. Presumably, it was for this reason that the authors tested whether incorporation of CMF, a pTyr mimic, at different positions affected the ability of the c-Src kinase to phosphorylate preferrred sequences. However, CMF is not a very good pTyr mimic, and if the authors wanted to determine possible roles for pTyr itself they should have used one of the recently described methods for incorporating pTyr as an unnatural amino acid into proteins expressed in bacteria (e.g., PMID: 28604693; PMID: 28604697).

3. From a methods perspective, some of the recombinant catalytic domains were preactivated by incubation with ATP, presumably leading to autophosphorylation. The sites phosphorylated on the activation loop of TKs can affect primary sequence specificity. Electrospray MS was used to show these preparations were multiply phosphorylated, but which autophosphorylation sites were occupied in the protein that was used to phosphorylate the library were not determined.

4. It is not clear why the surface display method should be "tuned to select a high Kcat". Does the *E. coli* DH5a strain used here secrete a nonspecific phosphatase that could act on the phosphorylated peptides; the assay buffer included orthovanadate, but it is not clear whether this would inhibit such a generic phosphatase?

5. The authors' c-Src results showed a strong preference for a +1 Asp/Glu/Ser in addition to the +1 Gly previously reported using oriented peptide libraries. However, in the end the improvements in kinetic parameters for their Src and Abl consensus peptides were relatively modest compared to prior published examples.

6. Figure 2B and Figure 2S1: Src, Abl, EPHB1 and EPHB2 all showed a preference in their display consensus for Pro at +4. While a preference for Pro at +4 is observed in natural Src and Abl substrates, one wonders whether this is a hidden constraint/bias of the method. In addition, the +2 – +4 WWW motif preferred by Fer is curious. Are there any reported (Fer) TK sites with this WWW motif, and does the closely related Fes TK also exhibit this preference? No Trp residues are found in the PhosphoSitePlus motif logo for Fer. Do these Trp residues contribute to the low μM Km for the Fer consensus peptide? Can the WWW peptide be modeled into the active site of the FES TK catalytic domain? Finally, in this regard, it would help the reader if this panel included -5 to +5 numbering under the alignment.

7. The SRC consensus has a preferred Cys at -2, which was not discussed – do the authors know whether this Cys was oxidized on the bacterial surface. If so, it could have provided a partial negative charge and serve a similar purpose to the -2 Glu in SrcTide. How important is the -2 Cys in the synthetic peptide substrates?

8. Figure 2C: The authors' data showed that EPHB1 displayed little selectivity across the consensus peptides and did not even prefer its own cognate consensus sequence. Moreover, although they did not comment on this, the same seems to be true for EPHB2. The basis for this lack of selectivity needs fuller discussion.

9. Figure 4F: The authors did not explain why the inclusion of Cys at +1 in the Ret Y981 peptide made it a better substrate for several of the 13 tested TKs.

10. While the authors showed that unnatural amino acids incorporation is compatible with the surface display method, only those modified amino acids for which a cell-permeant modified amino acid and a cognate tRNA charging enzyme have been developed can readily be used. In contrast, any modified amino acid can be used in the position-oriented peptide display approach.

*Reviewer #2 (Recommendations for the authors):*

This is a well-designed study addressing important biological issues – what are the specificities of the tyrosine kinases and SH2 domains which work in tandem in cell signal transduction and deregulation of the TK-SH2 signaling axis is associated with a host of diseases, notably cancer. Although these issues have been investigated in previous studies, the current study established a platform that is complementary to previous ones (eg., those based on synthetic peptide libraries) and potentially more quantitative. The application of this approach to known Tyr/pTyr sites with mutations in the flanking residues is novel and provides insights into the biochemical and functional consequences of the mutations. Although the work is comprehensive, and the data presented are of high-quality and supportive of the conclusions in general, there are a few concerns as enumerated below.

1. The authors appear to focus on justifying how their platform identifies the same specificity profiles as previous methods when it is more important to highlight the differences and discuss what they mean. While it is challenging to verify whether the specificity profiles obtained from the current study are closer to the physiological specificity of the TKs/SH2s, it'd be helpful to use the specificity information to predict in vivo substrates and to find out how many known substrates can be identified and how many are missed. After all, the specificity profile is valuable only when it can predict in vivo targets and the effect of mutations on TK/SH2-target interaction (the authors are to be commended for doing a decent job on the latter).

2. A systematic comparison of the context-independent specificity (obtained from the X5-Y-X5 library) with context-dependent effect (obtained from the pTyr-var library) is missing. It would be important to systematically compare the specificity profiles obtained from the two libraries and identify the common and distinct features (and explain why).

3. The platform seems to work better for certain TKs (eg., Src, Abl) than others (eg., EphB1/B2). Is this due to the distinct specificity of these kinases or differences in enzymatic activity? In this regard, how is the activity of the TKs benchmarked?

4. By the same token, what's the affinity of an SH2 domain used in the study for its cognate ligand? It should also be discussed/justified in more detail why the SHP2-C SH2 domain was picked when the tandem SH2 domains in SHP2 often work together in ligand recognition.

5. The manuscript would be significantly improved if one or more novel sites/substrates predicted using the specificity data are validated on the peptide and protein level.

6. The heatmaps (eg., Figure 5-SI5) look similar at a glance. Is there another way to present the data and show the differences in specificity in a more conspicuous manner? Bar graph – which is more tedious- may be a better choice especially for important positions. Grouping residues based on their physiochemical properties may also simplify the profile pattern and make it easier to read and understand.

7. The inclusion of Tyr in the X positions of the X5-Y-X5 library may obscure the specificity patter as Tyr at any position may be phosphorylated. I'd suggest repeating a couple of TKs using a library that contains only a central Tyr.

8. The AA read counts (eg. Figure S1) varied widely, why? Does this have an effect on the screening?

9. Figure 5-SI1, provide correlation coefficient; why is the apparent correlation for Abl poorer compared to EphB1 when the former appears to have a more defined specificity pattern?

10. Figure 6 – SI6 – The orthogonal specificity pattern for the CTK kinase and SH2 domains seems at odds with the established notion that the CTK kinase and SH2 domain specificities are related., eg, the pTyr sites created by the kinase are preferably bound by its own SH2 domain or closely related SH2 domains (Songyang et al. Nature 1995; 373:536-9). This needs to be discussed.

*Reviewer #3 (Recommendations for the authors):*

This manuscript reports modifications to previously published methods in which the substrate specificity of tyrosine kinases and binding specificity of phosphotyrosine interaction domains is analyzed using a bacterial surface display coupled to next-generation sequencing. This report modifies these methods in that phosphorylated bacteria are selected by magnetic bead immunoprecipitation rather than fluorescence-activated cell sorting (FACS). This allows for larger libraries of higher complexity to be used and for parallel processing to increase throughput. The strength of this manuscript lies in the use of a cutting edge technique offering substantial benefit over other methods. Its main weakness is that the major capabilities of the method are similar to those of the FACS-based approach, and the conclusions drawn here were also reached across the series of papers originally reporting that method. For example, the previous papers reported good correlations between enrichment scores and phosphorylation rates measured *in vitro*, the capacity to infer changes in phosphorylation rate from single amino acid substitutions, and the context-dependent impact of those substitutions; the orthogonal nature of SH2 binding and kinase specificity observed here has been seen in other studies using different methods. Given the precedent, one would have hoped to see the method applied in a way that new insight is gained to the nature of Tyr kinase specificity in general, or to how a particular kinase signals. One modification to the protocol that had not been previously reported was the use of an amber suppression-based orthogonal translation system allowing incorporation of non-natural or modified amino acids. Libraries incorporating carboxymethyl-Phe (a pTyr mimic) and acetylated lysine residues were generated and screened with the kinase c-Src. This is an interesting extension of the technique, and one could imagine bringing in other known PTMs such as methylated Lys or Arg. However, there is a potential limitation in that each non-native residue requires a separate screen to be performed. Overall, this manuscript employs state-of-the-art screening methods, but its impact is tempered by literature precedent.

Manuscript presentation:

1. The introduction provides an overview of existing methods for determining kinase specificity. However, the authors neglect to mention a couple of approaches that arguably best rival methods employed in the current manuscript from the standpoint of identifying context-dependent selectivity. These include MS-based approaches that use proteome-derived peptide libraries, from either protease digests of cell lysates or genetically encoded libraries (the cited Barber et al. paper is mischaracterized as performing analysis on cell lysates, and these papers were not cited: Kettenbach, et al. PMID: 22633412, Douglass et al. PMID: 22723110, Imamura et al. PMID: 24869485 and Xue, et al. PMID: 22451900). In addition, an approach using yeast surface display in which Tyr kinases are targeted to the secretory pathway deserves mention (Taft et al. PMID: 31339688). The authors could also consider less extensive referencing of historical methods from the 1990s that are no longer used.

Data availability:

2. I could not find the deep sequencing data for kinase selections of the X5-Y-X5 library. Did they apply a cutoff for the number of read counts to be included in their analysis?

Technical points:

3. Two versions of the X5-Y-X5 library were made – one with a strep tag and one with a myc tag. Were both subjected to quality control sequencing as shown in Figure 1 supp 1, and which is shown in the table? Were both of them used for screens?

4. It is not clear that the authors estimated the representation of the X5-Y-X5 library other than from the standpoint of pooled representation of each residue at each position ("1-10 million unique peptide sequences" were observed by sequencing). How many transformants were recovered during library cloning, and how does this compare to the number of sequencing reads used for quality control?

5. Selection of Tyr residues from the X5-Y-X5 library are likely due to "off target" phosphorylation of the non-central residue. This could muddy the specificity analysis as there would contribution of other selected residues at the "wrong" positions. For the Tyr-Var library, the authors avoided this issue by including Tyr to Phe substitutions. Do any of the results for X5-Y-X5 change if they exclude all peptides with Tyr at a non-central position?

6. Based on the concentration bacterial cells used in the kinase reactions, it seems likely that the authors are observing single turnover rather than multiple turnover kinetics. This could explain some of the differences between the results described here and prior analyses with the same kinases. Can the authors estimate the effective substrate concentration in their experiments? Can they comment on the potential impact of single vs multiple turnover kinetics on their results?

7. For Tyr kinases that were analyzed with both libraries, how do the specificity matrices derived from the Tyr-Var data (Figure 4 – Figure supp 5) compare to those from the X5-Y-X5 data (Figure 1B and Figure 2A)?

8. The observation that there is context dependence – the impact of a given amino acid substitution differs depending on the surrounding sequence – is not surprising given the author's previous work. However, the specific example shown here should be interpreted with caution, as undersampling may impact what is construed to be the "average" signal in the X5-Y-X5 library. It would be most convincing if there were examples where the impact of a given amino acid substitution were observed on two different peptide sequences, i.e. where enrichment scores for both the "WT" and variant sequence could be calculated. One would also want to see some verification by *in vitro* kinase assay that the same substitution caused different effects on reaction rates in the context of two different peptides.

*Reviewer #4 (Recommendations for the authors):*

In work by Li and colleagues, they utilized phage display technology to characterize phosphorylation site motifs of tyrosine kinases and pY binding motifs of SH2 domains. Building on previously published work by the corresponding author, they have significantly improved and broadened the application of their platform. The experimental approach here involves the external display of peptides on the surfaces of bacteria. These peptides contain tyrosines surrounded by defined amino acid sequences where each bacterium encodes a single sequence. The cells are subjected to *in vitro* kinase phosphorylation reactions, and the phosphorylated population is examined by deep sequencing to quantify the enrichment of amino acid residues and infer phosphorylation site motifs. In this manuscript, the authors have improved the sampling rate of their platform from <5000 to 1-10 million unique peptide sequences analyzed per experiment, made it more accessible (no longer requiring FACS instrumentation), combined it with amber codon suppression methods to expand the list of amino acids that can be incorporated, and repurposed it to profile SH2 binding.

Overall, this is a very nice extension of the corresponding author's previous work and is likely to make helpful contributions to the tyrosine kinase signaling field. Their manuscript surveys multiple applications of their phage display platform and provides reliable supporting experimental data. Additionally, the authors have identified 50-400 disease-associated mutation variants proximal to tyrosine phosphorylation sites that potentially alter phosphorylation by the kinases examined in this study, and 50-300 that potentially alter binding to the SH2 domains examined, which are all provided as resources. Along the way, they made interesting observations (e.g., mutations on RET and PTEN predicted to enhance tyrosine phosphorylation). Lastly, the authors utilized Amber codon suppression to incorporate non-canonical and PTM amino acids into their displayed peptides. They then showed that the SRC kinase prefers to phosphorylate tyrosine substrates nearby acetylated lysines versus unmodified lysines. This was perhaps to be expected, considering that SRC and many other tyrosine kinases generally disfavor positively charged amino acids, but it also highlights the interesting possibility of this form of PTM crosstalk between metabolism and growth factor signaling.

1. Concerning the peptide design in their random library (sequence: X5-Y-X5, where X = 20 natural amino acids falling within their 'NNS' codon constraint), neighboring tyrosines (TAC codons) theoretically account for ~3% (1/32) of residues at all random positions, meaning that as much as 30% of the peptide pool contains at least two potential tyrosine phosphoacceptors per peptide. For the heatmap motifs presented in Figure 1, Figure 2, Figure 4—figure supplement 5, and Figure 7, we can observe strong enrichment of tyrosines at multiple positions, especially for JAK2 and the RTKs, and it is not certain whether this is due to positional selection (that facilitates phosphorylation of the tyrosine at position zero) or direct phosphorylation of the neighboring tyrosine. This double phosphorylation effect may limit interpretation of the motif heatmaps presented in the figures that assume phosphorylation occurs only at the tyrosine at position zero. Have the authors considered generating separate heatmaps that omit the subset of enriched peptides containing two or more tyrosines? That might reduce background and further improve the quality of the data.

2. Figure 6B: Given that the authors have demonstrated essentially complete phosphorylation of the displayed peptides by their kinase cocktail in Figure 6—figure supplement 2, it seems more appropriate to replace the label above the heatmaps in 6B "position relative to tyrosine" with "position relative to phosphotyrosine". Moreover, this indicates that they have probably phosphorylated most of the neighboring tyrosines in the displayed peptides containing two or more tyrosines. How would the heatmap motifs for the SH2 domains appear if the authors excluded the subset of enriched peptides containing two or more (likely to be phosphorylated) tyrosines?

3. The authors have not included a single sequence logo to represent their phosphorylation site motifs or pY binding motifs. Is there a specific reason for this? Researchers less familiar with these approaches generally have an easier time interpreting sequence logos than heatmaps.

4. An advantage of the authors' platform over alternative approaches is its potential to decipher pairwise interactions between amino acids on substrate peptides. The authors report that, depending on the kinase, 5-15% of all significant mutations in the pTyr-Var screen (which are pairwise comparisons of single substitutions in fixed sequences) had the opposite effect of predictions from the randomized library (ensembles). And indeed, the correlation plots in Figure 5 —figure supplement 1 show quite a bit of scattering and hence negatively correlated sites between their predictions and their results. The authors explored one example of this for SRC, which preferred proline over serine at -2 in the randomized library, yet it more efficiently phosphorylated a site containing -2 serine versus proline in the context of the sequence XEYSFK, where X is the substituted position. Interestingly, this is the opposite effect of what has been previously published and referred to by the authors for EGFR, where -2 proline was preferred over serine in the context of -1 acidic residues (Cantor et al. 2018), indicating that these pairwise rules may not only be unique to specific kinase groups but mutually exclusive between different groups. Are the authors able to identify kinase-specific trends from their datasets for the pairwise selection of amino acids at positions -1 and -2 that negatively correlate with predictions?

5. Cannot access the code used to process and analyze the deep sequence data in the GitHub repository: (https://github.com/nshahlab/2022_Li-etal_peptide-display)

6. Based on the format of this journal, this work seems appropriate as a "Research Advance."

---

## [Author Response]

Reviewer #1 (Recommendations for the authors):Points:1. The potential issues with including Cys in the peptide libraries were not discussed. For instance, a Cys residue in a peptide may be partially oxidized on the surface because of its exposure to oxygen, or, because the Cys is unpaired, disulfide bonds may form between adjacent peptide molecules on the surface. In addition, having multiple Tyr in addition to the central Tyr and using high phosphorylation stoichiometries means that the extra tyrosine in the peptide may be the preferred target.

TCEP is used in the peptide display assay in order to avoid the potential issues discussed above. We agree that having multiple tyrosines in addition to the central Tyr in our peptide display screen is a valid concern. To address this, peptides in which non-central tyrosine residues were mutated to phenylalanine were also included in the pTyr-Var library (labeled “YF” sequences). Furthermore, in the data analysis pipeline for any library used with our platform, sequences containing cysteine residues or multiple tyrosine residues can trivially be filtered out prior to calculating enrichment values. In the revised manuscript, we show for the X_5_-Y-X_5_ libraries that this filtering of multi-Tyr sequences has no impact on our specificity maps. We have added the following text to the manuscript:

“Notably, our library includes peptides containing Cys residues and non-central Tyr residues, both of which are often excluded from tyrosine kinase specificity screens to avoid oxidation-related artifacts and challenges in interpreting signal from multi-Tyr sequences (Deng et al. 2014). These sequences can be filtered during data analysis, if needed, although they did not pose significant issues in our studies.”

2. As the authors showed, the method will be useful for studying the influence of neighboring modified amino acids on TK phosphosite selectivity. Indeed, it is already known that pTyr can serve as a positive determinant for TK phosphorylation. Presumably, it was for this reason that the authors tested whether incorporation of CMF, a pTyr mimic, at different positions affected the ability of the c-Src kinase to phosphorylate preferrred sequences. However, CMF is not a very good pTyr mimic, and if the authors wanted to determine possible roles for pTyr itself they should have used one of the recently described methods for incorporating pTyr as an unnatural amino acid into proteins expressed in bacteria (e.g., PMID: 28604693; PMID: 28604697).

In this study, we did not use CMF for the purposes of studying phospho-priming. CMF was used to show that non-canonical amino acids could be incorporated in our screens. We agree that if we were to study the possible roles of phosphotyrosine on substrate specificity, the most ideal amino acid would be phosphotyrosine itself. To avoid confusion, we no longer refer to CMF as a “phosphotyrosine analog” in the main text. Furthermore, the studies mentioned by the reviewer would definitely be a useful starting point for incorporating phosphotyrosine (or its analogs) into our libraries. The method in PMID 28604697 may prove challenging, as it requires a chemical transformation that is not likely to be tolerated by *E. coli*. Our preliminary efforts with the method described in PMID 28604693 have not been successful and require further optimization beyond the scope of this study.

3. From a methods perspective, some of the recombinant catalytic domains were preactivated by incubation with ATP, presumably leading to autophosphorylation. The sites phosphorylated on the activation loop of TKs can affect primary sequence specificity. Electrospray MS was used to show these preparations were multiply phosphorylated, but which autophosphorylation sites were occupied in the protein that was used to phosphorylate the library were not determined.

The determination of the autophosphorylation sites was not a primary focus for us because we did not think the phosphorylation of the tyrosine kinases would significantly alter substrate recognition. We were not aware of papers showing that tyrosine kinase activation loop phosphorylation alters sequence specificity at the peptide level. Our primary objective was to have kinase domains with sufficient activity for measurements (which often depends on their activation loop phosphorylation status). For the Src-family kinases, under the screening conditions used, the kinase domains rapidly autophosphorylate. This is not likely to be true for c-Abl or AncSZ, but these kinases showed sufficient activity without pre-activation.

4. It is not clear why the surface display method should be "tuned to select a high Kcat". Does the *E. coli* DH5a strain used here secrete a nonspecific phosphatase that could act on the phosphorylated peptides; the assay buffer included orthovanadate, but it is not clear whether this would inhibit such a generic phosphatase?

In this statement, we were pointing out that for some kinases (e.g. c-Src but not c-Abl), our screens yielded consensus sequences with higher k_cat_, but weaker K_M_ values than other methods. We simply point this out as an interesting observation for c-Src, but the precise reason for this is unclear. (We discuss this briefly in another response about single-turnover kinetics, below.) Furthermore, we note that the *E. coli* DH5a strain does not secrete a nonspecific phosphatase. We use sodium orthovanadate to inhibit any residual tyrosine phosphatase, YopH, from co-expression with the tyrosine kinase. While our purification methods remove any detectable amounts of YopH, orthovanadate is added as a precaution. This is a standard practice for many tyrosine kinase activity assays. See some of the following examples: (PMIDs 27700984, 29547119, 25699547, 32479050, 22928736).

5. The authors' c-Src results showed a strong preference for a +1 Asp/Glu/Ser in addition to the +1 Gly previously reported using oriented peptide libraries. However, in the end the improvements in kinetic parameters for their Src and Abl consensus peptides were relatively modest compared to prior published examples.

Based on our screening data, we expect no major difference between +1 Gly and +1 Asp/Glu/Ser in the specificity of c-Src. However, we do notice there is a slight preference for +1 Gly in c-Src when we use the PY20 antibody for labeling (see Author response image 1, which shows the enrichment for each amino acid at the -1 and +1 position). This difference in the antibody used may account for why we see less of an exclusive preference for +1 Gly. Regardless, we have characterized sequences with a change in the +1 position, and the effects on kinetic activity are marginal (Table 1). We also note this broader +1 preference has been observed for c-Src previously (PMID: 29547119). These experiments collectively suggest that the +1 Gly preference for c-Src is not as exclusive as previously observed using oriented peptide libraries.

**Author response image 1. sa2fig1:** 

6. Figure 2B and Figure 2S1: Src, Abl, EPHB1 and EPHB2 all showed a preference in their display consensus for Pro at +4. While a preference for Pro at +4 is observed in natural Src and Abl substrates, one wonders whether this is a hidden constraint/bias of the method. In addition, the +2 – +4 WWW motif preferred by Fer is curious. Are there any reported (Fer) TK sites with this WWW motif, and does the closely related Fes TK also exhibit this preference? No Trp residues are found in the PhosphoSitePlus motif logo for Fer. Do these Trp residues contribute to the low μM Km for the Fer consensus peptide? Can the WWW peptide be modeled into the active site of the FES TK catalytic domain? Finally, in this regard, it would help the reader if this panel included -5 to +5 numbering under the alignment.

The observation about the +4 Pro is very interesting. We do not know if this is a hidden bias/constraint of the method, but we do note that Pro is not the most preferred amino acid at the +4 position for every kinase tested (see Fer in the X_5_-Y-X_5_ library screens, and this can be seen for other kinases from the pTyr-Var screens). To our knowledge, the +2 to +4 WWW motif preferred by Fer has not been reported elsewhere, nor does it appear to be a feature of the few reported natural Fer substrates. The tryptophan residues seem to be contributing to high binding affinity (low K_M_), but this might manifest in two different ways that warrant a follow-up study: (1) For Fer, specifically, there is very strong +2 Trp enrichment, and this is also seen in the pTyr-Var library, where +2 Trp sequences are uniquely enriched for Fer. (2) It appears that downstream Trp residues show moderate enrichment for many of the kinases tested against both libraries, particularly at +3 and +4 positions. While this could be an artifact of the screens, indirect evidence suggests that these Trp residues actually contribute to substrate binding. For example, when the Fer consensus peptide is measured against c-Src, we observe a low K_M_ value (12 μM), similar to that of Fer (8 μM). This is much tighter than the Src consensus peptide (196 μM). However, the k_cat_ value for the Fer consensus peptide with c-Src is much lower than that of the Src consensus peptide (0.74 s^-1^ vs 4.9 s^-1^). Additionally, at low substrate concentrations, the Fer consensus peptide appears to be the preferred substrate for EPHB1, of the consensus peptides tested. These observations point to a potential role for Trp residues in enhancing kinase-substrate interactions in a productive way.

7. The SRC consensus has a preferred Cys at -2, which was not discussed – do the authors know whether this Cys was oxidized on the bacterial surface. If so, it could have provided a partial negative charge and serve a similar purpose to the -2 Glu in SrcTide. How important is the -2 Cys in the synthetic peptide substrates?

We expect that the cysteine is predominately reduced in our screening conditions due to the addition of TCEP. In regards to the importance of the -2 cysteine, we have synthesized variants of the Src consensus peptide in which the cysteine was replaced with an aspartate and did not see a substantial effect on the catalytic activity. Additionally, with the recent revisions, we measured the activity of the Src consensus peptide with a -2 proline using the RP-HPLC assay and observed activity comparable to that seen for the original Src consensus. Therefore, we do not think the -2 cysteine plays a critical role for c-Src, but we also do not think that its enrichment is due to oxidation.

8. Figure 2C: The authors' data showed that EPHB1 displayed little selectivity across the consensus peptides and did not even prefer its own cognate consensus sequence. Moreover, although they did not comment on this, the same seems to be true for EPHB2. The basis for this lack of selectivity needs fuller discussion.

It was surprising to us that both EPHB1 and EPHB2 displayed little selectivity across the consensus peptides. It is unclear why this is the case for EPHB1, however, upon closer inspection, we find that EPHB2 actually does show some selectivity. This was difficult to see in our original figures, where data for every kinase was displayed on an absolute scale. We have reformatted those graphs so that the data for each kinase is normalized to its own consensus peptide. Now, we can see that EPHB2 actually phosphorylates its own consensus, and the c-Abl consensus, preferentially over the other three peptides. Notably, the c-Abl and EPHB2 peptides are very similar, as noted in the main text.

We also wondered if our poor consensus designs for EPHB1/2 might reflect some unfavorable coupling between the most enriched amino acids in the position-weighted scoring matrices. To test this, we looked at the EPHB2 consensus sequence and calculated the enrichment scores of every possible residue pair in that sequence from data generated in the X_5_-Y-X_5_ screen. Based on this analysis, we observed that the -3 Glu sequences were enriched overall, but were distinctly depleted in the context of a +1 Glu. Thus, we modified the -3 position in EPHB2 sequence to an apparently more favorable residue in the +1 Glu context, a tryptophan. Measurement of the activity of EPHB2 against this peptide showed an enhancement of activity with the -3 Glu to Trp substitution. This suggests that the combination of the most favorable amino acids in each position does not always yield the best substrate. Finally, although this is an enticing approach to sequence design, we hope that the reviewers appreciate that we are still developing this idea and an in-depth exposition of this approach is out of the scope of this manuscript.

9. Figure 4F: The authors did not explain why the inclusion of Cys at +1 in the Ret Y981 peptide made it a better substrate for several of the 13 tested TKs.

The mutational effect is not true for every tyrosine kinase, which is why we think that it is a real kinase-specific effect and not an artifact of the assay. Furthermore, this mutational effect is conserved for Src, *FGFR1*, FGFR3, and MERTK in another peptide: CRYAA_Y48, and Src and MERTK in the peptide U2SURP_Y634. This might be due to the removal of an unfavorable positive charge from the +1 Arg, coupled with the other favorable sequence features in the resulting peptide. It is also possible that other mutations to +1 Arg, such as Ser, could show the same enhancement, but these other substitutions are not represented in our pTyr-Var library. Due to the addition of new data from revision experiments, we have moved the Ret Y981 panel to a figure supplement.

**Author response image 3. sa2fig3:** 

10. While the authors showed that unnatural amino acids incorporation is compatible with the surface display method, only those modified amino acids for which a cell-permeant modified amino acid and a cognate tRNA charging enzyme have been developed can readily be used. In contrast, any modified amino acid can be used in the position-oriented peptide display approach.

We agree that this is a limitation to the incorporation of non-canonical amino acids in the peptide display platform.

Reviewer #2 (Recommendations for the authors):1. The authors appear to focus on justifying how their platform identifies the same specificity profiles as previous methods when it is more important to highlight the differences and discuss what they mean. While it is challenging to verify whether the specificity profiles obtained from the current study are closer to the physiological specificity of the TKs/SH2s, it'd be helpful to use the specificity information to predict in vivo substrates and to find out how many known substrates can be identified and how many are missed. After all, the specificity profile is valuable only when it can predict in vivo targets and the effect of mutations on TK/SH2-target interaction (the authors are to be commended for doing a decent job on the latter).

To address this, we compared a curated list of reported kinase-substrate pairs from the PhosphositePlus database with our screening results. Although there is not a lot of overlap between our library and the substrates reported in this curated list, we were able to compare dozens of sequences for c-Src, Abl, and Fyn. Based on this analysis, we find that approximately ~30-40% of the reported substrates are efficiently phosphorylated in our screen (cSrc: 26/79, c-Abl: 8/21, Fyn: 6/17). This disparity is not surprising because we know there are other mechanisms for gaining kinase specificity and the curated list may not accurately represent bona-fide substrates for each kinase. We have added this discussion to the main text and added this annotated list to Figure 4-source data 1.

2. A systematic comparison of the context-independent specificity (obtained from the X5-Y-X5 library) with context-dependent effect (obtained from the pTyr-var library) is missing. It would be important to systematically compare the specificity profiles obtained from the two libraries and identify the common and distinct features (and explain why).

The position-specific amino acid preferences obtained using both libraries are very similar, as shown in Author response image 4. Importantly, those residues that are significantly enriched or depleted in one library show the same effect in the other library, and there are generally very few outlier features for the five kinases tested. While we can observe specific cases where amino acid substitutions have a context-dependent effect, as shown in Figure 5 and discussed in the associated text, the pTyr-Var library does not sample a large enough number of these mutations to extract specific rules. We can envision future experiments that focus on a small number of cognate sequences for each kinase, where libraries of single, double, and triple mutant peptides could be screened to dissect the rules for sequence context dependence.

**Author response image 4. sa2fig4:** 

3. The platform seems to work better for certain TKs (eg., Src, Abl) than others (eg., EphB1/B2). Is this due to the distinct specificity of these kinases or differences in enzymatic activity? In this regard, how is the activity of the TKs benchmarked?

We did not base the success of our platform for each kinase on whether we were able to obtain good selectivity for its consensus sequence. As mentioned earlier, we could not determine the best consensus sequences for EPHB1 and EPHB2 potentially due to unfavorable coupling between amino acid residues in those sequences. We believe our platform worked just as well for EPHB1/2 as the other TKs because we obtained a distinct specificity profile for EPHB1/2 which showed concordance between the two libraries tested. This is further supported by the fact that the two enzymes have very similar specificity (as expected for close paralogs), and that these enzymes showed distinct specificity when compared with other families of TKs. Finally, the activity of each TK was benchmarked by monitoring library phosphorylation rates using flow cytometry under a standardized set of conditions. We adjusted the concentration of each tyrosine kinase so that the phosphorylation levels of the libraries were similar to what we had achieved with c-Src. We chose c-Src as a reference point because much of the methods development and validation was done using c-Src.

4. By the same token, what's the affinity of an SH2 domain used in the study for its cognate ligand? It should also be discussed/justified in more detail why the SHP2-C SH2 domain was picked when the tandem SH2 domains in SHP2 often work together in ligand recognition.

The affinity of an SH2 domain for one of its cognate ligands is usually in the nM range, however this value can vary from single-digit to triple-digit nM. For c-Src, for which validation experiments were shown in Figure 6, the “cognate” ligand that was fluorescently labeled for competition binding assays had a K_D_ of 160 nM. For SHP2, while we acknowledge that the tandem SH2 domains do work together in ligand recognition, there is a benefit to studying the specificity of the SH2 domains independently. There is evidence that the C-SH2 domain of SHP2 can act independently of the N-SH2 domain. For example, the C-SH2 domain of SHP2 can bind one phosphosite on PD1 with high affinity (13 nM), driving SHP2 localization to PD1, whereas the N-SH2 domain of SHP2 binds ligands (e.g. other sites on PD1) with weaker affinity (2 μM) to drive phosphatase activation (see PMID: 32064351). This is in contrast to the more tightly-coupled tandem SH2 domains, such as those in ZAP-70 and Syk, where there is more interdependence. A follow up study to further disentangle the functional importance of SHP2 N- and C-SH2 specificity is currently underway.

5. The manuscript would be significantly improved if one or more novel sites/substrates predicted using the specificity data are validated on the peptide and protein level.

We agree with the reviewer’s comment and we attempted to validate the phosphorylation of a near-full-length version of SHP2 by c-Src, Fyn, and *FGFR1*. In this context, we also tested whether the mutational effects of D61V and D61N in our screen could be observed in the context of the full length protein. We have included these experiments in our manuscript (Figure 4F and Figure 4—figure supplement 9).

6. The heatmaps (eg., Figure 5-SI5) look similar at a glance. Is there another way to present the data and show the differences in specificity in a more conspicuous manner? Bar graph – which is more tedious- may be a better choice especially for important positions. Grouping residues based on their physiochemical properties may also simplify the profile pattern and make it easier to read and understand.

For the X_5_-Y-X_5_ datasets, we added sequence logos to the figure supplements to accompany heatmaps. We have chosen not to change the pTyr-Var data visualization to logos, both for space considerations, and because we feel that they reflect the data more clearly. We note that the amino acids in our heatmaps have already been ordered in one way that correlates with their physiochemical properties.

7. The inclusion of Tyr in the X positions of the X5-Y-X5 library may obscure the specificity patter as Tyr at any position may be phosphorylated. I'd suggest repeating a couple of TKs using a library that contains only a central Tyr.

We agree that the inclusion of non-central tyrosines could have potentially obscured the specificity patterns. We re-analyzed our data by filtering out peptides with more than one tyrosine residue. We see that the specificity patterns are retained for each tyrosine kinase. This is now reflected in the main text and in figure supplements.

8. The AA read counts (eg. Figure S1) varied widely, why? Does this have an effect on the screening?

The variance in read counts are primarily dependent on the quality of the uniformity of the degenerate oligonucleotide mixtures used to construct the library, but they also reflect codon redundancy in an NNS context (some amino acids are still encoded by 2 or 3 codons). The amino acid read counts for each position can be measured with high reproducibility (see the revised Figure 1—figure supplement 1). The range of frequency values are also narrow; almost all position-specific amino acid frequencies within 5-fold of the mean value (see the revised Figure 1—figure supplement 1). The variation in read counts does not have an effect on the screen because we normalize our read counts in a selected sample with that of an unselected (input) sample.

9. Figure 5-SI1, provide correlation coefficient; why is the apparent correlation for Abl poorer compared to EphB1 when the former appears to have a more defined specificity pattern?

The correlation coefficient has been included in the revised figure (now Figure 5-supplement figure 2). We do not know exactly why the apparent correlation for Abl is poorer compared to the other tyrosine kinases. We speculate that inter-residue coupling may play more of a role in the mutational effects observed in the screen than the other tyrosine kinases.

10. Figure 6 – SI6 – The orthogonal specificity pattern for the CTK kinase and SH2 domains seems at odds with the established notion that the CTK kinase and SH2 domain specificities are related., eg, the pTyr sites created by the kinase are preferably bound by its own SH2 domain or closely related SH2 domains (Songyang et al. Nature 1995; 373:536-9). This needs to be discussed.

We were also initially surprised by the orthogonality in specificity between the kinase and SH2 domain of c-Src, based on the mentioned Songyang et al. paper. However, we noticed our c-Src kinae and SH2 domain specificity patterns matched previously reported data (see the Scansite database). The major difference for c-Src appears to be that the kinase domain prefers a +3 Phe, whereas the SH2 domain prefers a +3 aliphatic residue. By contrast, much of the old data in this field focused on c-Abl, where both the kinase and SH2 domains have a distinctive +3 Pro preference. This is now explicitly addressed in the main text.

Reviewer #3 (Recommendations for the authors):Manuscript presentation:1. The introduction provides an overview of existing methods for determining kinase specificity. However, the authors neglect to mention a couple of approaches that arguably best rival methods employed in the current manuscript from the standpoint of identifying context-dependent selectivity. These include MS-based approaches that use proteome-derived peptide libraries, from either protease digests of cell lysates or genetically encoded libraries (the cited Barber et al. paper is mischaracterized as performing analysis on cell lysates, and these papers were not cited: Kettenbach, et al. PMID: 22633412, Douglass et al. PMID: 22723110, Imamura et al. PMID: 24869485 and Xue, et al. PMID: 22451900). In addition, an approach using yeast surface display in which Tyr kinases are targeted to the secretory pathway deserves mention (Taft et al. PMID: 31339688). The authors could also consider less extensive referencing of historical methods from the 1990s that are no longer used.

The manuscript was edited to include the relevant citations and also omit a few older ones. We have also altered the phrasing describing the Barber et al. to indicate that some of these studies have been done using purified genetically-encoded peptide libraries.

Data availability:2. I could not find the deep sequencing data for kinase selections of the X5-Y-X5 library. Did they apply a cutoff for the number of read counts to be included in their analysis?

All of our fastq and fasta files are now freely accessible in this publicly accessible dataset: https://doi.org/10.5061/dryad.0zpc86727. A cutoff for the number of read counts was not included in the analysis of the X_5_-Y-X_5_ library. When sequencing, we aimed for around a million reads per sample and that determined the number of read counts for each amino acid in our analysis. You can see an example of the read counts per amino acid per position in Figure 1—figure supplement 1, which shows thousands of reads per feature (except at the central tyrosine, where other amino acids were omitted by design).

Technical points:3. Two versions of the X5-Y-X5 library were made – one with a strep tag and one with a myc tag. Were both subjected to quality control sequencing as shown in Figure 1 supp 1, and which is shown in the table? Were both of them used for screens?

Both the myc and strep-tag versions of the X_5_-Y-X_5_ libraries were sequenced for quality control and were found to be comparable in quality. The data shown in Figure 1—figure supplement 1 was obtained using the strep-tag library. For the kinase domain screens, we used the myc-tagged X_5_-Y-X_5_ library to avoid any potential background binding between strep-tag and avidin bead. We used the strep-tagged library for the SH2 domain screens since our protocol was modified so that the avidin beads were already saturated with SH2 domains, although both libraries work fine in this format.

4. It is not clear that the authors estimated the representation of the X5-Y-X5 library other than from the standpoint of pooled representation of each residue at each position ("1-10 million unique peptide sequences" were observed by sequencing). How many transformants were recovered during library cloning, and how does this compare to the number of sequencing reads used for quality control?

Based on our estimations, we recovered ~40,000,000 transformants for the X_5_-Y-X_5_ library during library screening. We get ~1,000,000 reads per sample by deep sequencing, which does not cover the number of possible sequences in the library. While each replicate samples a different subset of the library, we find that the distribution of amino acid frequencies is conserved across replicates (see the correlation graph in Figure 1—figure supplement 1 for an example). The way we have estimated the total sequence diversity of the library is through an analysis of the unique sequences observed across all of our input and selected sequencing runs. Over the course of dozens of sequencing runs with the X_5_-Y-X_5_ library, we have observed roughly 10 million unique translated sequences.

5. Selection of Tyr residues from the X5-Y-X5 library are likely due to "off target" phosphorylation of the non-central residue. This could muddy the specificity analysis as there would contribution of other selected residues at the "wrong" positions. For the Tyr-Var library, the authors avoided this issue by including Tyr to Phe substitutions. Do any of the results for X5-Y-X5 change if they exclude all peptides with Tyr at a non-central position?

As mentioned in the responses to previous reviewers’ comments, we amended our X_5_-Y-X_5_ analysis to exclude peptides with tyrosines at non-central positions. We observed that this amendment of our analysis did not significantly alter the specificity profiles obtained for each kinase.

6. Based on the concentration bacterial cells used in the kinase reactions, it seems likely that the authors are observing single turnover rather than multiple turnover kinetics. This could explain some of the differences between the results described here and prior analyses with the same kinases. Can the authors estimate the effective substrate concentration in their experiments? Can they comment on the potential impact of single vs multiple turnover kinetics on their results?

We have also suspected that the concentration of bacterial cells might dictate what the resulting consensus sequences are and how they differ from other reported consensus sequences. In the past, we have conducted experiments with the library at ~10-fold lower cell densities and do not see a major difference in specificity. Unfortunately, it is difficult to achieve higher cell densities. In our experiments, the cells are approximately at an OD600 of 1, which equates to round 10^9^ cells/mL (1-2 pM cells). Even at a surface display density of 1,000 or 10,000 molecules per cell, this could put the kinase (500 nM) in excess of substrate, putting us in a single-turnover regime. Notably, both the kinase and substrate concentrations are significantly below typical tyrosine kinase-substrate K_M_ values. Taken together, it might be the case that our screens are reporting on sequence parameters that dictate enzyme-substrate complex formation and the phosphoryl transfer reaction, but not product release, which is critical for a high k_cat_ value. Strangely, our Src consensus peptide has a higher k_cat_ value than previously reported SrcTide peptides, although it is unclear if those peptides were designed from screens done in the single- or multi-turnover regime.

7. For Tyr kinases that were analyzed with both libraries, how do the specificity matrices derived from the Tyr-Var data (Figure 4 – Figure supp 5) compare to those from the X5-Y-X5 data (Figure 1B and Figure 2A)?

For each kinase analyzed, specificity profiles obtained from both the pTyr-Var and the X_5_-Y-X_5_ datasets were similar (Figure 1B, 2A, Figure 4—figure supplement 5). A comparison of the position-specific amino acid enrichments from each library, excluding multi-Tyr sequences, is shown above, in response to another reviewer. In the manuscript, we chose to emphasize the specificity profile obtained using the X_5_-Y-X_5_ library because the amino acid composition of the peptides in this library is not biased. In contrast, the amino acid composition of peptides in the pTyr-Var library is constrained by the ~10,000 sequences derived from the human proteome. So, we reason that any discrepancies in the specificity profiles between the pTyr-Var and X_5_-Y-X_5_ library may be due to biases from the pTyr-Var library itself.

8. The observation that there is context dependence – the impact of a given amino acid substitution differs depending on the surrounding sequence – is not surprising given the author's previous work. However, the specific example shown here should be interpreted with caution, as undersampling may impact what is construed to be the "average" signal in the X5-Y-X5 library. It would be most convincing if there were examples where the impact of a given amino acid substitution were observed on two different peptide sequences, i.e. where enrichment scores for both the "WT" and variant sequence could be calculated. One would also want to see some verification by in vitro kinase assay that the same substitution caused different effects on reaction rates in the context of two different peptides.

We agree that the specific example in our paper would be most convincing if we observed the opposite effect of the same amino acid substitution in a different sequence context. To address this, we made -2 proline and -2 serine mutations to the Src consensus peptide sequence and tested the activity of Src against these peptides. We found that the mutations had the same effect to what we would have predicted from the X_5_-Y-X_5_ library and this effect is opposite to what we observed in our specific example. This new data can be found in Figure 5 of the revised manuscript.

Reviewer #4 (Recommendations for the authors):1. Concerning the peptide design in their random library (sequence: X5-Y-X5, where X = 20 natural amino acids falling within their 'NNS' codon constraint), neighboring tyrosines (TAC codons) theoretically account for ~3% (1/32) of residues at all random positions, meaning that as much as 30% of the peptide pool contains at least two potential tyrosine phosphoacceptors per peptide. For the heatmap motifs presented in Figure 1, Figure 2, Figure 4—figure supplement 5, and Figure 7, we can observe strong enrichment of tyrosines at multiple positions, especially for JAK2 and the RTKs, and it is not certain whether this is due to positional selection (that facilitates phosphorylation of the tyrosine at position zero) or direct phosphorylation of the neighboring tyrosine. This double phosphorylation effect may limit interpretation of the motif heatmaps presented in the figures that assume phosphorylation occurs only at the tyrosine at position zero. Have the authors considered generating separate heatmaps that omit the subset of enriched peptides containing two or more tyrosines? That might reduce background and further improve the quality of the data.

The strong enrichment of tyrosines at multiple positions, especially for JAK2 and the RTKs, could potentially be evidence for phospho-priming. This has also been observed in past screens with another receptor kinase, EGFR (PMIDs: 26551075, 30012625). As mentioned in the response to previous reviewers’ comments, we amended our X_5_-Y-X_5_ analysis to exclude peptides with tyrosines at non-central positions (figure supplements associated with Figures 1 and 2). This amendment of our analysis did not significantly alter the specificity profiles obtained for each kinase.

2. Figure 6B: Given that the authors have demonstrated essentially complete phosphorylation of the displayed peptides by their kinase cocktail in Figure 6—figure supplement 2, it seems more appropriate to replace the label above the heatmaps in 6B "position relative to tyrosine" with "position relative to phosphotyrosine". Moreover, this indicates that they have probably phosphorylated most of the neighboring tyrosines in the displayed peptides containing two or more tyrosines. How would the heatmap motifs for the SH2 domains appear if the authors excluded the subset of enriched peptides containing two or more (likely to be phosphorylated) tyrosines?

We agree with the reviewer and have replaced the label on the heatmaps in Figure 6B with “position relative to phosphotyrosine.” We also constructed the heatmap profiles of the SH2 domains with the exclusion of peptides with non-central tyrosine residues and saw no difference in the specificity (Figure 6-supplement figure 3). There does not appear to be any significant impact of omitting sequences with non-central tyrosine (probably phosphotyrosine) sequences from these analyses.

3. The authors have not included a single sequence logo to represent their phosphorylation site motifs or pY binding motifs. Is there a specific reason for this? Researchers less familiar with these approaches generally have an easier time interpreting sequence logos than heatmaps.

The specific reason for this is that our group prefers heatmaps over logos. Nonetheless, we appreciate the reviewer’s point that others might find logos more intuitive, so we have added sequence logos for the X_5_-Y-X_5_ screens to the figure supplements. In addition, we have provided the numerical position-weighted matrices used to make every heatmap and logo in the paper (for both libraries) as source data files, so that other researchers can render the data in the way that best suits their needs.

4. An advantage of the authors' platform over alternative approaches is its potential to decipher pairwise interactions between amino acids on substrate peptides. The authors report that, depending on the kinase, 5-15% of all significant mutations in the pTyr-Var screen (which are pairwise comparisons of single substitutions in fixed sequences) had the opposite effect of predictions from the randomized library (ensembles). And indeed, the correlation plots in Figure 5 —figure supplement 1 show quite a bit of scattering and hence negatively correlated sites between their predictions and their results. The authors explored one example of this for SRC, which preferred proline over serine at -2 in the randomized library, yet it more efficiently phosphorylated a site containing -2 serine versus proline in the context of the sequence XEYSFK, where X is the substituted position. Interestingly, this is the opposite effect of what has been previously published and referred to by the authors for EGFR, where -2 proline was preferred over serine in the context of -1 acidic residues (Cantor et al. 2018), indicating that these pairwise rules may not only be unique to specific kinase groups but mutually exclusive between different groups. Are the authors able to identify kinase-specific trends from their datasets for the pairwise selection of amino acids at positions -1 and -2 that negatively correlate with predictions?

The opposite coupling effect for Src and EGFR is a very good point. This likely reflects a combination of the different sequence specificities for Src and EGFR, coupled with differences in what kinds of peptide conformations are tolerated by each kinase. This definitely warrants a deeper structural investigation. Unfortunately, there was not enough “depth” across mutations and sites in the pTyr-Var library to infer any significant rules for coupling (certainly not kinase-specific rules). We did notice that many of the mutations that could not be predicted well using the X_5_-Y-X_5_ library were at the -2, -1, +1 and +2 positions, further suggesting a role for local sequence context dictating peptide conformation.

5. Cannot access the code used to process and analyze the deep sequence data in the GitHub repository: (https://github.com/nshahlab/2022_Li-etal_peptide-display)

Due to a formatting error, a hyphen was missing from the link. Here is the correct link: https://github.com/nshahlab/2022_Li-et-al_peptide-display

6. Based on the format of this journal, this work seems appropriate as a "Research Advance."

We agree that this paper could be a Research Advance, and indeed a precursor to this paper (*eLife* 7:e35190) was a Research Advance coupled with (*eLife* 5:e20105). However, given the significant expansion in technical scope for the peptide screening platform described in this paper, relative to the previous articles, we felt that a “Tools and Resources” article was appropriate. We defer to editors as to whether this work should be reported as a “Research Advance” or a “Tools and Resources” article.